# One Model for All Tasks: Leveraging Efficient World Models in Multi-Task Planning

**Yuan Pu**[1,*]    **Yazhe Niu**[1,2,*]    **Jia Tang**[1,3]    **Junyu Xiong**[1,4]
**Shuai Hu**[5]    **Hongsheng Li**[2,6,†]

[1]Shanghai Artificial Intelligence Laboratory    [2]The Chinese University of Hong Kong MMLab
[3]Nanjing University of Aeronautics and Astronautic
[4]University of Science and Technology of China
[5]Novosibirsk State University    [6]Centre for Perceptual and Interactive Intelligence

## ABSTRACT

In heterogeneous multitask decision-making, tasks not only exhibit diverse observation and action spaces but also vary substantially in their underlying complexities. While conventional multitask world models like UniZero excel in single-task settings, we find that when handling a broad and diverse suite of tasks, gradient conflicts and the loss of model plasticity often constrain their sample efficiency. In this work, we address these challenges from two complementary perspectives: the single learning iteration and the overall learning process. First, to mitigate the gradient conflicts, we systematically investigate key architectural designs for extending UniZero. Our investigation identifies a Mixture-of-Experts (MoE) architecture as the most effective approach. We demonstrate, both theoretically and empirically, that this architecture alleviates gradient conflicts by routing task-specific representations to specialized sub-networks. This finding leads to our proposed model, *ScaleZero*. Second, to dynamically allocate model capacity throughout the learning process, we introduce an online Dynamic Parameter Scaling (DPS) strategy. This strategy progressively integrates LoRA adapters in response to task-specific progress, enabling adaptive knowledge retention and parameter expansion. Evaluations on a diverse set of standard benchmarks (Atari, DMC, Jericho) demonstrate that ScaleZero, utilizing solely online reinforcement learning with one model, performs on par with specialized single-task agents. With the DPS strategy, it remains competitive while using just 71.5% of the environment interactions. These findings underscore the potential of ScaleZero for effective multitask planning. Our code is available at https://github.com/opendilab/LightZero.

## 1 INTRODUCTION

Unified world models (Reed et al., 2022; Lee et al., 2022; Gallouédec et al., 2024) represent a significant step towards generalist agents, offering a cohesive framework for multi-modal perception (Zhang et al., 2022), long-horizon prediction (Ni et al., 2024), and decision-making *by learning a shared latent space representation*. By encoding the environment into a compact latent space, these learned representations enable effective planning. Algorithms like Monte Carlo Tree Search (MCTS) (Świechowski et al., 2023) are well-suited for this task, as they can perform lookahead efficiently within this abstract representation (Schrittwieser et al., 2019; 2021). MCTS excels at dynamically balancing exploration and exploitation, a combination that has achieved well-established success in homogeneous task domains like board games (Schrittwieser et al., 2019; Ye et al., 2021). Extending this paradigm to heterogeneous multitask reinforcement learning (MTRL) (Yu et al., 2024) is a key objective for creating generalist agents. However, this ambition is impeded by a formidable obstacle: training a single shared model on diverse tasks with potentially conflicting dynamics and objectives is notoriously difficult (Cho et al., 2024). While prior work (Sun et al., 2022; Fernando et al., 2023; Shi et al., 2023) has attempted to mitigate inter-task interference, these methods have

---

*These authors contributed equally to this work.
†Corresponding author.

been predominantly studied in supervised learning and model-free RL settings. The challenges and dynamics of multitask planning within a shared world model remain largely unexplored.

Within this specific domain of multitask planning, our work identifies and addresses two critical, intertwined obstacles not fully resolved by existing methods: (1) *Representational bottlenecks and plasticity collapse:* In diverse multitask settings, a shared model is susceptible to gradient dominance from simpler, faster-converging tasks (Cho et al., 2024). This imbalance leads to representation interference (Yu et al., 2020; Bejnordi et al., 2024), where the learning signals for more complex tasks are suppressed. This culminates in a progressive loss of network plasticity (Dohare et al., 2024; Todorov et al., 2025)—*the fundamental ability of a model to adjust its parameters to learn from new data*—imposing a hard ceiling on the model's overall learning capacity and leading to performance collapse on challenging tasks. (2) *Static resource allocation:* Conventional architectures employ a uniform, one-size-fits-all forward pass, applying the same learning resources (e.g., data collection and model updates) to every task irrespective of its intrinsic difficulty. This static strategy results in profound computational inefficiency, as resources are squandered on converged or show diminishing returns, instead of being directed toward those that require further learning. To dissect these issues, we begin with a quantitative diagnosis using UniZero (Pu et al., 2024), a contemporary unified world model, as a representative testbed. We train a single model via online reinforcement learning across eight canonical Atari games and observe a stark failure pattern, as shown in Figure 1. While the model rapidly masters simple tasks like *Pong*, it exhibits initial progress followed by a catastrophic performance collapse on complex games with disparate dynamics and visual styles, such as *Seaquest*, whose complex exploration demands starkly contrast with the reactive gameplay of tasks like *Pong*. We provide quantitative evidence linking this failure to specific internal model dynamics (Lyle et al., 2022): an uncontrolled inflation of the latent state norm and a corresponding spike in the dormant neuron ratio (Sokar et al., 2023) within the transformer backbone, signaling a collapse in model plasticity, rendering the network unable to adapt to new information from challenging tasks.

Based on this diagnosis, we tackle these challenges from two complementary perspectives. First, as an internal, architectural solution, we undertake a principled exploration of the design space across the core dimensions *input representation*, *model architecture*, and *optimization strategy*. This systematic investigation, which evaluates five key axes—task conditioning, encoder architectures (ResNet (He et al., 2016) vs. ViT (Dosovitskiy et al., 2020)), latent normalization schemes, backbone design, and multitask optimization strategies (Fernando et al., 2023)—culminates in a new, powerful model we term *ScaleZero*. Among the evaluated design choices, the integration of a sparse Mixture-of-Experts (Shazeer et al., 2017) backbone yields the most significant gains by fundamentally addressing plasticity collapse. To provide a deeper understanding of this key architectural choice, we present a dedicated empirical and theoretical analysis in Section 5.3, explaining MoE's effectiveness in multitask planning. Validated on a comprehensive suite of online RL benchmarks spanning diverse modalities—including 26 visually-complex Atari games (Bellemare et al., 2013), 18 state-based DeepMind Control (DMC) Suite tasks (Tunyasuvunakool et al., 2020), and 4 text-based Jericho environments (Hausknecht et al., 2020)—*ScaleZero* robustly matches and often surpasses the performance of single-task expert agents.

Second, as an external, procedural strategy to optimize the overall learning process, we introduce *Dynamic Parameter Scaling (DPS)*, a novel online mechanism that couples model capacity to learning progress. DPS adaptively curates the set of active tasks based on real-time return feedback and orchestrates a phased expansion of model capacity by injecting lightweight LoRA adapters (Hu et al., 2022). By strategically freezing previously trained parameters, DPS creates a curriculum of model capacity that directs computational resources where they are most needed. Our experiments demonstrate that when augmented with this strategy, our method achieves performance nearly on par with single-task agents while reducing total environment interactions by around 28.5% on the DMC benchmark, offering a superior trade-off between final performance and computational budget.

Our main contributions are summarized as follows: (1) We provide the quantitative diagnosis of plasticity collapse in unified world models within heterogeneous MTRL, establishing a concrete link between performance degradation and internal learning dynamics. (2) We conduct a systematic architectural exploration that yields *ScaleZero*, a unified world model that demonstrates exceptional performance and generalization across distinct tasks from three distinct benchmarks. (3) We propose *Dynamic Parameter Scaling*, an adaptive training strategy that dynamically allocates model capacity and computational resources, reducing total environment interactions by around 28.5%.

## 2    RELATED WORK

**MCTS with Learned World Models.**    Planning in a learned latent space, popularized by MuZero (Schrittwieser et al., 2019), is a dominant RL paradigm (Hubert et al., 2021; Antonoglou et al., 2021; Danihelka et al., 2022; Xuan et al., 2024; Niu et al., 2024). Recent works have integrated Transformers to enhance representational capacity (Micheli et al., 2022; Pu et al., 2024; Zhou et al., 2024), achieving state-of-the-art performance in single-task domains. However, their monolithic design is a critical liability in heterogeneous multitask settings, where they suffer from representational interference and plasticity collapse. Our work directly confronts this architectural bottleneck.

**Multitask Reinforcement Learning.** MTRL aims to improve data efficiency by sharing knowledge across tasks (Vithayathil Varghese & Mahmoud, 2020; Kumar et al., 2022; Hansen et al., 2023; Lee et al., 2022; Gallouédec et al., 2024). Existing methods mitigate interference through architectural solutions like task-specific modules (Schmied et al., 2023; Sun et al., 2022) or optimization-based approaches that manage gradient conflicts (Fernando et al., 2023; Ma et al., 2023). These strategies, however, have been predominantly studied outside of latent-space planning, where the unique challenge of disentangling dynamics prediction remains largely unaddressed.

**Sparse and Parameter-Efficient Architectures.** We draw inspiration from two complementary paradigms. Sparse MoE models offer a natural architectural prior for multitask specialization by enabling conditional computation (Dai et al., 2024; Obando-Ceron et al., 2024). Concurrently, parameter-efficient methods like LoRA (Hu et al., 2022) provide a lightweight mechanism for adaptation. However, conventional LoRA approaches are typically restricted to static, offline fine-tuning (Agiza et al., 2024; Huang et al., 2023). To adapt these principles to non-stationary online data streams, our approach diverges from standard methods that depend on predefined task boundaries. Instead, we introduce a mechanism that autonomously allocates parameter space in response to distinct distribution shifts. By unifying MoE with this adaptive DPS strategy within a *transformer-based world model*, we create a system that effectively balances architectural specialization with dynamic plasticity for large-scale MTRL. See Appendix F for more discussion.

## 3    BACKGROUND

### 3.1    MCTS WITH LEARNED WORLD MODELS

Monte Carlo Tree Search is a powerful planning algorithm that has demonstrated remarkable success in domains with known rules (Silver et al., 2016; 2017). Methods employing MCTS for planning within a learned latent space represent the state-of-the-art in complex sequential decision-making (Schrittwieser et al., 2019; Ye et al., 2021). These approaches learn a world model comprising three components: (i) a *representation model* $h_\theta$ that encodes observation-action history into a latent state $z_t$; (ii) a *dynamics model* $g_\theta$ that predicts the next latent state and reward $(\hat{z}_{t+1}, \hat{r}_t) = g_\theta(z_t, a_t)$; and (iii) a *prediction model* $f_\theta$ that outputs a policy and value $(p_t, v_t)$ to guide the MCTS planner. Recent architectures, exemplified by UniZero (Pu et al., 2024), unify these components into a single, monolithic Transformer. This design enables end-to-end optimization via a composite loss that aligns the model's predictions with targets derived from the MCTS planner: a policy target ($\pi_t$) based on the root node's visit counts, and a bootstrapped TD value target ($\hat{v}_t^{targ}$) (Schrittwieser et al., 2019). These are combined with objectives for predicting the reward ($r_t$) and the next latent state ($z_{t+1}$):

$$\mathcal{L}_{\text{Unified}} = \sum_{t=0}^{H-1} \left( \mathcal{L}_{\text{value}}(v_t, \hat{v}_t^{targ}) + \mathcal{L}_{\text{policy}}(p_t, \pi_t) + \mathcal{L}_{\text{reward}}(\hat{r}_t, r_t) + \mathcal{L}_{\text{dynamics}}(\hat{z}_{t+1}, \text{sg}(z_{t+1})) \right),$$

(1)

where $H$ is the context length, $\text{sg}(\cdot)$ means stop-gradient. While sample-efficient and hyperparameter-insensitive for single-task training, this unified scheme exposes a critical vulnerability: in multitask settings, the shared transformer backbone is updated by an aggregated gradient. This shared update mechanism becomes the nexus of interference, impeding a single model's ability to learn multiple complex task and leading directly to the *plasticity collapse* we identified in our diagnosis.

### 3.2    GRADIENT CONFLICT IN UNIFIED ARCHITECTURES

The challenge of MTRL is to train a single, task-conditioned policy $\pi_\theta(a|s, k)$ on a distribution of $K$ tasks, $\{\tau_1, \ldots, \tau_K\}$. Even within a single task, the composite losses of a world model (e.g.,

for dynamics, reward, and policy) can conflict, creating a challenging multi-objective optimization problem (Ma et al., 2023). In MTRL, this challenge is compounded by inter-task gradient conflicts. In a unified architecture with shared parameters $\theta$, the total loss is the sum of individual task losses, $\mathcal{L}_{\text{MTRL}}(\theta) = \sum_{k=1}^{K} \mathcal{L}^k$, where $\mathcal{L}^k$ is the loss from Eq. 1 for task $k$. Consequently, the learning dynamics are driven by the sum of per-task gradients: $G_{\text{total}} = \sum_{k=1}^{K} g_k$, where $g_k = \nabla_\theta \mathcal{L}^k$. The core problem, known as *gradient conflict* (Liu et al., 2021), arises when these per-task gradients are misaligned, a condition quantified by their cosine similarity: $\cos(g_i, g_j) < 0$. A negative similarity indicates that an update improving performance on task $i$ actively degrades performance on task $j$. In a monolithic architecture where all core parameters are shared, frequent gradient conflicts lead to destructive interference (Sodhani et al., 2021). This dynamic forces the model into a suboptimal compromise, creating the representational bottlenecks and catastrophic performance drops characteristic of plasticity collapse (Dohare et al., 2024).

### 3.3 ARCHITECTURAL PARADIGMS FOR MITIGATING INTERFERENCE

To overcome the limitations of monolithic designs, architectural solutions have been proposed to enable parameter specialization and mitigate interference. Two prominent paradigms are particularly relevant. Sparse *Mixture-of-Experts* replaces the dense feed-forward network in a Transformer block with a set of $N$ parallel "expert" networks (Cai et al., 2025). For each input, a trainable gating function $G(x)$ sparsely selects a small subset of experts (e.g., top-k) to process the token: $\text{MoE}(x) = \sum_{i=1}^{N} G_i(x) \cdot \text{Expert}_i(x)$. This conditional computation creates specialized pathways within the model, allowing different inputs (e.g., from different tasks) to be processed by distinct subsets of parameters, thereby reducing interference and increasing model capacity without a proportional rise in computational cost. Recent designs also explore hybrid approaches, such as including a shared expert alongside specialized ones, to balance generalization and specialization (Dai et al., 2024). *Low-Rank Adaptation* offers a parameter-efficient method for adapting large pre-trained models (Hu et al., 2022). Instead of fine-tuning the entire weight matrix $W_0$, LoRA freezes $\theta_B$ and injects a trainable, low-rank "update" matrix, $\Delta\theta = BA$. The modified forward pass becomes $(\theta_B + \alpha BA)x$. By training only the small low-rank factors $(A, B)$, LoRA can efficiently learn task-specific modifications while preserving the general knowledge in the pre-trained weights. This approach has proven effective for creating extensive multitask capabilities on top of a single base model, where different LoRA modules can be composed to handle diverse tasks (Huang et al., 2023).

## 4 METHOD

We begin in Sec. 4.1 by quantitatively diagnosing *plasticity collapse* in multitask training. To address this, Sec. 4.2 presents a systematic design space exploration across five axes—which culminates in our proposed model, *ScaleZero*. Finally, to resolve the static allocation issue, we introduce *Dynamic Parameter Scaling* in Sec. 4.3, a LoRA-based strategy that adaptively allocates model capacity.

### 4.1 DIAGNOSING PLASTICITY COLLAPSE IN MULTITASK PLANNING

To empirically substantiate the challenges of representational bottlenecks and plasticity collapse, we conduct a quantitative analysis of a unified world model's internal dynamics during multitask training. We track two key metrics indicative of network plasticity degradation: (1) $DormantRatio(l) = \frac{1}{N_l} \sum_{i=1}^{N_l} \mathbf{1}! \left( |h_i^l| \leq \epsilon \right)$ : This metric quantifies the proportion of neurons whose activation magnitude $|h_i^l|$ falls below a threshold $\epsilon$. A high ratio signifies a loss of active pathways and reduced network plasticity. (2) *Latent State Norm:* We compute the average L2 norm of the latent state ($|z_t|_2$). An uncontrolled inflation of this norm is a well-known indicator of training instability (Team et al., 2025).

Diagnostic experiments on UniZero, conducted across a multitask Atari8 suite (Appendix B), reveal a critical failure mode in complex learning scenarios. While simpler tasks like *Pong* achieve stable convergence, more complex ones such as *Seaquest* exhibit initial learning followed by a catastrophic performance collapse (Figure 1). This collapse coincides with a sharp increase in the dormant neuron ratio and an inflated latent state norm. The representational nature of this failure is corroborated by a marked decline in the feature effective rank (Dohare et al., 2024), indicating a degradation of the learned feature space (Figure 5). We attribute this failure mode to two intertwined factors: (1) *Gradient Competition*, where gradients from simpler tasks overwhelm those from complex ones,

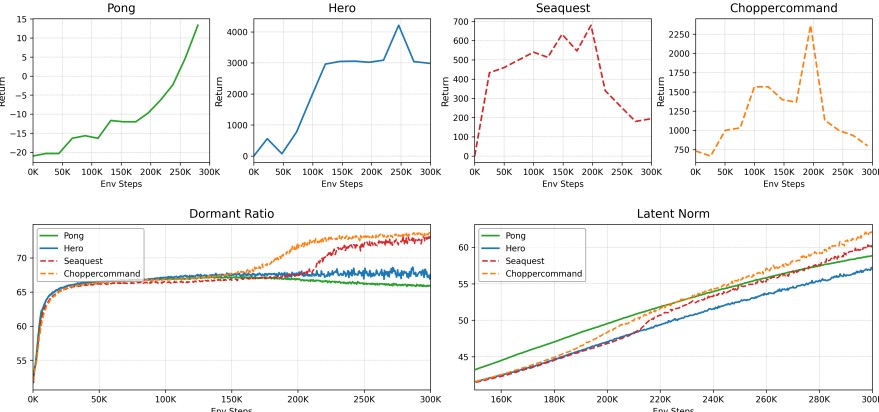

**Figure 1: Plasticity collapse in the baseline (UniZero) on a multitask Atari benchmark.** While simple tasks like *Pong* and *Hero* show stable learning, complex tasks such as *Seaquest* and *ChopperCommand* suffer a catastrophic performance collapse in later training (Top). This failure is precisely correlated with a sharp spike in the dormant neuron ratio of the transformer (Bottom Left) and an uncontrolled inflation of the latent state norm (Bottom Right), empirically validating the link between external performance and internal learning dynamics.

and (2) *Representation Interference*, where the shared network fails to maintain diverse features, leading to a representational bottleneck. Based on this diagnosis, our method tackles plasticity collapse from two complementary perspectives: (1) an *internal*, architectural solution, *ScaleZero* (Section 4.2), designed to mitigate interference within a single learning iteration. (2) an *external*, procedural strategy, *Dynamic Parameter Scaling* (Section 4.3), which optimizes resource allocation across the entire learning process.

## 4.2    ARCHITECTURE DESIGN OF SCALEZERO VIA PRINCIPLED EXPLORATION

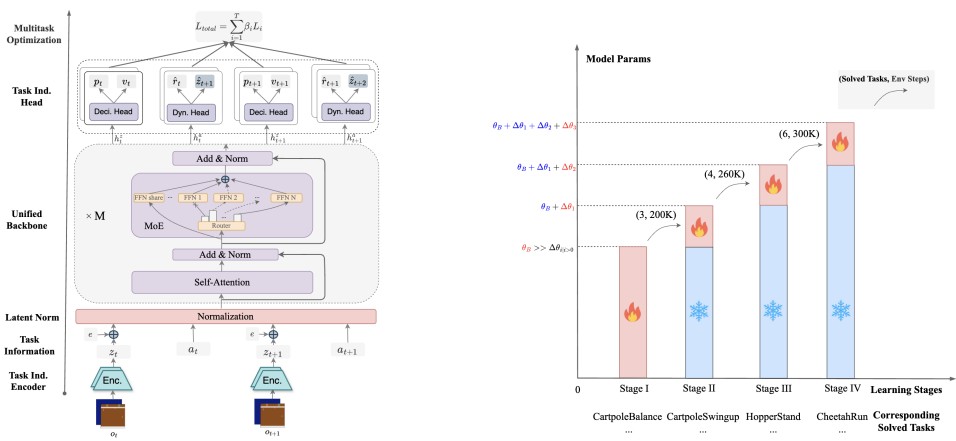

**(a)** Design Space of UniZero for Multitask learning        **(b)** Dynamic Parameter Scaling

**Figure 2: (a)** A systematic exploration of the UniZero design space across five axes: task conditioning, encoder architecture, latent normalization, backbone design, and optimization. This investigation informs the design of our proposed *ScaleZero* model. **(b)** A conceptual diagram of *Dynamic Parameter Scaling (DPS)*. DPS progressively expands model capacity by injecting LoRA adapters in stages, triggered by learning progress. This creates a curriculum of model, directing resources toward unsolved tasks while preserving existing knowledge.

Motivated by the above diagnosis of plasticity collapse, we conduct a principled exploration of the UniZero design space to forge a robust solution on Atari8 multitask benchmark (see Appendix B for experimental details). We systematically evaluate five key design axes (Figure 2a) drawn from three core dimensions (input representation, model architecture, and optimization strategy): *task conditioning*, *encoder architecture*, *latent normalization*, *backbone design*, and *optimization strategy*. Specific implementation details are provided in Appendix A.3. The comprehensive results, summarized in Figure 3, reveal that the integration of a MoE backbone provides a uniquely significant and robust

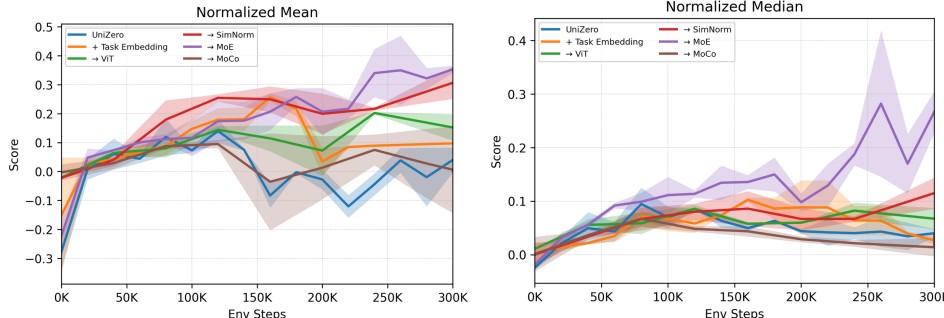

**Figure 3: Performance impact of architectural modifications on the Atari8 multitask benchmark.** This ablation across the UniZero design space reveals that replacing the Transformer backbone with a *Mixture-of-Experts architecture yields the most significant and consistent performance gains*. In contrast, other interventions, with the partial exception of SimNorm, provide marginal or inconsistent benefits. These results underscore the centrality of the MoE's conditional computation in overcoming the limitations of a shared, dense backbone.

performance uplift, directly countering the failure modes identified in our initial diagnosis. This is further corroborated by the complete plasticity metric analysis in Figure 7, which demonstrates that the MoE-based model consistently maintains healthy latent dynamics (i.e., lower dormant ratios and stable latent norms) across all tasks. Below, we detail the findings of each axis.

**Explicit Task Conditioning.** A shared model processing raw observations may face ambiguity in discerning the underlying task, especially when initial states are similar across environments (Rakelly et al., 2019; Sodhani et al., 2021). To test if providing an explicit task identity $k$ could resolve ambiguity, we concatenate a learnable task embedding $e_k$ to the latent state $z$. While this approach accelerates initial convergence on simpler tasks, it offers marginal long-term benefits for complex ones and fails to mitigate plasticity collapse, as evidenced by minimal improvement in metrics like the dormant neuron ratio (Figure 7). This indicates that such input-level task conditioning are insufficient to resolve deep-seated representational conflicts in online MTRL.

**Encoder Architecture (ResNet vs. ViT).** We contrast a standard ResNet-like encoder in Pu et al. (2024) with a Vision Transformer to assess the encoder's impact on visual representation quality. On the Atari8 benchmark, the choice of encoder does not yield a significant performance difference. We attribute this to the limited visual heterogeneity in this specific task suite, but posit that the scalability and powerful feature extraction of ViT are critical for broader, more diverse multitask domains.

**Latent Normalization (LayerNorm vs. SimNorm).** To directly combat the observed latent norm inflation, we evaluate SimNorm (Hansen et al., 2023) against the standard LayerNorm (Ba et al., 2016). SimNorm, which projects latents onto a simplex to enforce a norm constraint, effectively stabilized training and prevented performance collapse, consistent with its reported success in less heterogeneous settings (Hansen et al., 2023). However, in our highly diverse multitask setup, this hard constraint appears to curtail representational expressiveness, leading to suboptimal final performance. This highlights a critical trade-off between stability and capacity for online heterogeneous RL training.

**Mixture-of-Experts Backbone.** The most impactful modification is replacing the dense feed-forward networks with sparse MoE layers in UniZero's transformer. This conditional computation paradigm directly addresses the core challenges of multitask planning by mitigating the intertwined issues of representational interference and gradient conflict, as it routes task-specific computations and their corresponding gradient updates through distinct, specialized expert sub-networks. As validated in Figures 3 and 7, the MoE backbone yields substantial performance gains. We attribute this success to its inherent ability to mitigate gradient conflict, a mechanism we analyze in depth in Section 5.3. There, we provide both empirical validation and a theoretical analysis demonstrating that the upper bound on gradient conflict for an MoE layer is *strictly lower* than that of its dense counterpart.

**Multitask Gradient Correction.** We also explored a dynamic gradient re-weighting scheme inspired by MoCo (Fernando et al., 2023) to directly mitigate gradient interference between tasks. While this approach showed promise on some tasks, it introduced significant computational overhead by increasing per-step training time by approximately 40%, and demonstrated inconsistent efficacy across our heterogeneous task suite. Given this unfavorable performance-cost trade-off, we defer the investigation of more efficient gradient correction methods to future work.

**The ScaleZero Architecture.** The result of our systematic investigation is *ScaleZero*, a novel architecture designed as a efficient and scalable world model. It is defined by the integration of three core components: (1) a *ViT-like Encoder* for powerful and scalable feature extraction; (2) an *MoE Backbone* that uses sparse, conditional computation to increase model capacity while mitigating task interference; and (3) an explicit *Layer Normalization* applied to the encoded latent state to ensure a robust balance between stability and representational expressiveness. This core is complemented by a modular design that decouples task-specific encoders and heads, enabling flexible handling of diverse input and output modalities. As validated in Section 5, ScaleZero provides an effective blueprint for building generalist world models. Differences from *UniZero* are detailed in Appendix A.2.

## 4.3 Dynamic Parameter Scaling for Efficient Multitask Learning

To tackle the second challenge in multitask planning—static resource allocation—we propose *Dynamic Parameter Scaling (DPS)*. As illustrated in Figure 2b, this strategy departs from the conventional *one-size-fits-all* paradigm by dynamically aligning model capacity with learning progress. DPS operates on two synergistic principles: (1) *adaptive task curation*, which focuses computational effort exclusively on unsolved tasks, and (2) *progressive capacity expansion*, which injects new parameters only when necessitated by task demands. This combination establishes a "curriculum of model complexity," directing resources precisely where they are most required. DPS employs dynamic, multi-stage training to manage tasks and parameters.

- **Adaptive Task Curation:** Let $\mathcal{T} = \{\tau_i\}_{i=1}^N$ denote the full set of tasks. A task $\tau_i$ is deemed "solved" once its performance metric, $\text{Metric}(\tau_i)$, surpasses a predefined threshold $\varepsilon_i$. At any training step $t$, we maintain an active set of unsolved tasks, $\mathcal{U}_t \subseteq \mathcal{T}$. To maximize efficiency, both data collection and gradient updates are performed *only* for tasks within $\mathcal{U}_t$. Once solved, a task is removed from this active set, thereby ceasing all computational overhead associated with it.

- **Staged Capacity Expansion:** The training proceeds in $S + 1$ stages over a total of $T_{\max}$ iterations. *Stage 0 (Warm-up)* trains the base model $\theta_B$ for an initial $T_0$ iterations. This stage establishes a robust foundation by training the shared base model on all tasks to learn general-purpose representations, which are crucial for the subsequent progressive introduction of specialized modules. Triggered by learning progress, each subsequent *Stage $s \geq 1$ (Expansion)* incorporates a new, independent LoRA module, $\Delta\theta_s = B_s A_s$. Concurrently, a set of learnable scaling factors is defined to modulate the contribution of both the base model and all adapters. Specifically, a scaling factor $\alpha_0$ is associated with the base model weights $\theta_B$, while each LoRA module $\Delta\theta_j$ is assigned a corresponding scaling factor $\alpha_j$ for $j \in \{1, \ldots, S\}$. All scaling factors are initialized to 1.

**Adaptive Stage Transition Triggers.** The transition from stage $s-1$ to $s$ is triggered upon satisfying either of two criteria, balancing progress against a fixed computational budget: (1) *Progress-based Trigger:* A transition is triggered if the number of newly solved tasks during the current stage reaches a quota $Q_s$. (2) *Budget-based Trigger:* To prevent stagnation on persistently arduous tasks, a transition is forced if the iteration count in the current stage exceeds a pre-allocated limit, $\lceil (T_{\max} - T_0)/S \rceil$.

**Optimization and Parameter Isolation.** The process starts with the base weight matrices from $\theta_B$, which are pre-trained in Stage 0 and subsequently frozen. For any base matrix $W_0 \in \theta_B$, its effective weight is progressively augmented. At the beginning of stage $s \geq 1$, the effective matrix $W^{(s)}$ is defined as: $W^{(s)} = \alpha_0 W_0 + \sum_{j=1}^s \alpha_j \Delta\theta_j = \alpha_0 W_0 + \sum_{j=1}^s \alpha_j B_j A_j$. A key principle of DPS is parameter isolation. Upon advancing to stage $s$, all previously learned parameters—the backbone $\theta_B$ and adapters $\{\Delta\theta_j\}_{j=1}^{s-1}$—are *frozen*. To resolve scale ambiguity, the scaling factor $\alpha_s$ for the currently active adapter $\Delta\theta_s$ is temporarily fixed at 1. Optimization is thereby focused on the newly added LoRA module $\Delta\theta_s$, the base model scaling factor $\alpha_0$, and the set of scaling factors corresponding to previously frozen adapters, $\{\alpha_j\}_{j=1}^{s-1}$.

This strategy yields two principal advantages. First, it implements a *resource-efficient training curriculum*, where new parameters are only introduced when required by the remaining unsolved tasks. Second, by isolating new learning within dedicated adapters, DPS provides *targeted plasticity* for difficult tasks while *preventing catastrophic forgetting or negative transfer* to previously mastered skills, whose knowledge is preserved in the frozen backbone. DPS thus achieves a favorable trade-off between computational efficiency and final performance. (See Appendix A.4 for pseudocode).

## 5 EXPERIMENTS AND ANALYSIS

Our experiments are designed to validate the two central claims of this work. First, in Section 5.1, we demonstrate that the proposed *ScaleZero* effectively mitigates the representational bottlenecks and plasticity collapse issue. We assess its ability to achieve powerful performance as a single, generalist agent across a suite of heterogeneous benchmarks spanning visual, proprioceptive-based, and text-based domains (Atari (Bellemare et al., 2013), DMControl (Tunyasuvunakool et al., 2020), and Jericho (Hausknecht et al., 2020)). Second, in Section 5.2, we quantify the efficiency gains afforded by our DPS strategy, showing its effectiveness in overcoming static resource allocation. Finally, in Section 5.3, we provide a dedicated empirical and theoretical analysis of the MoE, substantiating its role as a foundational element of ScaleZero. All experiments are conducted in a purely online RL setting, without reliance on expert data. Implementation details are provided in Appendix A.

### 5.1 MULTITASK LEARNING BENCHMARKS

#### 5.1.1 ATARI 100K BENCHMARK

**Setup.** We evaluate ScaleZero on the 26 games of the Atari 100k benchmark, a standard testbed for heterogeneous MTRL due to its diverse game mechanics. We compare one multi-task (MT) ScaleZero agent against the strong single-task (ST) UniZero baseline, where 26 separate models are trained for each game. Performance is measured using the standard Human-Normalized Score (HNS) (Agarwal et al., 2021), reporting both mean and median to capture overall capability. Further experimental details are available in Appendix B.

**Results.** As clearly summarized in Table 1, ScaleZero, as a single generalist agent, achieves a higher mean normalized score than the set of 26 independently trained single-task specialists. This result demonstrates that ScaleZero effectively mitigates the negative interference—a common challenge in heterogeneous MTRL, enabling positive knowledge transfer. Notably, performance gains are particularly marked on notoriously difficult exploration tasks like *Seaquest*—a game we identified as a primary failure case for the baseline in our initial diagnosis (Figure 1) While the median score is marginally lower, influenced by a few hard-exploration games, the higher mean score confirms the overall advantage and generalization ability of our approach.

**Table 1:** Performance comparison of our multitask model, ScaleZero, against the single-task UniZero baseline across discrete (Atari) and continuous (DMControl) benchmarks.

| Algorithm | Norm. Mean | Norm. Median |
|---|---|---|
| UniZero (ST) | 0.38 | **0.21** |
| UniZero (MT) | 0.31 | 0.16 |
| ScaleZero (MT) | **0.39** | 0.16 |

**(a)** Human-normalized scores on 26 games in Atari 100k benchmark. ScaleZero achieves a higher mean score. Full training curves are in Appendix B. We omit other MT baselines as, to the best of our knowledge, no prior work has tackled all 26 games with a single online RL agent.

| Task | UniZero (ST) | ScaleZero (MT) |
|---|---|---|
| acrobot-swingup | 400.3 | **501.0** |
| cartpole-balance | 952.2 | **990.5** |
| cartpole-balance_sparse | 1000.0 | 1000.0 |
| cartpole-swingup | **801.3** | 769.0 |
| cartpole-swingup_sparse | **752.5** | 708.1 |
| cheetah-run | **517.6** | 510.9 |
| ball_in_cup-catch | **961.6** | 954.2 |
| finger-spin | **810.7** | 574.2 |
| finger-turn_easy | 1000.0 | 1000.0 |
| finger-turn_hard | 884.5 | **982.0** |
| hopper-hop | 120.5 | **138.0** |
| hopper-stand | **602.6** | 583.1 |
| pendulum-swingup | 865.6 | **866.0** |
| reacher-easy | **993.3** | 943.1 |
| reacher-hard | **988.8** | 943.5 |
| walker-run | **587.9** | 562.7 |
| walker-stand | **976.4** | 919.9 |
| walker-walk | **954.6** | 908.7 |
| Mean | **787.2** | 769.7 |
| Median | 875.1 | **887.3** |

**(b)** Raw scores on 18 DMControl tasks. ScaleZero achieves a superior median score, indicating robust generalist performance. Full training curves are in Appendix C. For reference, L2M (MT) Schmied et al. (2023) reported a mean score of 840 on an easier 10-task benchmark that largely overlaps with ours, trained via offline supervised learning from expert data.

#### 5.1.2 DEEPMIND CONTROL SUITE

**Setup.** To evaluate ScaleZero's effectiveness to continuous control, we conduct experiments on 18 tasks from the DeepMind Control Suite. These tasks, based on the MuJoCo engine (Todorov et al., 2012), are characterized by low-dimensional state inputs and continuous action spaces, demanding precise dynamics modeling. Given the vector-based nature of observations, we use separate MLP encoders for each task. The experimental protocol remains consistent, comparing one MT ScaleZero agent against 18 individually trained single-task UniZero models. Details are provided in Appendix C.

**Results.** Table 1b demonstrate that ScaleZero achieves competitive performance, surpassing the baseline on several tasks and achieving a superior median score. This metric is particularly informative for evaluating a generalist agent, as it shows robust performance across the majority of tasks rather than excellence limited to a simpler subset. This success validates the versatility of the ScaleZero, proving its effectiveness not only in visual domains but also in complex continuous control.

### 5.1.3    JERICHO: TEXT-BASED ADVENTURE BENCHMARK

**Setup.** To assess performance on tasks requiring language understanding and long-horizon planning, we employ the Jericho benchmark. These text-based games present challenges through their combinatorial action spaces and the sparse reward nature. We evaluate on four representative games and compare against UniZero and CALM+OC (Sudhakar et al., 2023), a powerful language-model-based method. Both ScaleZero and UniZero use the same BGE text encoder (Xiao et al., 2023).

**Results.** As reported in Table 2, ScaleZero achieves performance on par with specialized single-task agents and is competitive with the strong CALM+OC baseline. This shows that the ScaleZero agent can effectively learn capable policies in environments where both states and actions are linguistic. These findings substantiate that the design of ScaleZero are modality-agnostic, successfully extending its efficacy to the domain of text-based planning. Experimental details are provided in Appendix D.

**Table 2:** Average returns comparison on four Jericho tasks. Detailed learning curves are in Appendix D.

| Algorithm | Acorncourt | Omniquest | Detective | Zork1 |
|---|---|---|---|---|
| CALM+OC (ST) (Sudhakar et al., 2023) | – | 7.8 | 288.5 | 38.0 |
| UniZero (ST) | 10 | 10 | **295** | **44** |
| ScaleZero (MT) | **10** | **10** | 280 | **44** |

### 5.2    EFFICIENCY EVALUATION OF DYNAMIC PARAMETER SCALING

Having established the superior multitask performance of ScaleZero, we now evaluate the efficiency gains afforded by our *Dynamic Parameter Scaling* strategy. We compare the standard ScaleZero against its *ScaleZero-DPS* variant on the DMC18 benchmark. The primary metric is the number of environment interactions required to match the strong performance of the single-task baselines. As shown in Figure 4, ScaleZero-DPS consistently achieves this performance target using only *71.5%* of the interactions required by ScaleZero. This corresponds to a substantial 28.5% reduction in data sampling and subsequent training cost, directly validating the effectiveness in resolving the static resource allocation problem. The steeper learning curves shown in Figure 10 further confirm that these efficiency gains persist throughout the dynamic training process. A mechanistic analysis in Appendix C.4 reveals the agent's efficiency stems from an emergent hierarchical strategy: the model learns to preserve foundational knowledge in early layers while applying targeted plasticity in later layers to adapt to unsolved tasks, providing a clear mechanistic explanation for its performance.

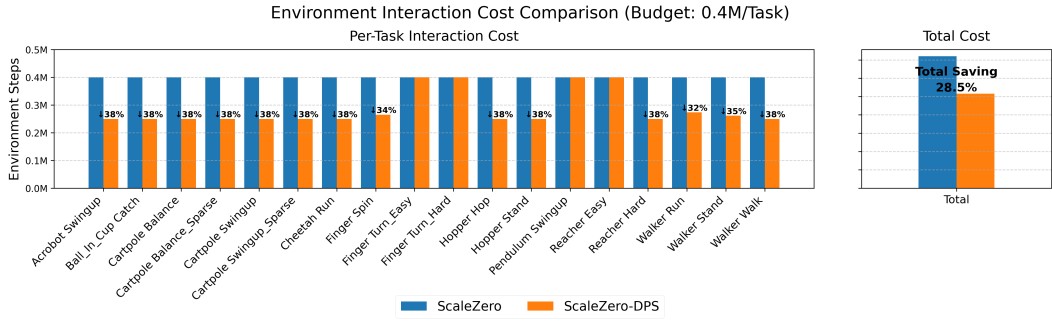

**Figure 4:** Interaction cost comparison for ScaleZero vs. ScaleZero-DPS on DMControl. Tha latter reaches the target performance with a 28.5% reduction in the environment cost. Detailed curves are in Appendix C.

## 5.3 Understanding MoE Efficacy: An Empirical and Theoretical Analysis

The superior performance of ScaleZero motivates a deeper investigation into its core architecture: the MoE transformer backbone. Here, we analyze the mechanisms of MoE from both empirical and theoretical perspectives. Our empirical findings (detailed in Appendix E.1) reveal that replacing dense MLP layers with MoE counterparts not only reduces gradient conflict within the MoE layers themselves but also in adjacent networks. To explain these observations, we provide a theoretical justification grounded in the two-phase training characterization of MoE (Chen et al., 2022). We introduce the following informal theorem, which asserts that an MoE layer maintains a strictly lower upper bound on gradient conflict than a dense counterpart—a property governed primarily by the router's specialization. Detailed explanations, proofs and corollaries are provided in Appendix E.2.

**Theorem 5.1** (Upper Bound on Gradient Conflict in MoE Layers *(informal)*). *In a Mixture-of-Experts (MoE) layer, consider two tasks $t_1$ and $t_2$ with routing weights $\lambda_m^{t_1}, \lambda_m^{t_2}$ over $M$ experts, and per-task gradients on expert $m$, denoted as $g_{t_1}^{(m)}$ and $g_{t_2}^{(m)}$. Let $G$ be the maximum gradient conflict on any single expert, and $\cos(u, v)$ denote the cosine similarity between vectors $u$ and $v$. Then the following results hold: (1). (Generally) Define the $m$-th element of vectors $u, v$ as $u_m = \lambda_m^{t_1} \|g_{t_1}^{(m)}\|, \; v_m = \lambda_m^{t_2} \|g_{t_2}^{(m)}\|, \; m = 1, \ldots, M$. The full-layer gradient conflict admits a general upper bound:* $\mathrm{conflict}(G_{t_1}, G_{t_2}) \leq G \cdot \cos(u, v)$. *(2). (Sparse Router) In the exploration stage, where tasks uniformly select experts, the expected gradient conflict of the MoE layer is approximately bounded by $G \cdot (1 - M!/((M - K)! \, M^K))$, where $K$ is the number of tasks. (3). (Sparse Router) In the router learning stage, where expert sets stabilize, the gradient conflict is roughly upper bounded by $\mathbb{E}[\mathrm{conflict}_{i,j}] \approx G \cdot \frac{U}{ab}$, where $a = |S_i|, b = |S_j|$ are the sizes of the expert sets for tasks $i$ and $j$, and $U = |S_i \cap S_j|$ is their intersection size.*

## 6 Conclusion and Future work

In this work, we diagnose and addressed the critical issue of plasticity collapse in multitask world models. We introduced *ScaleZero*, a unified architecture incorporating a sparse MoE backbone to create specialized computational pathways. Our experiments across 48 distinct tasks (Atari, DMControl, Jericho) demonstrate that ScaleZero robustly matches or surpasses the performance of specialized single-task agents. Furthermore, we propose DPS, a LoRA-based strategy that achieves competitive performance on DMControl while using just 71.5% of the environment interactions, highlighting a promising path toward greater sample efficiency. Our work provides a robust architectural foundation for developing more capable and efficient generalist agents. See Appendix G for a full discussion.

## 7 Acknowledgements

We extend our gratitude to several team-members of the Shanghai AI Laboratory for their invaluable assistance, support, and feedback on this paper and the associated codebase.

ETHICS STATEMENT

We have read and adhered to the ICLR Code of Ethics. Our research focuses on the development of novel algorithms for multitask reinforcement learning, specifically proposing *ScaleZero* and the Dynamic Parameter Scaling (DPS) strategy. All experiments were conducted on standard, publicly available benchmarks (Atari, DMC, and Jericho), which are simulated environments. Our work does not involve human subjects, nor does it use any personally identifiable or sensitive data. The proposed methods are foundational and do not have immediate, direct real-world applications that could pose ethical risks or societal harm. To the best of our knowledge, our research does not create or exacerbate discrimination or bias, as the training environments do not contain demographic or social data. We have strived to uphold the principles of scientific integrity and transparency by providing a thorough description of our methods and experimental setup. We declare no conflicts of interest.

REPRODUCIBILITY STATEMENT

To ensure the reproducibility of our findings, we provide comprehensive details in the appendix of this paper. The appendix includes: (1) A detailed description of the implementation of our proposed model, *ScaleZero*, and the online Dynamic Parameter Scaling (DPS) strategy. (2) A thorough account of the experimental setup, including all hyperparameters, evaluation protocols, and the computational infrastructure used. (3) The complete theoretical proofs for the claims made in the paper. Furthermore, we commit to releasing our source code publicly upon the completion of the peer-review process to allow for full verification of our results and to facilitate future research in this area.

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

APPENDIX CONTENTS

# A  IMPLEMENTATION DETAILS

This section provides a comprehensive overview of the implementation details for ScaleZero and the associated experimental methodology. For clarity, we first establish a baseline by providing a concise summary of the UniZero baseline (Pu et al., 2024), including its training and MCTS procedures. Building upon this foundation, we then sequentially detail the core modifications that define ScaleZero, our explorations of key design spaces, the Dynamic Parameter Scaling (DPS) strategy, and finally, the full hyperparameter and computational setup to ensure reproducibility. Our code is available at https://github.com/opendilab/LightZero.

## A.1  UNIZERO

As outlined above, we begin by detailing the foundational framework of our baseline, UniZero, to provide the necessary context for our contributions. The learning process revolves around two core procedures operating in a loop:

1. **`collect_experience`:** The agent interacts with the environment to gather trajectories. At each timestep, it uses Monte Carlo Tree Search within its learned latent world model to generate an improved policy, from which an action is sampled. The resulting experience tuples (observation, action, reward, done) and the MCTS-improved policy are stored in a replay buffer.

2. **`update_world_model`:** The world model, policy network, and value network are jointly optimized using batches of data sampled from the replay buffer. This step updates the model's understanding of the environment's dynamics and improves its decision-making capabilities.

### A.1.1  MCTS IN THE LEARNED LATENT SPACE

The MCTS procedure is central to UniZero's planning capability and is used during the `collect_experience` phase. For each action selection, a search tree is built in the latent space over a fixed number of simulations. Each simulation consists of three phases:

- **Selection:** Starting from the root node (the latent state of the current observation), the search traverses the tree by selecting actions that maximize an upper confidence bound score. This score, calculated using the PUCT formula, balances exploiting high-value actions (Q-value) and exploring less-visited actions (policy prior P). The action $a^*$ is chosen according to:

$$a^* = \arg\max_a \left[ Q(\hat{z}, a) + P(\hat{z}, a) \frac{\sqrt{\sum_b N(\hat{z}, b)}}{1 + N(\hat{z}, a)} \left( c_1 + \log \left( \frac{\sum_b N(\hat{z}, b) + c_2 + 1}{c_2} \right) \right) \right] \quad (2)$$

  where $N$ is the visit count, $Q$ is the mean action-value, $P$ is the policy prior, and $c_1, c_2$ are exploration constants.

- **Expansion:** When a leaf node is reached, the dynamics network is called to predict the next latent state $\hat{z}^{l+1}$ and reward $\hat{r}^l$. The decision network predicts a policy $p^l$ and value $v^l$ for the new state. This new node is added to the tree, and its outgoing edges are initialized with the predicted policy priors.

- **Backup:** The value $v^l$ predicted at the new leaf node is used to update the statistics of the nodes along the simulation path. The Q-values and visit counts $N$ for all parent state-action pairs are updated recursively back to the root.

After all simulations are complete, the visit counts at the root node are normalized to form a temperature-controlled distribution, which serves as the improved policy for action selection.

### A.1.2  MODEL AND POLICY OPTIMIZATION

The `update_world_model` step optimizes the network parameters by minimizing a joint loss function over batches of trajectories sampled from the replay buffer. The optimization objective in

UniZero is composed of four main prediction losses:

$$\mathcal{L}_{\text{UniZero}}(\theta) \doteq \mathop{\mathbb{E}}_{(\dots)\sim\mathcal{B}} \Big[ \sum_{t=0}^{H-1} \Big( \beta_z \underbrace{\|\hat{z}_{t+1} - \text{sg}(\bar{h}(o_{t+1}))\|_2^2}_{\text{next-latent prediction}} + \beta_r \underbrace{\text{CE}(\hat{r}_t, r_t)}_{\text{reward prediction}} \\ + \beta_p \underbrace{\text{CE}(p_t, \pi_t)}_{\text{policy prediction}} + \beta_v \underbrace{\text{CE}(v_t, \hat{v}_t^{targ})}_{\text{value prediction}} \Big) \Big]$$

(3)

where $H$ is the training context length and the loss components are defined as follows:

- **Next-latent Prediction:** The model predicts the next latent state $\hat{z}_{t+1}$. The target is generated by a slowly-updating target encoder $\bar{h}$ and is detached from the computation graph using a stop-gradient operator (sg).

- **Reward and Value Prediction:** The model predicts the immediate reward $\hat{r}_t$ and the state value $v_t$. To handle varying scales across tasks, both predictions are framed as discrete classification problems and optimized using a cross-entropy (CE) loss. The value target $\hat{v}_t^{targ}$ is a bootstrapped $n$-step TD target computed using future rewards and a target value network.

- **Policy Prediction:** The model's policy head $p_t$ is trained to mimic the MCTS-improved policy $\pi_t$. This serves as a policy distillation step, which enhances training stability compared to direct policy gradient methods.

The terms $\beta_z, \beta_r, \beta_p, \beta_v$ are fixed coefficients that balance the contribution of each component.

## A.2    CORE MODIFICATIONS OF SCALEZERO OVER UNIZERO

Our core model, ScaleZero, is built upon the UniZero baseline, with key components upgraded to enhance learning efficiency and final performance in complex multitask environments. The primary architectural differences are summarized in Table 3, with detailed explanations provided in the subsequent sections.

**Table 3:** Summary of core architectural differences between UniZero and ScaleZero.

| Component | UniZero (Baseline) | ScaleZero (Our Model) |
|---|---|---|
| **Backbone** | Standard Transformer blocks. | Mixture-of-Experts (MoE) Transformer blocks. |
| **Latent State Normalization** | SimNorm (L1 regularization). | Standard LayerNorm. |
| **Visual Encoder** | ResNet-like architecture trained from scratch. | Vision Transformer (ViT) trained from scratch. |

### A.2.1    BACKBONE: FROM DENSE TRANSFORMERS TO SPARSE MoE

The most significant modification lies in the model's backbone. While both models employ a Transformer architecture to process state, action, and task tokens, their internal structure differs fundamentally.

- **UniZero (Baseline):** The backbone consists of $N$ standard Transformer blocks (e.g., $N = 8$ for Atari26).

- **ScaleZero (Our Model):** To maintain a comparable parameter count while increasing model capacity, the backbone is composed of $N/2$ (for Atari) MoE Transformer blocks. In each block, the standard feed-forward network (FFN) is replaced by an MoE layer. This layer comprises 8 specialized expert networks and one shared expert, governed by a sparse top-1 routing gate. This design creates conditional computation pathways, mitigating task interference.

### A.2.2 ENCODERS AND NORMALIZATION

**Encoders** For encoding observations, ScaleZero adopts a similar strategy to the baseline but with a clear decision justified by empirical results.

- **Visual Encoder (e.g., Atari):** Our primary encoder is a Vision Transformer (ViT) trained from scratch. The input image (64x64x3) is divided into 8x8 patches, projected into token embeddings, and processed by a standard ViT-Base encoder to produce a 768-dimensional latent state.
- **Text Encoder (e.g., Jericho):** For text-based environments, we use a pre-trained BGE (BAAI General Embedding) model (Xiao et al., 2023) to encode textual observations into fixed-dimensional vectors.

**Justification for From-Scratch Visual Encoder** In preliminary experiments, we explored leveraging a powerful pre-trained visual encoder, specifically a frozen DINOv2 model (`dinov2_vits14`) (Oquab et al., 2023), to potentially improve feature quality. However, this approach did not yield significant or consistent performance gains over the from-scratch ViT. We hypothesize this is due to the domain gap between the natural images DINOv2 was trained on and the distinct visual characteristics of Atari frames. This finding justifies our final choice and highlights the importance of domain-specific visual representations. We leave a more in-depth investigation of pre-trained models, including fine-tuning strategies, as future work.

**Latent State Normalization** ScaleZero replaces the L1-based *SimNorm* used in UniZero with standard *LayerNorm* for normalizing the latent state. This change was found to be effective within our modified architecture.

### A.2.3 TASK-SPECIFIC ARCHITECTURAL CONFIGURATIONS

It is important to note that certain architectural choices are adapted based on the benchmark, rather than being a modification of ScaleZero over UniZero. These configurations apply to our entire framework.

- **Encoders:** For the DeepMind Control (DMC) suite, which features heterogeneous vector observation spaces, we employ independent MLP encoders for each task. For benchmarks with a unified input modality (images for Atari, text for Jericho), a single shared encoder is used across all tasks.
- **Prediction Heads:** To accommodate the distinct action spaces of each task, independent prediction heads (for policy, value, etc.) are universally employed across all benchmarks.

### A.3 EXPLORATIONS ON KEY DESIGN SPACES

We conducted several exploratory extensions to key designs of UniZero to validate their effectiveness in multitask settings.

**Task Information Encoding:** To mitigate representational conflicts in multitask learning, we explored explicit encoding of task information. By introducing a learnable task embedding matrix, `task_embed = nn.Embedding(task_id)`, and concatenating it with the state representation, `new_latent_state = concat(latent_state, task_embed)`, we aimed for the model to better distinguish between different tasks. In our experiments, the `latent_state` dimension was 672, and the `task_embed` dimension was 96.

**Vision Transformer (ViT):** Our ViT implementation is adapted from the `lucidrains/vit-pytorch`[1] library. The core hyperparameters, corresponding to a ViT-Base level, are summarized in Table 4.

**Latent Normalization Strategy (SimNorm):** As part of our ablation studies, we explored the SimNorm strategy, inspired by the TD-MPC2 paper. SimNorm partitions the latent state $z$ into $L$ groups, where each group is a $V$-dimensional simplex $g$. The transformation is given by:

$$z_{\text{sim\_norm}} = [g_1, \ldots, g_i, \ldots, g_L], \quad g_i = \frac{\exp(z_{i:i+V}/\tau)}{\sum_{j=1}^{V} \exp(z_{i:i+V_j}/\tau)} \tag{4}$$

---

[1] https://github.com/lucidrains/vit-pytorch

**Table 4:** Hyperparameters for the Vision Transformer (ViT) encoder.

| Hyperparameter | Value |
|---|---|
| image_size | 64x64 |
| patch_size | 8x8 |
| dim (Embedding Dimension) | 768 |
| depth (Transformer Layers) | 6 |
| heads (Attention Heads) | 6 |
| mlp_dim (MLP Hidden Dimension) | 2048 |
| dropout | 0.1 |
| emb_dropout (Embedding Dropout) | 0.1 |

In our experiments, we set the simplex dimension $V = 8$.

**Mixture-of-Experts (MoE):** Our MoE layer implementation leverages the efficient design patterns from the `mistral-inference`[2] library. Specifically, we adopt a "1 Shared + 8 Routed" configuration where every token is processed by a dedicated shared expert while a gating network simultaneously routes it to the Top-1 expert among eight specialized experts. We do not claim this configuration is the theoretical global optimum, but rather a pragmatic trade-off between computational efficiency and performance. This design aligns with advanced LLM architectures (e.g., DeepSeek-MoE Dai et al. (2024)) that explicitly separate shared knowledge from specialized knowledge. As a Proof of Concept, it demonstrates that introducing even moderate sparsity (Routing) significantly mitigates the interference observed in dense baselines. In the future, drawing inspiration from Ludziejewski et al. (2025) and Zhao et al. (2025), we plan to investigate the optimal MoE configurations specifically for online MTRL.

**Gradient Correction for MTRL.**

Conflicting gradients are a primary challenge in MTRL, often impeding stable optimization. To mitigate this, we adopt the MoCo algorithm from the LibMTL library (Fernando et al., 2023)[3]. MoCo is a gradient correction method that finds a consensual update direction by dynamically adjusting task weights. This process operates on momentum-smoothed gradients, which provide a more stable estimate of each task's long-term descent direction compared to noisy single-step gradients. The procedure involves two main steps:

- **Momentum-based Gradient Smoothing.** To obtain a stable estimate of each task's descent direction, we maintain a momentum-based moving average, $y_i(t)$, of its past gradients. For each task $i$ at step $t$, the update rule is:

$$y_i(t) = (1 - \beta)y_i(t-1) + \beta g_i'(t) \tag{5}$$

  where $g_i'(t)$ is the raw gradient $g_i(t)$ normalized and scaled by its task loss, and $\beta$ is the momentum coefficient. This smooths out noisy single-step gradients.

- **Dynamic Weight Adjustment.** The task weights $\lambda(t) \in \mathbb{R}^N$ are updated to minimize the squared norm of the aggregated gradient, formulated as $\min_\lambda \frac{1}{2}\|\sum_{i=1}^N \lambda_i y_i(t)\|^2$. We perform an approximate gradient descent step on this objective. The correct weight update is:

$$\lambda(t) = \text{softmax}\left(\lambda(t-1) - \gamma(Y(t)^T Y(t))\lambda(t-1)\right) \tag{6}$$

  where $Y(t)$ is the matrix whose columns are the momentum gradients $\{y_i(t)\}$, and $\gamma$ is the weight learning rate. The final corrected gradient applied to the shared parameters $\theta_s$ is the weighted sum $g_s(t) = \sum_{i=1}^N \lambda_i(t)y_i(t)$.

- Our implementation is tailored for a Distributed Data Parallel (DDP) environment. All gradient correction computations occur on the master process (rank 0), which then broadcasts the final corrected gradient to all other processes for synchronized updates. The procedure is detailed in Algorithm 1. Key hyperparameters are listed in Table 5.

---

[2]https://github.com/mistralai/mistral-inference/blob/main/src/mistral_inference/moe.py

[3]https://github.com/median-research-group/LibMTL

---

**Algorithm 1** MoCo in a Distributed Data Parallel (DDP) Setting

---

**Require:** Shared parameters $\theta_s$, task losses $\{L_i\}_{i=1}^N$.
**Require:** Momentum gradients $\{y_i\}_{i=1}^N$, task weights $\lambda \in \mathbb{R}^N$.
**Require:** Hyperparameters: Momentum $\beta$, weight learning rate $\gamma$.
**Ensure:** Updated shared parameters $\theta_s$.
 1: **for** each training step **do**
       *# On each process in parallel*
 2:    Compute local raw gradients $\{g_i^{\text{local}}\}$ and losses $\{L_i^{\text{local}}\}$.
 3:    Aggregate all $\{g_i^{\text{local}}\}$ and $\{L_i^{\text{local}}\}$ to the master process (rank 0).

       *# On the master process (rank 0) only*
 4:    Let $\{g_i\}_{i=1}^N$ and $\{L_i\}_{i=1}^N$ be the aggregated global gradients and losses.
 5:    **for** $i = 1$ **to** $N$ **do**
 6:        Normalize and scale: $g_i' \leftarrow \frac{g_i}{\|g_i\|+\epsilon} \cdot L_i$.
 7:        Update momentum gradient: $y_i \leftarrow (1 - \beta)y_i + \beta g_i'$.
 8:    **end for**
 9:    Construct gradient matrix: $Y \leftarrow [y_1, \ldots, y_N]$.
10:    Update task weights: $\lambda \leftarrow \text{softmax}(\lambda - \gamma(Y^T Y)\lambda)$.
11:    Compute final corrected gradient: $g_s \leftarrow \sum_{i=1}^N \lambda_i y_i$.
12:    Broadcast the corrected gradient $g_s$ to all other processes.

       *# On each process in parallel*
13:    Apply corrected gradient: $\nabla_{\theta_s} \leftarrow g_s$.
14:    Perform optimizer step to update $\theta_s$.
15: **end for**

---

**Table 5:** Hyperparameters for the MoCo algorithm.

| Hyperparameter | Symbol | Value |
|---|---|---|
| Momentum Coefficient | $\beta$ | 0.99 |
| Weight Learning Rate | $\gamma$ | 10.0 |
| Weight Regularization | $\rho$ | 0.0 |

## A.4 DYNAMIC PARAMETER SCALING (DPS)

This section provides a detailed breakdown of the *Dynamic Parameter Scaling (DPS)* strategy introduced in Section 4.3. DPS is designed to dynamically couple model capacity with learning progress, creating a "curriculum of model complexity" that directs resources precisely where they are most required. Detailed pseudocode is provided in Algorithm 2.

**Task Curation and Active Set Management.** As described in the main text, we define the full set of tasks as $\mathcal{T} = \{\tau_i\}_{i=1}^N$. A task $\tau_i$ is deemed "solved" once its performance metric, $\text{Metric}(\tau_i)$, surpasses a predefined threshold $\varepsilon_i$. At any time $t$ during training, we maintain an *active set* of all unsolved tasks, $\mathcal{U}_t \subseteq \mathcal{T}$. To maximize computational efficiency, both data collection and gradient updates are performed **exclusively** for tasks within this active set $\mathcal{U}_t$. Once a task is solved, it is removed from the active set, thereby ceasing all computational overhead associated with it.

**Staged Capacity Expansion.** The training process is partitioned into $S + 1$ stages, following the protocol outlined in the main text:

- **Stage 0 (Warm-up):** During an initial period of $T_0$ iterations, only the shared backbone network parameters, $\theta_B$, are trained. This stage establishes a robust foundation by training the shared base model on all tasks to learn general-purpose representations.

- **Stage $s \geq 1$ (Expansion):** Each subsequent stage introduces a new, independent LoRA module, $\Delta\theta_s = B_s A_s$. Concurrently, a set of learnable scaling factors is defined to modulate the contribution of both the base model and all adapters. A scaling factor $\alpha_0$ is associated with the base model weights $\theta_B$, and each LoRA module $\Delta\theta_j$ is assigned a corresponding scaling factor $\alpha_j$. All scaling

---

**Algorithm 2** Dynamic Parameter Scaling (DPS)

---

**Require:** Set of tasks $\mathcal{T} = \{\tau_i\}_{i=1}^N$; Performance thresholds $\{\varepsilon_i\}_{i=1}^N$; Backbone params $\theta_B$;
**Require:** Hyperparameters: Total stages $S$, Warm-up iterations $T_0$, Max total iterations $T_{\max}$, Stage quotas $\{Q_s\}_{s=1}^S$.
**Ensure:** Trained parameters: $\theta_B$, $\alpha_0$, $\{\Delta\theta_s, \alpha_s\}_{s=1}^S$.
 1: Initialize active set of unsolved tasks: $\mathcal{U} \leftarrow \mathcal{T}$.
 2: Initialize base model scaling factor: $\alpha_0 \leftarrow 1$.
 3: **for** stage $s = 0$ **to** $S$ **do**
 4:  **if** $s == 0$ **then**                                          ▷ **Stage 0: Warm-up Phase**
 5:   Set trainable parameters $\theta_{\text{train}} \leftarrow \theta_B$.
 6:   Set stage iteration limit $L_s \leftarrow T_0$.
 7:   **for** $t_{\text{stage}} = 1$ **to** $L_s$ **do**
 8:    Sample a batch exclusively from tasks in active set $\mathcal{U}$.
 9:    Perform forward pass with backbone: $W(x) = W_0 x$.
10:    Compute loss $\mathcal{L}$ and update $\theta_{\text{train}}$.
11:    Update active set: $\mathcal{U} \leftarrow \{\tau_i \in \mathcal{T} \mid \text{Metric}(\tau_i) < \varepsilon_i\}$.
12:   **end for**
13:  **else**                                          ▷ **Stage $s \geq 1$: Expansion Phase**
14:   Initialize new LoRA module $\Delta\theta_s = B_s A_s$ and its scalar $\alpha_s \leftarrow 1$.
15:   Freeze $\theta_B$ and all previous modules $\{\Delta\theta_j\}_{j=1}^{s-1}$.
16:   Set trainable parameters $\theta_{\text{train}} \leftarrow \{\Delta\theta_s\} \cup \{\alpha_0\} \cup \{\alpha_j\}_{j=1}^{s-1}$.
17:   Record solved tasks at stage start: $\mathcal{S}_{\text{start}} \leftarrow \mathcal{T} \setminus \mathcal{U}$.
18:   Set budget for current stage: $T_{\text{budget}} \leftarrow \lceil (T_{\max} - T_0)/S \rceil$.
19:   $t_{\text{stage}} \leftarrow 0$.
20:   **while** $t_{\text{stage}} < T_{\text{budget}}$ **do**                                          ▷ **Budget-based Trigger**
21:    $t_{\text{stage}} \leftarrow t_{\text{stage}} + 1$.
22:    Sample a batch exclusively from tasks in active set $\mathcal{U}$.
23:    Perform forward pass: $W^{(s)}(x) = (\alpha_0 W_0 + \sum_{j=1}^s \alpha_j (B_j A_j))x$.
24:    Compute loss $\mathcal{L}$ and update $\theta_{\text{train}}$.
25:    Update active set: $\mathcal{U} \leftarrow \{\tau_i \in \mathcal{T} \mid \text{Metric}(\tau_i) < \varepsilon_i\}$.
26:    **if** $|\mathcal{T} \setminus \mathcal{U}| - |\mathcal{S}_{\text{start}}| \geq Q_s$ **then**                                          ▷ **Progress-based Trigger**
27:     **break**       ▷ Quota of newly solved tasks met, transition to next stage.
28:    **end if**
29:   **end while**
30:  **end if**
31: **end for**

---

factors are initialized to 1. These factors allow the model to dynamically re-weigh and reuse knowledge from the base model and earlier adaptations.

**Adaptive Stage Transition Triggers.** The transition from stage $s - 1$ to stage $s$ is governed by a dual-trigger mechanism, balancing learning progress against a fixed budget:

1. **Progress-based Trigger:** A new stage is initiated if the number of newly solved tasks within the current stage reaches a predefined quota $Q_s$.

2. **Budget-based Trigger:** To prevent stagnation on particularly arduous tasks, a transition is forced if the iteration count within the current stage exceeds a pre-allocated limit, calculated as $\lceil (T_{\max} - T_0)/S \rceil$.

**Optimization and Parameter Isolation.** A key principle of DPS is parameter isolation. At any given stage $s \geq 1$, the effective weight matrix $W^{(s)}$ is dynamically composed from the base weight $W_0 \in \theta_B$ and all accumulated adapters:

$$W^{(s)} = \alpha_0 W_0 + \sum_{j=1}^s \alpha_j \Delta\theta_j = \alpha_0 W_0 + \sum_{j=1}^s \alpha_j B_j A_j. \tag{7}$$

Upon entering a new stage $s$, all prior parameters—the backbone $\theta_B$ (pre-trained in Stage 0) and the matrices of all previously introduced adapters $\{\Delta\theta_j\}_{j=1}^{s-1}$—are *frozen*. To resolve scale ambiguity

between the new adapter and its scalar, the scaling factor $\alpha_s$ for the currently active adapter $\Delta\theta_s$ is temporarily fixed at 1. Optimization is thereby focused *exclusively on* the newly added LoRA module $\Delta\theta_s$, the base model's scaling factor $\alpha_0$, and the set of scaling factors corresponding to previously frozen adapters, $\{\alpha_j\}_{j=1}^{s-1}$.

This strategy yields two principal advantages:

- It implements a *resource-efficient training curriculum*, as new parameters are only introduced when required by the remaining unsolved tasks.

- By isolating new learning within dedicated adapters, DPS provides *targeted plasticity* for difficult tasks while *preventing catastrophic forgetting or negative transfer* to previously mastered skills, whose knowledge is preserved in the frozen backbone and prior adapters.

**Stabilized Learnable Scaling Factors.** To prevent training instability from unbounded learnable scalars, we constrain the scaling factors $\alpha_j$ within a narrow, stable range (e.g., centered at 1.0). This is achieved via a re-parameterization trick, where each $\alpha_j$ is computed from an unbounded underlying parameter $\hat{\alpha}_j$ as follows:

$$\alpha_j = \text{offset} + \text{range} \cdot \tanh(\hat{\alpha}_j) \tag{8}$$

This design ensures that the contribution of each adapter is modulated smoothly without causing drastic shifts in the output distribution. The specific DPS hyperparameters used in our ScaleZero-DPS experiments are detailed in Table 6.

**Table 6:** Hyperparameter configuration for Dynamic Parameter Scaling in our ScaleZero-DPS experiments.

| Hyperparameter | Symbol | Value | Description |
|---|---|---|---|
| curriculum_stage_num | $S+1$ | 5 | Total number of stages (1 warm-up + 4 expansion). |
| lora_r | $r$ | 64 | The rank of the LoRA matrices. |
| lora_alpha | - | 1 | Conventional LoRA hyperparameter (scaling = alpha/r). *Note: This is distinct from our learnable $\alpha_j$ factors.* |
| lora_scale_init | $\alpha_{j,\text{init}}$ | 1.0 | Initial value of the DPS learnable scaling factors, $\alpha_j$. |
| lora_scale_range | - | 0.2 | Range for DPS scalars, yielding $\alpha_j \in [0.8, 1.2]$. |
| min_stage0_iters | $T_0$ | 10,000 | Number of iterations for the warm-up stage. |
| max_stage_iters | - | 5,000 | Per-stage iteration budget, i.e., $\lceil (T_{\max} - T_0)/S \rceil$. |

## A.5 HYPERPARAMETER SETTINGS

Our hyperparameter configuration for ScaleZero largely follows the original UniZero paper to ensure a fair comparison. We employ the `AdamW` optimizer for all experiments. Table 7 lists the hyperparameters that are kept consistent across all benchmarks. Architectural configurations and loss weights are detailed subsequently, followed by benchmark-specific settings in Tables 9, 10, and 11.

Our ScaleZero model incorporates Mixture-of-Experts layers within its Transformer backbone for the three benchmarks to enhance model capacity and specialization. When MoE is used, each MoE layer consists of 8 specialist experts and 1 shared expert, with the router selecting the top-1 expert per token. The final loss is a weighted sum of several components, with weights varying between discrete and continuous action space environments, as detailed in Table 8.

### A.5.1 ATARI-SPECIFIC SETTINGS

For Atari, observations are $64 \times 64 \times 3$ images processed by a ViT encoder. Training is conducted for 400,000 total environment steps.

### A.5.2 DMC-SPECIFIC SETTINGS

For the DMC suite, state vectors are processed by an MLP encoder. Training runs for 400,000 total environment steps.

**Table 7: Common Hyperparameters**. This table lists the hyperparameters that remain constant across all benchmarks (Atari, DMC, and Jericho tasks) for ScaleZero and UniZero.

| Hyperparameter | Value |
|---|---|
| **Planning** | |
| Number of MCTS Simulations ($sim$) | 50 |
| Temperature | 0.25 |
| Dirichlet Noise ($\alpha$) | 0.3 |
| Dirichlet Noise Weight | 0.25 |
| Coefficient $c_1$ | 1.25 |
| Coefficient $c_2$ | 19652 |
| **Architecture** | |
| Embedding Dimension | 768 |
| Number of Heads | 8 |
| Dropout Rate ($p$) | 0.1 |
| Activation Function | GELU |
| Reward/Value Bins | 101 for DMC and Jericho, 601 for Atari |
| SimNorm Dimension ($V$) | 8 |
| SimNorm Temperature ($\tau$) | 1 |
| **Optimization** | |
| Optimizer | AdamW |
| Learning Rate | $1 \times 10^{-4}$ |
| Policy Entropy Coefficient | $1 \times 10^{-4}$ |
| Weight Decay | $10^{-4}$ |
| Max Gradient Norm | 5 |
| Discount Factor | 0.997 |
| Soft Target Update Momentum | 0.05 |
| Temporal Difference (TD) Steps | 5 |

**Table 8:** Loss function weights for different environment types.

| Loss Term | Discrete Action Spaces (Atari, Jericho) | Continuous Action Spaces (DMC) |
|---|---|---|
| Value Loss | 0.5 | 0.1 |
| Reward Loss | 1.0 | 0.1 |
| Policy Loss | 1.0 | 0.1 |
| Observation Loss | 10.0 | 10.0 |

**Table 9:** Key hyperparameters for Atari experiments.

| Hyperparameter | Value |
|---|---|
| *General Training* | |
| Effective Global Batch Size | 512 |
| Discount Factor ($\gamma$) | 0.997 |
| *World Model* | |
| Num. Transformer Layers | 2 (Atari-8), 4 (Atari-26) |
| Training Context Length ($H$) | 10 |
| Encoder Type | ViT |
| *MCTS* | |
| Inference Context Length ($H_{\text{infer}}$) | 4 |
| *Replay Buffer* | |
| Replay Buffer Size | 500,000 |
| Replay Ratio | 0.25 |

**Table 10:** Key hyperparameters for DMC experiments.

| Hyperparameter | Value |
|---|---|
| *General Training* | |
| Effective Global Batch Size | 512 |
| Discount Factor ($\gamma$) | 0.99 |
| Frame Skip | 4 (8 for Pendulum) |
| *World Model* | |
| Num. Transformer Layers | 4 |
| Training Context Length ($H$) | 5 |
| Encoder Type | MLP |
| Num. Sampled Actions | 20 |
| *MCTS* | |
| Inference Context Length ($H_{\text{infer}}$) | 2 |
| *Replay Buffer* | |
| Replay Buffer Size | 1,000,000 |
| Replay Ratio | 0.25 |

### A.5.3 JERICHO-SPECIFIC SETTINGS

For the text-based Jericho suite, a pre-trained language model (`BAAI/bge-base-en-v1.5`) is used as the text encoder. Training runs for 500,000 total environment steps.

**Table 11:** Key hyperparameters for Jericho experiments.

| Hyperparameter | Value |
|---|---|
| *General Training* | |
| Effective Global Batch Size | 256 |
| Discount Factor ($\gamma$) | 0.997 |
| *World Model* | |
| Num. Transformer Layers | 2 |
| MoE Layers | None |
| Training Context Length ($H$) | 10 |
| Encoder Type | Text (`BAAI/bge-base-en-v1.5`) |
| *MCTS* | |
| Inference Context Length ($H_{\text{infer}}$) | 4 |
| *Replay Buffer* | |
| Replay Buffer Size | 500,000 |
| Replay Ratio | 0.1 |

### A.6 TRAINING FRAMEWORK AND COMPUTATIONAL ANALYSIS

This section details the distributed training framework employed for our multi-task experiments and provides a comprehensive analysis of the model parameter specifications and computational costs for the Atari, DMC, and Jericho benchmarks.

### A.6.1 DISTRIBUTED MULTI-TASK TRAINING IMPLEMENTATION

Our multi-GPU, multitask training framework is implemented based on PyTorch's Distributed Data Parallel (DDP) (Li et al., 2020). The design is optimized for scalability and efficiency when training a single unified model across highly diverse task sets. The core implementation logic is as follows:

- **Static Task Partitioning and Resource Allocation:** To handle the diverse set of environments (e.g., 26 distinct Atari games), we employ a static partitioning strategy. The total

set of tasks is divided among the available GPUs (ranks). Each GPU is assigned an exclusive subset of tasks and instantiates dedicated resources for each assigned task, including independent data collectors, evaluators, and replay buffers. This modular design isolates task-specific operations, ensuring robust memory management and preventing resource contention between tasks.

- **Heterogeneous Batch Construction:** During each training iteration, every GPU constructs a local training batch by sampling data segments from the replay buffers of its assigned tasks. This results in a heterogeneous batch containing experiences from multiple different environments. To ensure load balancing, the micro-batch size sampled from each task's buffer is uniform and pre-calculated based on global memory constraints.

- **Gradient Accumulation for Large-Scale Training:** Training transformer-based world models requires significant memory. To simulate a large effective batch size necessary for stable convergence without exceeding GPU memory limits, we utilize gradient accumulation. The number of accumulation steps is dynamically adjusted in conjunction with the micro-batch size. This strategy ensures that the model parameters are updated using gradients derived from a sufficiently large and diverse dataset, promoting effective generalization.

- **Synchronized Global Optimization:** Once the gradients are computed locally on the heterogeneous batches, the DDP framework synchronizes them across all GPUs via an `AllReduce` operation. This results in a globally averaged gradient that reflects the collective experience of all tasks in the benchmark. A single optimization step is then performed on the shared model parameters, ensuring that the unified model learns jointly from the entire multi-task distribution.

This integrated approach enables the joint optimization of a single, powerful model using data streams from multiple environments in a distributed and highly efficient manner, which is key to ScaleZero's strong performance in multitask scenarios.

## A.7 PARAMETER ANALYSIS AND COMPUTATIONAL COST

To provide a granular analysis of resource utilization, Table 12 compares the network parameter scales and specific training costs between our multi-task approach, ScaleZero (MT), the multi-task baseline UniZero (MT), and the single-task baseline UniZero (ST). Experiments for multi-task models were conducted on a computing node equipped with $8\times$ ($4\times$ for Jericho-4) NVIDIA A100 (80GB) GPUs, whereas single-task models were trained on individual $1\times$ NVIDIA A100 (80GB) GPUs per task. Table 12 summarizes the parameter counts and approximate wall-clock training times for the main results presented in section 5.

Note on Parameter Counts: The *Total Params* column encompasses all model components, including embeddings and auxiliary heads; thus, it exceeds the simple sum of the Encoder, Backbone, and Head parameters. For *UniZero (ST)*, the values listed represent the model size for a single task. The total training time for ST is the aggregate time required to train all tasks within the benchmark sequentially or in parallel resource equivalents.

## A.8 JUSTIFICATION FOR THE MODEL-FREE MULTI-TASK BASELINES

In the experimental comparisons presented in Table 1a, we focus primarily on Model-Based approaches, excluding online model-free MTRL baselines. This exclusion is driven by fundamental disparities in data regimes and architectural paradigms:

The Atari-26 benchmark presents significant challenges due to its heterogeneity in visual dynamics and mechanics. A survey of recent literature (Xu et al., 2022; Ye et al., 2022) indicates that existing multi-task methods predominantly rely on offline pre-training or fine-tuning rather than learning *ab initio*. While methods such as IMPALA with PopArt (Hessel et al., 2019) have shown promise, they operate in a *high-throughput regime*, typically requiring over 200 million frames to converge. In contrast, ScaleZero targets the *sample-efficient regime* (consistent with the MuZero/UniZero research trajectory), operating within a strict budget of 100k–300k environment steps—approximately 0.2% of the data required by Hessel et al. (2019). Under such severe constraints, high-throughput model-free RL methods fail to learn meaningful policies, rendering a direct comparison methodologically

**Table 12:** Detailed comparison of network parameters, wall-clock training time, and hardware resources. "nlayer" denotes the number of transformer layers in the world model backbone. For UniZero (ST), the training time indicates the cost per task and the total cumulative time for the benchmark. Colors indicate method type: **Blue** for ScaleZero (MT), **Gray** for UniZero (MT), and **Orange** for UniZero (ST).

| Benchmark | Method | Network Parameters | | | | Training Time | Hardware |
|---|---|---|---|---|---|---|---|
| | | Encoder | Backbone | Head | Total | | |
| **Atari-26** | ScaleZero (MT) (nlayer=4) | 26.8 M | 254.9 M (act.≈56M) | 81.5 M | 366.4 M | ∼6 Days | 8× A100 (80G) |
| | UniZero (MT) (nlayer=8) | 18.8 M | 56.7 M | 81.5 M | 160.2 M | ∼5 Days | 8× A100 (80G) |
| | UniZero (ST) (nlayer=2) | 7.8 M | 14.2 M | 3.9 M | 43.8 M | 7 Hours/Task Total: ∼8 Days | 1× A100 (80G) |
| **DMC-18** | ScaleZero (MT) (nlayer=4) | 10.8 M | 254.9 M (act.≈56M) | 66.9 M | 339.9 M | ∼2 Days | 8× A100 (80G) |
| | UniZero (ST) (nlayer=2) | 0.60 M | 14.2 M | 11.2 M | 26.1 M | 3 Hours/Task Total: ∼ 2.3 Days | 1× A100 (80G) |
| **Jericho-4** | ScaleZero (MT) (nlayer=2) | 110.1 M | 127.4 M (act.≈27.9M) | 14.2 M | 250.2 M | ∼3 Days | 4× A100 (80G) |
| | UniZero (ST) (nlayer=2) | 110.1 M | 14.2 M | 3.2 M | 127.5 M | 13 Hours/Task Total: ∼ 2.1 Days | 1× A100 (80G) |

unsound. To date, no prior work has reported a single online MTRL agent capable of mastering the full Atari-26 suite from scratch under this data budget.

Similarly, the current landscape for the DMC-18 benchmark is dominated by offline approaches or methods utilizing expert demonstrations (Schmied et al., 2023; Haldar et al., 2024). Comparing our online learning framework, which learns purely from interaction, against offline paradigms that benefit from extensive pre-collected datasets creates an inequitable experimental setting due to the vast disparity in data availability.

Existing Model-Free MTRL methods, such as PaCo (Sun et al., 2022) and Soft Modularization (Yang et al., 2020), introduce architectural innovations specifically tailored for Policy or Value networks. Conversely, ScaleZero operates within a Model-Based framework focused on learning environmental dynamics. The inductive biases required for accurate dynamics modeling differ fundamentally from those optimized for value estimation. Consequently, directly transplanting architectural mechanisms designed for policy optimization into a World Model structure lacks theoretical compatibility and straightforward implementation.

# B   ATARI EXPERIMENT DETAILS

## B.1   BENCHMARK SETUP

This section elaborates on the experimental setup for the Atari benchmarks discussed in Section 5.1. We use the Arcade Learning Environment (ALE) (Bellemare et al., 2013) as our primary simulation platform.

Our evaluation protocol compares our multitask model, *ScaleZero (MT)*, against a strong single-task baseline consisting of individually trained *UniZero (ST)* models. We adopt this comparative framework for two reasons: first, standard online MTRL baselines for the full Atari suite are currently lacking; second, the multitask variant of UniZero demonstrates severe performance degradation. Consequently, the UniZero ST baseline serves as a rigorous "specialist" upper bound, providing a challenging standard for evaluating the net positive knowledge transfer of our multitask approach.

The experimental evaluation is structured into two phases: first, we conduct model design and ablation studies on the *Atari8* (multitask) benchmark (a subset of 8 games: Alien, Boxing, ChopperCommand, Hero, MsPacman, Pong, RoadRunner, and Seaquest). Second, we evaluate the final ScaleZero architecture on the full *Atari26* (multitask) benchmark.

For all games, observations are preprocessed into 64x64 RGB images. The state input to the model is a single frame, yielding an input tensor of shape 3x64x64 without any frame stacking. We employ standard environment wrappers, including sticky actions (p=0.25) and a frame skip of 4 (Mnih et al., 2013). Performance is quantified by the Human-Normalized Score (HNS), with both mean and median scores reported to assess aggregate performance and robustness. To ensure reproducibility and

fair comparison, all environmental configurations strictly adhere to those in the UniZero paper (Pu et al., 2024) and DI-engine (Niu et al., 2021).

## B.2 Supplementary Analysis of Plasticity and Representational Collapse

This section provides a deeper empirical analysis of the plasticity loss and representational collapse phenomena diagnosed in Section 5.

### B.2.1 Training Dynamics and Representational Collapse in UniZero

While Figure 1 in the main text illustrates the performance degradation and plasticity loss for UniZero on complex tasks, Figure 6 presents the same metrics for the four other tasks in the Atari8 set.

To quantitatively diagnose the cause of this degradation, we measure the *effective rank* (Dohare et al., 2024) of the model's representation space. A low effective rank indicates representational collapse, where the model fails to maintain the feature diversity required for multitask learning. For a matrix $\mathbf{\Phi} \in \mathbb{R}^{n \times m}$ with normalized singular values $p_k$, the effective rank is the exponential of the Shannon entropy of its singular value distribution:

$$\text{erank}(\mathbf{\Phi}) \doteq \exp\left(-\sum_{k=1}^{q} p_k \log(p_k)\right) \tag{9}$$

Specifically, we compute the effective rank of a hidden layer's activation matrix from a minibatch of 256 sequence samples. As shown in Figure 5, the performance collapse in the baseline model is directly accompanied by a decline in the representation's effective rank, providing quantitative evidence for the representational collapse hypothesis.

### B.2.2 Analysis of Representational Collapse

The correlations observed in our experiments reveal a critical failure mode in standard multitask architectures. These models face inherent difficulties in reconciling conflicting task demands, leading to severe gradient conflict. We hypothesize that as gradients from distinct tasks interfere destructively, the magnitude of the aggregated update signal diminishes—a phenomenon we term *Gradient Norm Collapse*. This hypothesis is supported by the experimental analysis in Ma et al. (2025), which demonstrates that dense connectivity inherently induces such destructive interference, thereby impeding the model's ability to track non-stationary data distributions. Consequently, the feature space degenerates into a low-dimensional, task-agnostic subspace, which is suboptimal for specialized tasks. This is the mechanism of *representational collapse*, quantitatively diagnosed by the low effective rank shown in Figure 5.

This contraction of the feature space renders many neurons redundant, leading to a *rising dormant neuron ratio*. Specifically, when weights fail to adapt to shifting input patterns, neuronal pre-activations may systematically drift into negative regions. Once a neuron becomes inactive, its gradient vanishes, effectively freezing the associated weights and resulting in irreversible *Plasticity Loss* (Sokar et al., 2023). To compensate for this impoverished representational capacity, normalization layers (e.g., LayerNorm) amplify the magnitude of the few remaining active features, causing an *inflation of the latent norm*. Ultimately, this collapsed representation lacks the capacity to model task diversity, resulting in suboptimal performance.

In contrast, ScaleZero's MoE architecture is designed to mitigate this issue. By routing tasks to specialized expert sub-networks, it enforces dynamic sparsity. This design is consistent with the findings of Ma et al. (2025), which suggest that network sparsity is crucial for unlocking the scaling potential of deep reinforcement learning. By reducing the interference highlighted in their work, our MoE design theoretically lowers the upper bound of gradient conflict. This preserves distinct representational subspaces. As shown in Figure 7, the MoE architecture effectively maintains neuron activity (low dormant ratio) and stabilizes representation magnitudes (controlled latent norm), thereby preventing representational collapse and enabling superior multitask performance.

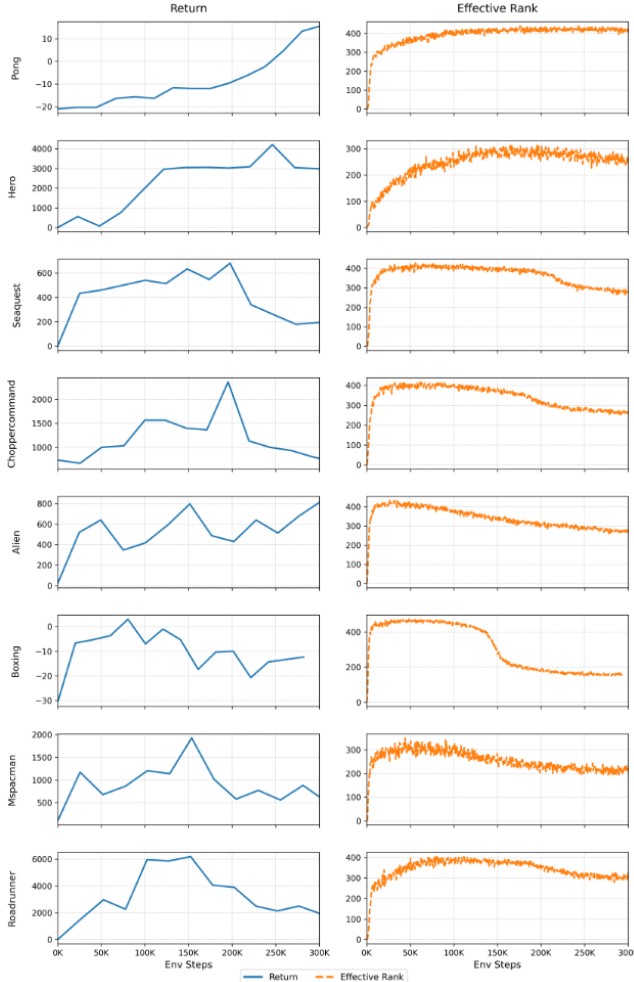

**Figure 5: Analysis of representation effective rank.** This figure supplements the diagnosis in Figure 1, showing the relationship between game return (Left) and representation effective rank (Right) for the baseline model. The sharp drop in performance correlates strongly with a decline in effective rank, substantiating the claim that performance failure is linked to a collapse in the dimensionality of the model's representation space.

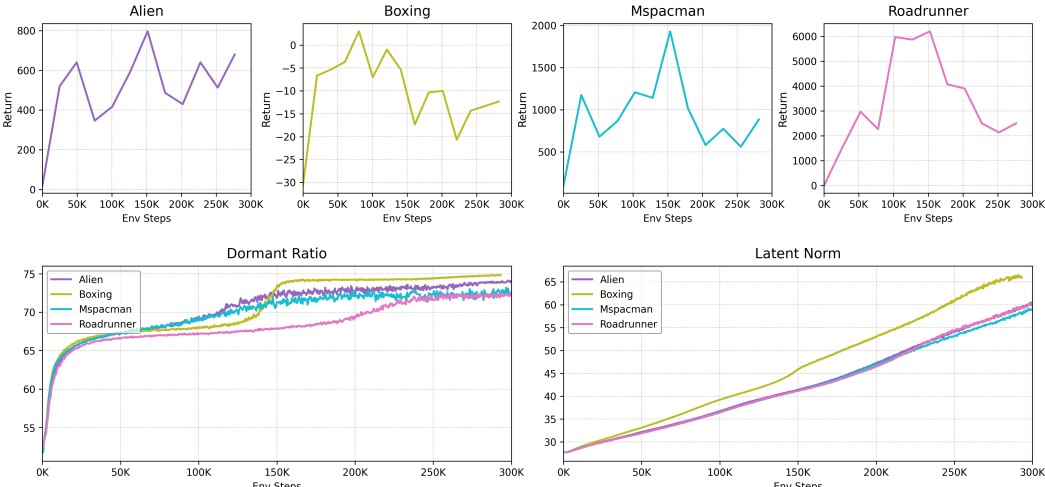

**Figure 6: Training dynamics of UniZero on the other four Atari8 tasks.** Complementing Figure 1 from the main text, this figure shows performance curves, dormant neuron ratios, and latent state norms for UniZero on Boxing, MsPacman, RoadRunner, and Alien.

### B.3 FULL PERFORMANCE ON THE ATARI26 BENCHMARK

This section provides the complete learning curves corresponding to the summary results in Table 1a. As reported in the main text, the single multitask ScaleZero (MT) agent achieves a higher mean HNS than the average of 26 specialized UniZero (ST) agents. This result demonstrates that ScaleZero achieves net positive knowledge transfer, which we attribute to the MoE architecture's ability to maintain plasticity and transfer general priors (e.g., object physics) from simple to complex tasks.

However, the median HNS of ScaleZero is lower than the ST baseline. This is primarily influenced by suboptimal performance on a few hard-exploration or mechanically unique games (e.g., *PrivateEye*), indicating that negative interference is not fully eliminated. The complete learning curves for all 26 games are shown in Figure 8.

## C DMC EXPERIMENT DETAILS

### C.1 BENCHMARK SETUP

This section details the experimental setup for the DeepMind Control (DMC) Suite (Tunyasuvunakool et al., 2020), which serves as our benchmark for continuous control as presented in Section 5.1. Experiments were conducted on a suite of 18 tasks (e.g., *walker_walk*, *cheetah_run*), following the experimental setup of Pu et al. (2024). Following the methodology of our Atari experiments, we compare a single *ScaleZero (MT)* model against a baseline of 18 individually trained *UniZero (ST)* models. Model inputs are low-dimensional state vectors from the environment, and the action space is a continuous vector normalized to $[-1, 1]$. Performance is measured as the average return over 8 evaluation episodes, with results averaged across 2 random seeds.

### C.2 PERFORMANCE COMPARISON: SCALEZERO VS. UNIZERO

As summarized in Table 1b and detailed in Figure 9, the multitask ScaleZero model achieves performance competitive with, and often superior to, the single-task UniZero baseline. Notably, ScaleZero achieves a higher median score, indicating robust generalist performance across the majority of tasks rather than excelling on only a few. This validates that the architectural principles of ScaleZero are effective for complex, state-based continuous control problems. We hypothesize this success stems from the MoE layers learning to specialize in shared physical priors and control primitives (e.g., balancing vs. locomotion), which are then composed by the router based on the task.

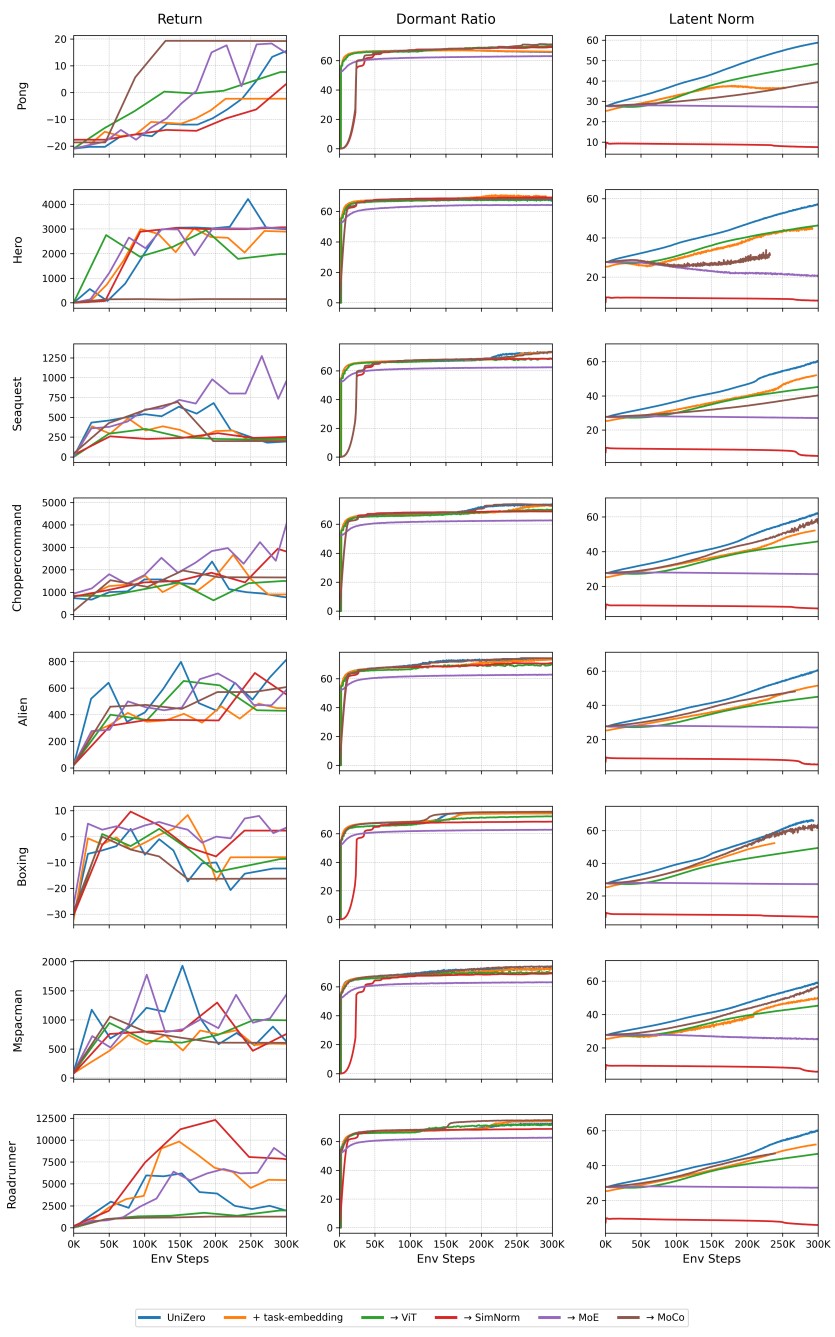

**Figure 7: Complete plasticity metric analysis for the ScaleZero design space exploration.** This figure provides the full data for the ablation study, detailing the evolution of performance (Return), Dormant Ratio, and Latent Norm for different model variants across all 8 tasks in the Atari8 multitask benchmark. The MoE-based model consistently outperforms others by maintaining superior plasticity metrics.

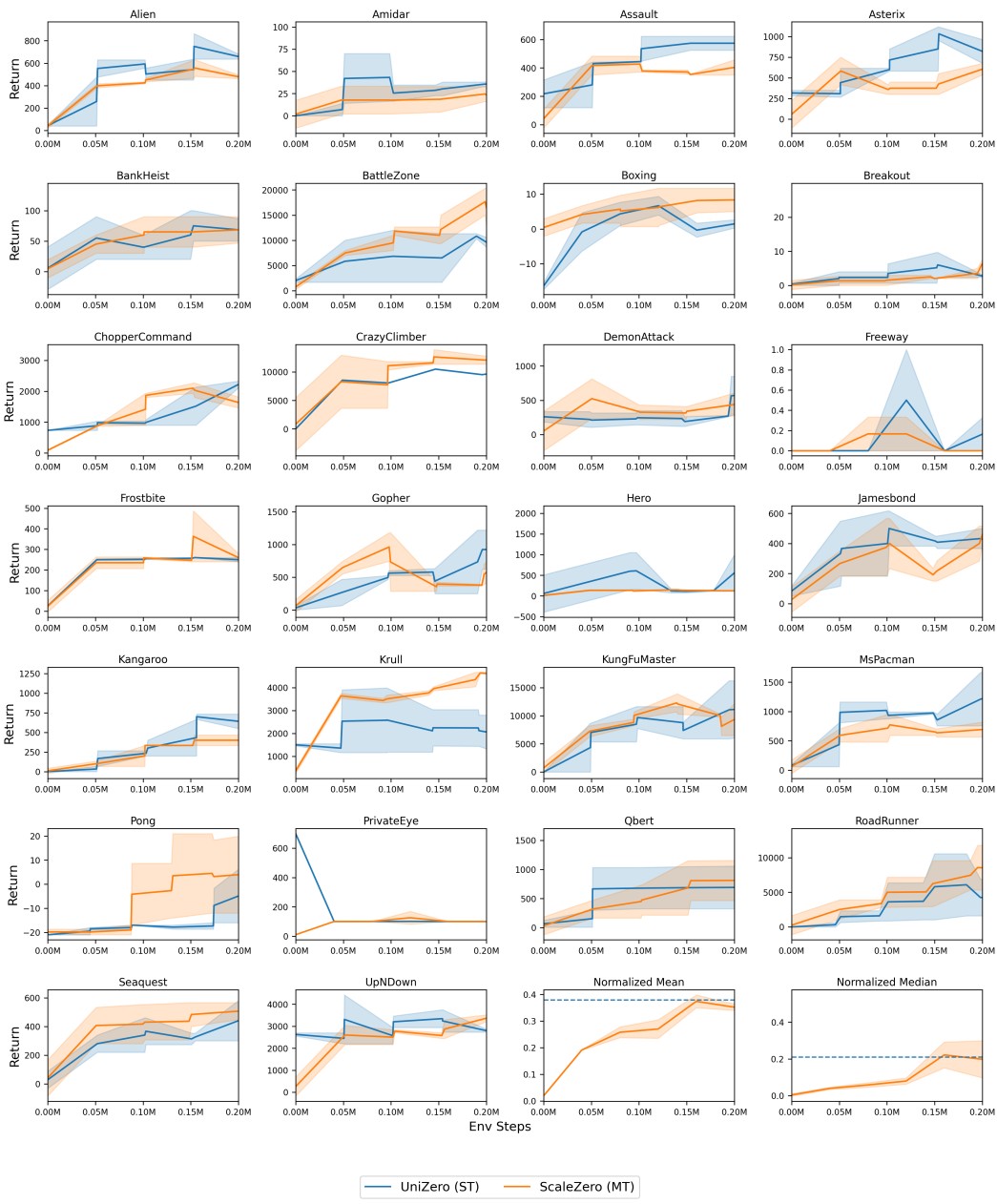

**Figure 8: Learning curves for ScaleZero (MT) vs. UniZero (ST) on the Atari26 benchmark.** This figure shows the full per-game learning curves (mean and 95% confidence interval) for the multitask ScaleZero agent compared against 26 single-task UniZero baselines.

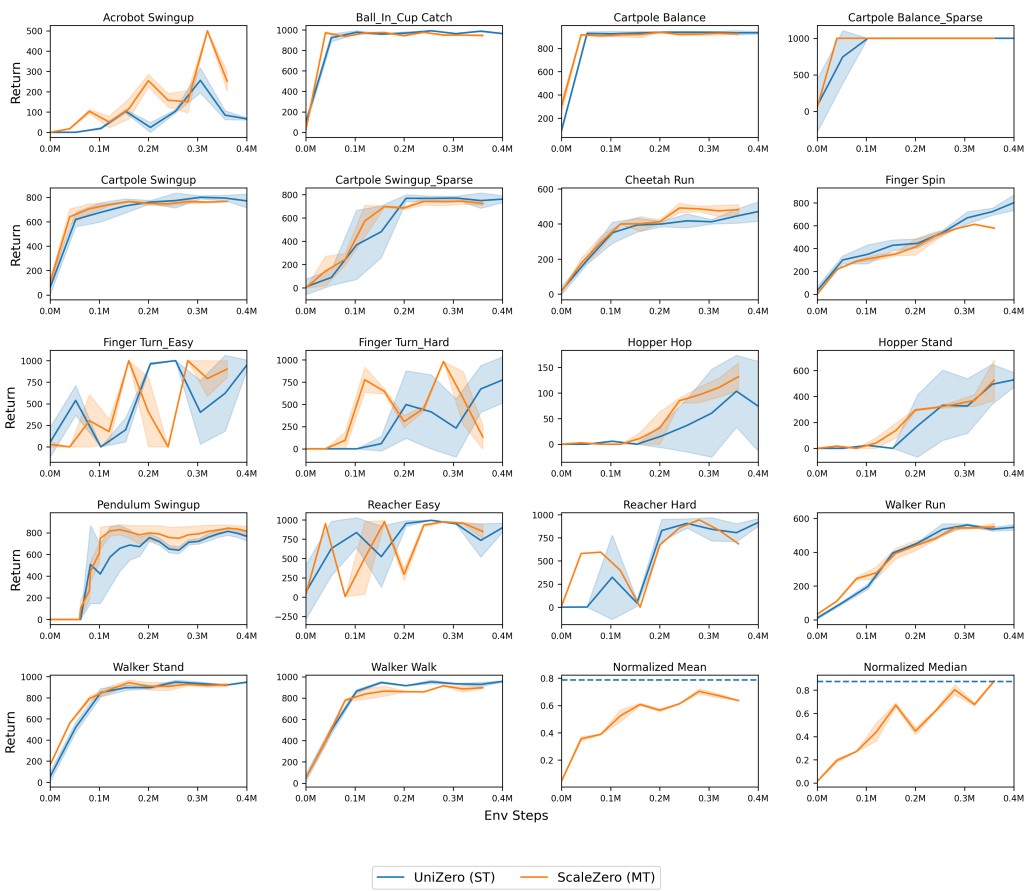

**Figure 9: Learning curves of ScaleZero (MT) vs. UniZero (ST) on the DMC18 benchmark.** This figure provides the full per-task learning curves, comparing the performance of the single multitask ScaleZero agent against 18 specialized UniZero baselines. Solid lines represent the mean performance over 2 random seeds, and the shaded area indicates the 95% confidence interval.

## C.3    Efficiency Evaluation of Dynamic Parameter Scaling (DPS)

This section provides the detailed learning curves that substantiate the efficiency gains of Dynamic Parameter Scaling (DPS), as reported in Section 5.2. Figure 10 compares the per-task sample efficiency of **ScaleZero-DPS** against the standard ScaleZero model.

The curves demonstrate that ScaleZero-DPS (orange) learns significantly faster on most tasks, achieving target performance levels with fewer environment interactions. This validates the claim that DPS leads to a substantial reduction in sample complexity. The asterisks (*) denote tasks where training was terminated early by the DPS policy upon reaching a "solved" threshold, visually confirming the dynamic allocation of computational resources away from mastered tasks.

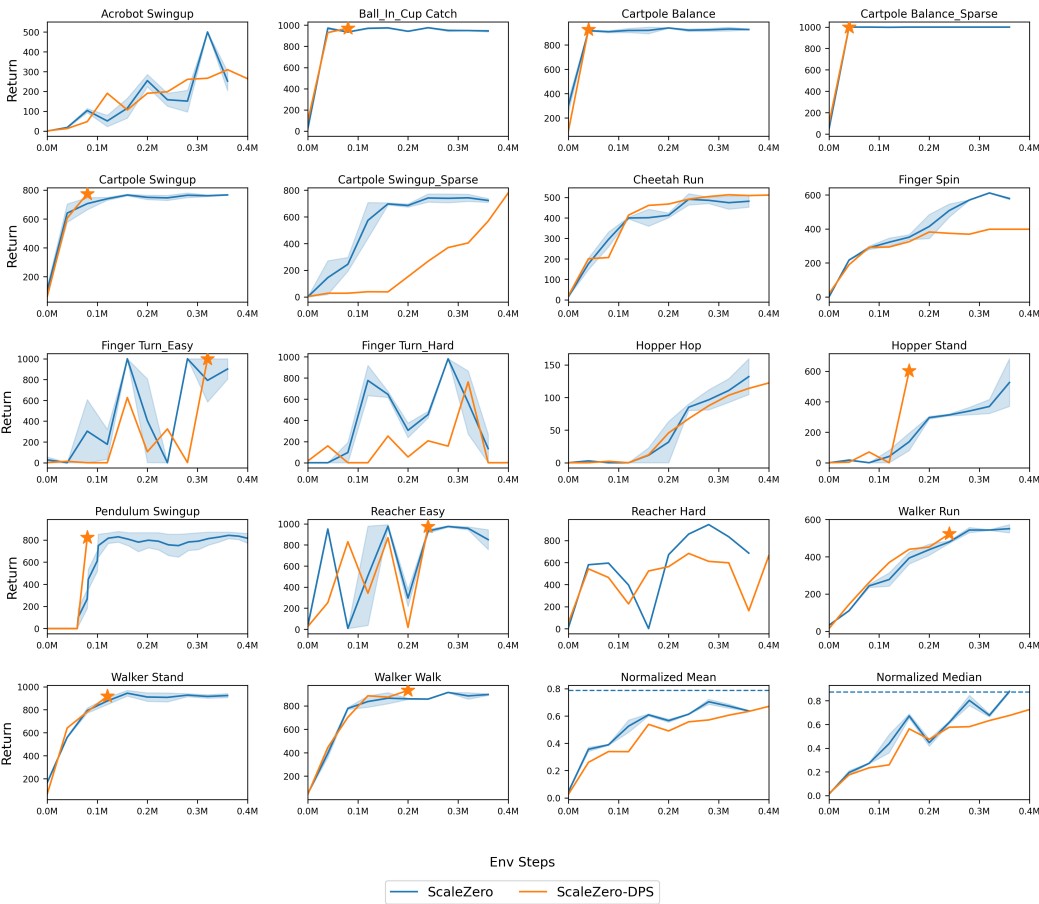

Figure 10: **Sample efficiency comparison of ScaleZero-DPS vs. standard ScaleZero on DMC18.** This figure shows the learning curves (return vs. environment steps) for both models across 18 tasks. The faster rise of the ScaleZero-DPS curves (orange) illustrates its superior sample efficiency. An asterisk (*) indicates that the DPS policy halted training for that task, demonstrating the dynamic resource allocation mechanism in action.

## C.4    Mechanistic Analysis of Dynamic Parameter Scaling

To understand the mechanisms by which Dynamic Parameter Scaling (DPS) achieves its sample efficiency, we conduct a granular analysis of the agent's internal parameter dynamics. This section presents two complementary visualizations: a time-series plot showing the continuous evolution of key metrics (Figure 11), and a suite of "importance matrices" providing discrete snapshots at critical training junctures (Figure 12).

### C.4.1 SETUP

**Time-Series Dynamics (Figure 11):** This figure provides a macroscopic view of the training process. The top panel correlates high-level metrics—normalized performance scores and the number of solved tasks—with the progression of training stages. The bottom panel offers a view into the internal mechanics by tracking the average learnable importance ('alpha' scale) of LoRA adapters. It specifically compares the behavior of adapters in the first Transformer layer (Layer 0, solid lines) versus the last (Layer 3, dashed lines), highlighting the emergence of different dynamic behaviors at different model depths.

**Stage-Wise Importance Matrices (Figure 12):** This multi-panel figure offers a more granular, event-based perspective. Each matrix is an "importance matrix" where a cell at '(row r, column c)' shows the average 'alpha' scale of the adapter introduced in Stage 'c', as measured at the end of Stage 'r''s training. This reveals how the model re-evaluates the contribution of all historical adapters after each phase of targeted learning. To provide a comprehensive view, this analysis is presented for:

- The overall average across all layers and adapter types (Panel a).
- A per-layer breakdown for Layer 0 and Layer 3 to show hierarchical differences (Panels b-c).
- A per-type breakdown for adapters applied to the Query, Key, Value, and output Projection linear layers to reveal functional specialization (Panels d-g).

### C.4.2 RESULTS AND ANALYSIS

The analysis of these figures reveals several key dynamics of the DPS-managed learning process. The time-series plot (Figure 11) illustrates the staged training protocol. In this particular experiment, the stage transitions (vertical dashed lines) are triggered by a pre-allocated iteration budget, consistent with the method's budget-based trigger mechanism. These capacity expansions occur as overall performance and the number of solved tasks increase over the course of training. The divergence in 'alpha' scales between Layer 0 and Layer 3 suggests that the model applies different update strategies to shallow versus deep layers as training progresses.

The importance matrices in Figure 12 provide further detail on this behavior:

- **Overall Strategy (Panel a):** The data is consistent with a dual mechanism of knowledge retention and plasticity. The importance scale of the foundational adapter from Stage 0 (column 0) is not static; it increases from 1.0 to 1.097 by the end of training. This suggests that the model increases the relative contribution of its initial knowledge as it encounters more complex tasks. Concurrently, the diagonal values show how the model focuses on the newest adapter during each stage, which is then re-weighted in subsequent stages as its knowledge is integrated.

- **Hierarchical Differentiation (Panels b vs. c):** A notable difference is observed between Layer 0 and Layer 3. In Layer 0 (Panel b), the importance of the initial adapter (cell (4,0)) exhibits high stability with moderate growth (to 1.058), indicating its role as a stable foundation for low-level features. In contrast, Layer 3 (Panel c) shows more pronounced dynamic re-weighting. The foundational adapter's importance grows more significantly (to 1.099), suggesting that higher-level, abstract representations rely heavily on and amplify this core knowledge. The greater variance in Layer 3's alpha values indicates that later layers are more involved in adaptive, task-specific strategy modifications.

- **Functional Differentiation by Layer Type (Panels d-g):** The analysis reveals distinct behaviors based on which linear layer within the attention mechanism an adapter modifies.
  - **Query (Q) and Key (K) Adapters** (Panels d, e): Adapters applied to the Q and K layers, which produce the representations for calculating attention scores, generally exhibit importance scales that increase over time. For instance, the Stage 0 Key adapter's scale reaches 1.104. This suggests a continuous strengthening of the mechanisms that determine token-to-token relevance.
  - **Value (V) and Projection (Proj) Adapters** (Panels f, g): In contrast, adapters for the V layer (which provides the content to be aggregated) and the output Projection layer (which combines attention head outputs) show more complex dynamics, including suppression (alpha < 1.0). For example, the Value adapter from Stage 3 is down-weighted to 0.889 after Stage 4. This behavior may indicate a more cautious integration of new value-based information or a re-balancing of attention head outputs to integrate new skills without disrupting existing ones.

In summary, the DPS method facilitates a structured learning process. The model appears to preserve and amplify foundational knowledge, primarily in early layers, while enabling dynamic, adaptive strategy-switching in later layers. Furthermore, it modulates the importance of adapters based on their specific function within the self-attention mechanism (Query/Key vs. Value/Projection). This combination of knowledge retention and targeted, hierarchical plasticity offers a mechanistic explanation for the method's observed sample efficiency.

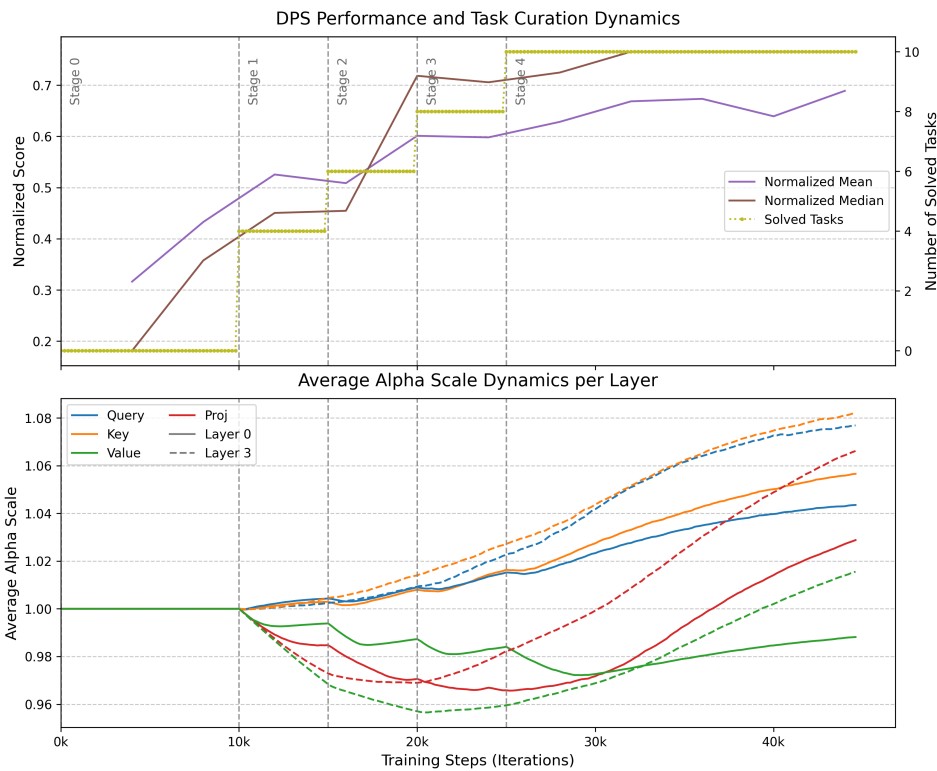

**Figure 11: Internal dynamics of ScaleZero-DPS during training. (Top)** Performance and task-solving progress across stages. **(Bottom)** Average 'alpha' scales for adapters in the first (Layer 0, solid) and last (Layer 3, dashed) layers, showing the emergence of hierarchical specialization.

## D  JERICHO EXPERIMENT DETAILS

### D.1  BENCHMARK SETUP

**Environment Overview:** Jericho (Hausknecht et al., 2020) is a reinforcement-learning benchmark built on classic text-adventure games, played entirely through natural-language interaction. Unlike Atari or DMC, it demands robust language understanding of free-form scene, item, and state descriptions while contending with a combinatorial, effectively unbounded action space—at each step the environment surfaces only a small candidate set, yet many legal commands outside it still execute—and coping with sparse, delayed rewards that require long-horizon exploration and planning, making Jericho a stringent testbed for text-based RL.

We conduct experiments on four representative tasks from the Jericho benchmark, namely *Detective*, *Acorncourt*, *Omniquest*, and *Zork1*. Examples of expert interaction trajectories for *Zork1* and *Detective* are illustrated in Figure 13. Following the protocol of the previous two benchmarks, we compare our multitask ScaleZero (MT) with the single-task UniZero (ST) baseline. ScaleZero uses the same text encoder (bge-base-en-v1.5), the same *inference context length* of 4, and the same per-step *max sequence length* of 512 tokens as UniZero. The *max action num* and *max steps* for the four tasks are given in Table 13. Results are reported as the mean return over two random seeds. If, at

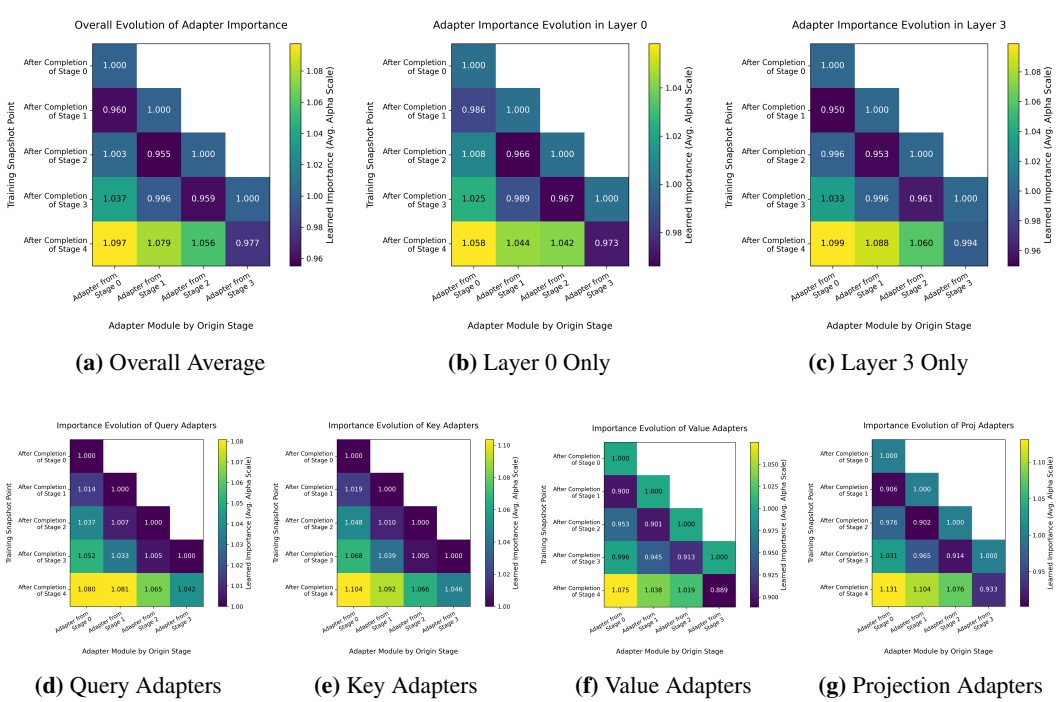

**Figure 12: Evolution of Adapter Importance Across Training Stages and Model Components.** Each matrix displays the learned importance (average 'alpha' scale) of adapters. A cell at '(row r, col c)' represents the importance of the adapter from Stage 'c' after Stage 'r''s training is complete. **(a)** The overall average shows a clear pattern of knowledge preservation (column 0) and targeted plasticity (diagonal). **(b-c)** The layer-specific breakdown reveals that foundational knowledge is more rigidly preserved in early layers (Layer 0) while later layers (Layer 3) are more dynamic and adaptive. **(d-g)** The type-specific breakdown shows functional differentiation in how adapters are utilized over time, reflecting their distinct roles in the attention mechanism.

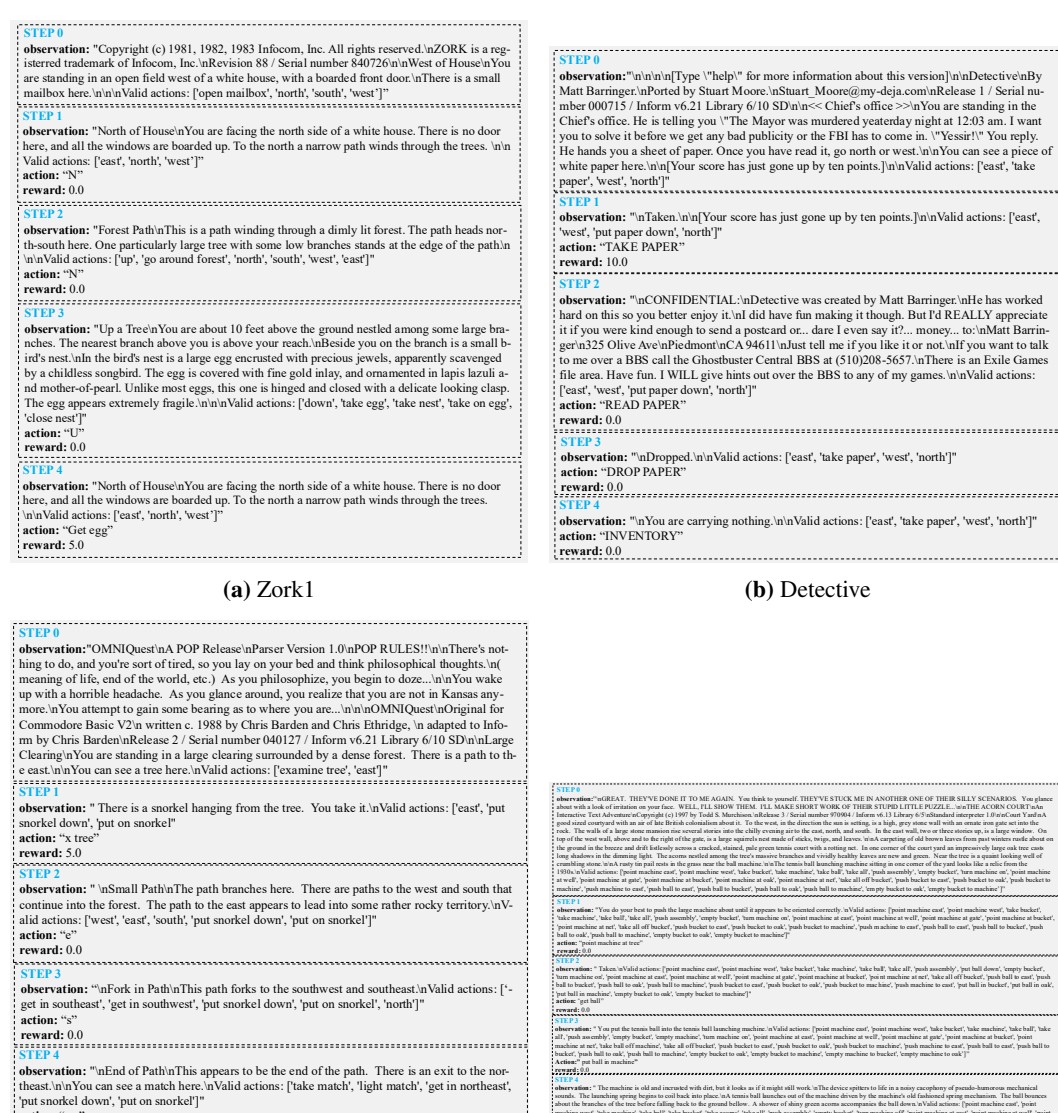

**Figure 13:** First five steps of expert interaction trajectories across environments. At each step, the environment supplies a set of valid candidate actions (used in our experiments); however, additional legal actions outside this set may also be executed, so the action space is, in principle, unbounded.

any step, the valid action set is smaller than *max action num*, we pad to *max action num* and mark the padded entries with an *action mask* so they are excluded from sampling.

## D.2 PERFORMANCE COMPARISON: SCALEZERO VS. UNIZERO (ST)

As shown in Figure 14, the multitask ScaleZero attains returns comparable to the single-task UniZero baseline on most Jericho games, with learning curves exhibiting an "easy-first, hard-later" progression. Specifically, ScaleZero converges faster on *Acorncourt*, *Omniquest*, and *Detective*; on the most challenging *Zork1*, it initially lags but accelerates after 150k environment interactions to reach parity with UniZero. We hypothesize that this stems from multitask training inducing transferable language priors and interaction routines, which help filter feasible commands and reduce fruitless exploration.

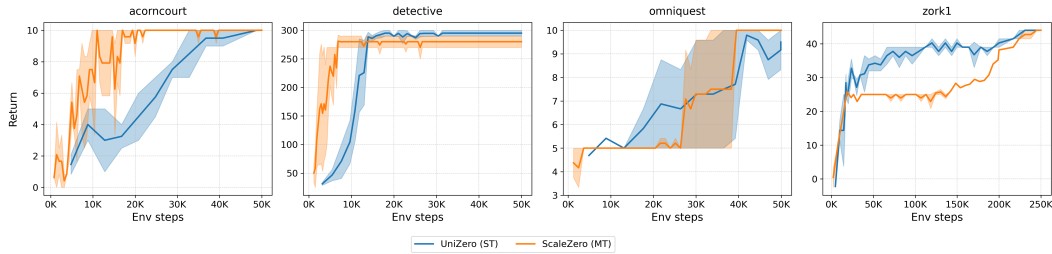

**Figure 14: Learning curves of ScaleZero (MT) vs. UniZero (ST) on the Jericho benchmark.** This figure compares the performance of the multitask ScaleZero and single-task UniZero models on 4 Jericho tasks. Solid lines represent the mean performance over 2 random seeds, and the shaded area indicates the 95% confidence interval.

| Game | maximum valid actions (in 200 episodes) | maximum steps per episode (in 200 episodes) | max action num | max steps |
|---|---|---|---|---|
| Acorncourt | 34 | 17 | 45 | 100 |
| Omniquest | 24 | 78 | 25 | 100 |
| Zork1 | 53 | 396 | 55 | 500 |
| Detective | 11 | 51 | 12 | 100 |

**Table 13:** Key game statistics for the four Jericho tasks. Maximum valid actions denotes the largest number of valid action candidates exposed by the environment at any step, observed over 200 evaluation episodes; this statistic informs the *max action num* parameter used in our experiments. Maximum steps per episode is the largest episode length observed over the same 200 episodes, while *max steps* parameter is the episode-length cap adopted during experiments.

# E    GRADIENT ANALYSIS IN MOE

## E.1    EXPERIMENTAL ANALYSIS

To understand *why* Mixture-of-Experts (MoE) architectures excel in multitask settings, this study investigates the core mechanisms driving their performance. Focusing on multitask reinforcement learning, we conduct a dual analysis combining theoretical inquiry with empirical validation. The following section details the experimental protocol designed for this purpose.

### E.1.1    EXPERIMENT 1: ANALYZING GRADIENT CONFLICTS IN MOE-BASED TRANSFORMERS

We conduct our experiments on Atari-8. Concretely, our network architecture consists of an encoder (ViT), a backbone (Transformer), and corresponding heads. We compare two baseline methods with different backbones:

(1) **Naive Transformer:** The backbone consists of four standard Transformer blocks.

(2) **MoE-based Transformer:** The backbone also consists of four Transformer blocks, but the MLP layer in each block is replaced with an MoE layer, which comprises one shared expert and eight non-shared experts(Liu et al., 2024). Shared experts provide cross-task general representations, enhancing generalization ability and ensuring training stability, while non-shared experts learn task-specific representations to strengthen the model's discriminability and task adaptability. During the forward pass, all non-shared experts are selectively activated by a *sparse gating network*, which determines which specific expert to use for each input.

For both baseline models, we investigate gradient conflicts at different components: *(1) Input before the MoE layer. (2) Output of the encoder. (3) Parameters of the MoE, including the shared expert, non-shared experts, and the entire MoE layer.* We measure gradient conflict between tasks using the maximum negative cosine similarity, defined as follows:

$$\text{Max Gradient Conflict} = \max_{i,j} \left( -\frac{\nabla\theta_i \cdot \nabla\theta_j}{\|\nabla\theta_i\|\|\nabla\theta_j\|} \right) \qquad (10)$$

where $\nabla\theta_i$ and $\nabla\theta_j$ denote the gradients of the $i$-th and $j$-th tasks, respectively. A higher value of the *Max Gradient Conflict* represents a greater degree of gradient conflict. We choose the *maximum*

pairwise cosine similarity because it directly identifies the most severe gradient conflict between any two tasks. Unlike *averaging* operation, which can hide critical issues, the maximum value pinpoints the 'bottleneck pair' that most significantly impedes stable multitask training and overall convergence, even if other tasks cooperate well.

In MoE, when computing the gradient of an entire layer, if task $A$ selects expert $i$ while task $B$ selects expert $j$ with $i \neq j$, then the gradient of task $B$ on expert $i$ is filled with zero. The reason is that expert $i$ is not involved in the forward propagation of task $B$, and thus makes no contribution to its loss; consequently, its gradient during backpropagation should naturally be zero.

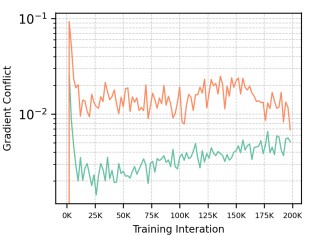 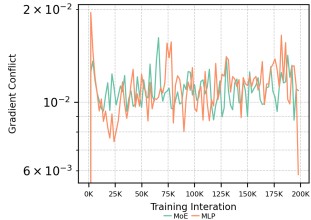 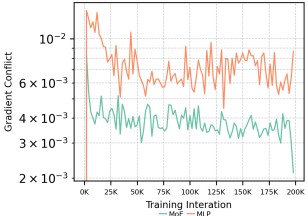

**(a)** Conflict comparison of MoE layer gradients.

**(b)** Conflict comparison of encoder output gradients.

**(c)** Conflict comparison of MoE input gradients.

**Figure 15:** Comparison of gradient conflicts between MoE and MLP baselines across different components. MoE-based Transformer exhibits fewer gradient conflicts in MoE input and MoE layer.

**Observation 1:** As shown in Figure 15 (a), the MoE-based Transformer exhibits fewer gradient conflicts in the MoE layer than its MLP counterpart. Furthermore, Figure Figure 15 (c) shows that introducing MoE also reduces gradient conflicts at the MoE layer's input, implying that MoE alleviates conflicts in other components to some extent. However, at the output of the shared encoder, as depicted in Figure Figure 15 (b), the gradient conflict levels for the MoE and MLP models are largely comparable and do not show a significant difference. This indicates that the advantage of using MoE in the backbone to mitigate gradient conflicts is primarily localized to the MoE layers themselves and their immediate downstream connections; this effect does not substantially propagate back to the upstream shared encoder.

We posit that since the encoder functions as a general feature extractor for all tasks, its gradient dynamics are likely dominated by the need to learn common representations, making it less sensitive to architectural changes in downstream modules.

### E.1.2 EXPERIMENT 2: INVESTIGATING MoE GATING MECHANISMS

We are particularly interested in the internal gating mechanism of MoE. Previous studiesChen et al. (2022) have shown that, under supervised learning, *MoE can implicitly uncover latent cluster structures within input space*. However, when faced with non-stationary data distributions generated through agent-environment interactions, it remains unclear whether MoE experts still differentiate effectively. To answer the question, we analyze the *entropy of expert selection distributions*, which quantifies the uncertainty of the choice of which expert to use for a task, with low entropy indicating high specialization and decisive selection, while high entropy implies uncertainty or an average utilization across experts. We also record the *Wasserstein distance* (Rüschendorf, 1985) between the expert selection distributions of different tasks, where smaller values indicate a greater proximity to the expert selection of two tasks. We aim to quantify this relationship to find the underlying connections in expert selection among various tasks. The specific experimental steps are as follows:

*(STEP1)* During a forward pass at a given training step, we record the expert choices in the final MoE-based Transformer block.

*(STEP2)* For a specific training step $s_t$, we collect expert selection data over a past window of size $S$ and compute the frequency of each expert's activation to form a probability distribution. For a given window size $S$, task $i$ and a specific task $j$, we denote the task selection distribution as $P_i^S$ and a

specific probability for task $j$ as $P_{i,j}^S$. Different window sizes reflect attention to data over different temporal scales in non-stationary learning.

*(STEP3)* Based on this probability distribution, we calculate the *expert selection entropy* for each task $E_{task_i}$:

$$E_{task_i} = -\sum_{j=1}^{N} P_{i,j}^S \log_2(P_{i,j}^S)$$

and *Wasserstein Distance between task $i$ and task $i'$ $(W_{i,i'})$*

We consider two sizes $S$: *immediate* = 100, *short* = 1,000.[4] We present the entropy results in Figure 16. Due to space limitations, we show in Figure 19 the Wasserstein distances between expert selections of different tasks at multiple training stages.

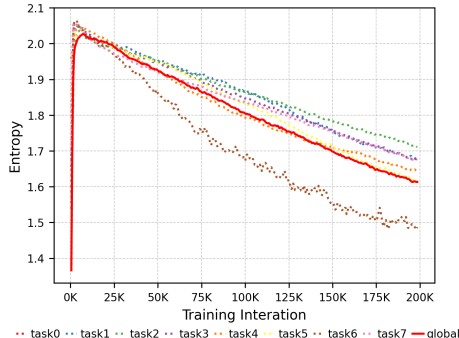

**Figure 16:** Line plot showing the evolution of expert selection entropy in a multitask learning setting with eight tasks. The dashed lines correspond to the entropy values of individual tasks, and the solid red line represents the aggregated entropy across all tasks. Higher entropy reflects more uniform and uncertain expert utilization, while lower entropy reflects more concentrated and specialized expert selection.

**Observation 2:** MoE differentiation and expert specialization. As shown in Figure 16, as tasks progress, the entropy of the expert distribution gradually decreases, indicating a reduction in distributional uncertainty. This suggests that expert selection becomes concentrated on a few outcomes, reflecting lower uncertainty. Furthermore, in Figure 19, we visualize the expert selection distributions of different tasks at multiple training stages, offering additional evidence of the progressive specialization in expert utilization.

### E.1.3 EXPERIMENT 3: ANALYZING GRADIENT CONFLICTS BETWEEN SHARED EXPERT AND NON-SHARED EXPERT

To further investigate the relationship between expert selection and gradient dynamics within MoE, we employ a MoE-based Transformer to analyze gradient conflicts across different MoE components. Our experimental results are shown below.

**Observation 3:** As shown in Figure 17, the shared expert exhibits significantly higher gradient conflicts compared to the task-specific experts. Among the non-shared experts, the level of conflicts does not show substantial differences. Overall, the shared expert accumulates even more conflicts than the entire MoE layer, indicating that most gradient conflicts within the MoE layer are concentrated on the shared expert. The introduction of several task-specific experts effectively reduces the overall gradient conflicts of the MoE layer, which is consistent with our theoretical analysis in Theorem E.4.

We further investigated the conflicts among different experts within MoE. Results is shown in Figure 17 *We found that shared experts bear most of the gradient conflicts, whereas individual (non-shared) experts experience almost no conflict.* A plausible explanation is that in a standard gating MoE, different samples are routed to different experts, so each expert primarily receives gradient updates from a specific type of sample. This effectively performs an implicit task partitioning, leading

---

[4]Note that at the very beginning of training, when the window cannot be fully populated, we use the available data within the window, which may lead to larger fluctuations at early steps.

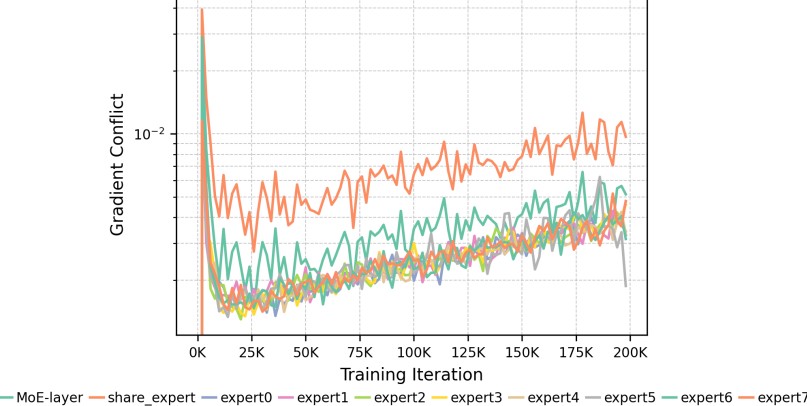

**Figure 17:** Gradient conflicts across experts in MoE training. The plot shows the evolution of gradient conflicts for the shared expert and 8 non-shared experts during training interations. The logarithmic y-axis reveals that different experts experience varying levels of gradient conflicts, with the shared expert and individual experts (expert0–expert7) demonstrating distinct conflict patterns, indicating effective expert specialization in the multitask learning framework.

to generally consistent gradient directions within the same expert and minimal conflicts. In contrast, in shared-expert MoE, all samples pass through the shared experts, whose parameters must adapt simultaneously to multiple tasks, diverse semantics, and even different domain distributions. This can result in highly inconsistent gradient directions, significantly increasing conflicts and making it difficult for shared experts to balance diverse task requirements during optimization.This observation naturally raises a simple question: *is the alleviation of gradient conflicts mainly due to the sparse selection by the gating network?* We provide a theoretical justification from the perspective of gating.

### E.2 THEORETICAL ANALYSIS

We observe that the magnitude of the gradient conflict in an MoE model is primarily influenced by the routing coefficient, $\rho_i$. In a sparse MoE, this coefficient acts as a binary variable determined by the router network. For simlicity, let us consider a top-1 sparse MoE, where only the expert with the highest routing score is selected for a given input. An effective router would learn to assign tasks that are prone to high gradient conflict to different experts. In this case, the routing coefficient for any conflicting task pair would be zero, thereby nullifying their contribution to the overall conflict. Conversely, a poorly performing router might allocate conflicting tasks to the same expert, in which case the conflict remains unmitigated.

**Theorem E.1** (Upper Bound of Full-layer MoE Gradient Conflict with Sparse/Soft Routing). *Consider a MoE layer with $M$ experts. Let each expert's gradient norm satisfy $\|g_t^{(m)}\| \leq R, \quad m = 1, \dots, M$, and let routing weights for a task be $\lambda_1, \dots, \lambda_M \geq 0$. The full-layer gradient is $G_t = \text{concat}(\lambda_1^t g_t^{(1)}, \dots, \lambda_M^t g_t^{(M)})$, Assume that for any task pair $(t_1, t_2)$, the gradient conflict on a single expert is bounded by $G$. Then the full-layer MoE gradient conflict satisfies the explicit upper bound*

$$\text{conflict}(G_{t_1}, G_{t_2}) \leq G \cdot \frac{\sum_{m=1}^{M} \lambda_m^{t_1} \lambda_m^{t_2} \|g_{t_1}^{(m)}\| \|g_{t_2}^{(m)}\|}{\sqrt{\sum_{m=1}^{M} (\lambda_m^{t_1})^2 \|g_{t_1}^{(m)}\|^2} \sqrt{\sum_{m=1}^{M} (\lambda_m^{t_2})^2 \|g_{t_2}^{(m)}\|^2}} \leq G$$

*Proof.* From the block-level conflict bound we have

$$\text{conflict}(G_{t_1}, G_{t_2}) \leq G \cdot \frac{\sum_{m=1}^{M} \lambda_m^{t_1} \lambda_m^{t_2} \|g_{t_1}^{(m)}\| \|g_{t_2}^{(m)}\|}{\|G_{t_1}\| \|G_{t_2}\|},$$

with

$$\|G_{t_1}\| = \sqrt{\sum_{m=1}^{M} (\lambda_m^{t_1})^2 \|g_{t_1}^{(m)}\|^2}, \qquad \|G_{t_2}\| = \sqrt{\sum_{m=1}^{M} (\lambda_m^{t_2})^2 \|g_{t_2}^{(m)}\|^2}.$$

Define vectors $u, v \in \mathbb{R}^M$ as

$$u_m = \lambda_m^{t_1} \|g_{t_1}^{(m)}\|, \qquad v_m = \lambda_m^{t_2} \|g_{t_2}^{(m)}\|.$$

Then the numerator equals $\langle u, v \rangle$, and the denominator equals $\|u\| \|v\|$. By the Cauchy–Schwarz inequality,

$$\langle u, v \rangle \leq \|u\| \|v\|,$$

which implies that the fraction is at most $1$. Substituting this bound back, we obtain

$$\text{conflict}(G_{t_1}, G_{t_2}) \leq G.$$

Equality holds if and only if $u$ and $v$ are collinear (i.e., there exists $c > 0$ such that $u = c\, v$), and each block-level conflict achieves its upper bound simultaneously. $\qquad\square$

**Remark 1** (Effect of Routing Strategies on Full-layer Gradient Conflict). *The upper bound of full-layer MoE gradient conflict depends strongly on the routing strategy. We summarize two representative cases:*

*a. **Dense Routing (Soft Gating,** $\sum_{m=1}^{M} \lambda_m = 1$**)***

- *All experts contribute to the full-layer gradient.*

- *The fraction in Theorem 1 can be interpreted as the cosine similarity between the weighted vectors $u = \lambda^{t_1} \odot \|g_{t_1}\|$ and $v = \lambda^{t_2} \odot \|g_{t_2}\|$:*

$$\frac{\sum_{m=1}^{M} \lambda_m^{t_1} \lambda_m^{t_2} \|g_{t_1}^{(m)}\| \|g_{t_2}^{(m)}\|}{\|G_{t_1}\| \|G_{t_2}\|} = \cos(u, v) \leq 1.$$

- *If the expert norms are roughly equal ($\|g_{t_1}^{(m)}\| \approx \|g_{t_2}^{(m)}\|$), the conflict upper bound is largely determined by $\cos(\lambda^{t_1}, \lambda^{t_2})$.*

- *Uniform weights ($\lambda_m = 1/M$) do not automatically reduce conflict; alignment of weighted gradients matters more.*

- *Overlap or alignment of routing vectors increases conflict, while orthogonal or disjoint routing vectors reduce it.*

*b. **Sparse Routing (Top-1 gating, one** $\lambda_m = 1$**, others** $0$**)***

- *Non-overlapping case (different tasks select different experts):Each task's full-layer gradient contains only its chosen expert. Inner products between task gradients are near zero because tasks lie in different expert subspaces. Full-layer conflict is strictly less than $G$, possibly near 0.*

- *Collapsed case (all tasks select the same expert): Full-layer gradient reduces to the chosen expert's gradient. Conflict equals the single-expert bound $G$, since all gradients reside in the same subspace.*

> **Takeaways**
>
> - *The gradient conflict of MoE with multiple experts is lower than that within a single MLP-based expert.*
>
> - *The mitigation of gradient conflict in MoE is closely related to the routers across tasks—more orthogonal routing coefficients tend to yield fewer gradient conflicts.*

Therefore, the key to mitigating gradient conflict lies in the router's ability to foster specialization among experts, ensuring that tasks with potentially conflicting objectives are handled by distinct, differentiated experts. This naturally raises a critical question: Can a Mixture-of-Experts architecture indeed achieve this, and if so, what are the underlying mechanisms that enable this capability? In the following, we answer this question affirmatively. By building upon existing foundational theories of Mixture-of-ExpertsChen et al. (2022), we present a formal derivation that explains precisely when and how this is achieved.

### E.2.1 ANALYSIS

Previews workChen et al. (2022) theoretically demonstrated that, *in a single-task supervised learning setting, when the input distribution exhibits distinguishable cluster structures, the sparse gating Mixture-of-Experts (MoE) router can automatically learn the cluster center features and route samples to the most suitable expert*, thereby significantly improving performance. We further hypothesize that, in multitask learning, the input spaces of different tasks naturally correspond to distinct clusters and the signal for the cluster center can be task ID or the latent pattern on the state space. Under this assumption, the conclusions from the original work can be directly extended to the multitask setting.

Next, we adapt Chen et al. (2022) theoretical analysis to the reinforcement learning context using a tabular example. Specifically, we treat each task as an independent Markov Decision Process (MDP) and construct a multitask MDP set with a shared state space but task-specific transitions. In this setting, we derive the gradient conflicts that arise in the MoE layer.

We also discuss the case where no clustering signal exists between tasks and point out that the introduction of task embeddings can significantly enhance the clustering structure between tasks, thereby promoting the isolation of expert selection across different tasks in MoE.

### E.2.2 PROBLEM SETTING: MULTITASK MDP WITH TASK-SPECIFIC LATENT CLUSTERS

We consider a Markov decision process (MDP) defined by the tuple $(\mathcal{S}, \mathcal{A}, P, R, \gamma)$, where $\mathcal{S}$ is the state space, $\mathcal{A}$ the action space, $P(s' \mid s, a)$ the transition kernel, $R(s, a)$ the reward function, and $\gamma \in [0, 1)$ the discount factor. There are $K$ tasks $T_1, \ldots, T_K$. Each task $T_t$ corresponds to a disjoint state subspace $\mathcal{S}^{(t)}$, so that

$$\mathcal{S} = \bigcup_{t=1}^{K} \mathcal{S}^{(t)}, \qquad \mathcal{S}^{(t)} \cap \mathcal{S}^{(t')} = \varnothing \quad \text{for } t \neq t'.$$

**Patch-based state representation.** Each state $s \in \mathcal{S}_k^{(t)}$ (i.e., belonging to task $t$ and cluster $k$) is represented as an unordered collection of $P$ patches in $\mathbb{R}^d$. The patches are randomly permuted before being presented to the model. Every state contains the following patch types:

1. **Action-signal patch:** exactly one patch equals $\alpha \, v_k^{(t)}$, where $v_k^{(t)} \in \mathbb{R}^d$ encodes the optimal-action feature.

2. **Cluster-center patch:** exactly one patch equals $\beta \, c_k^{(t)}$, indicating the (task-specific) cluster identity; a router (e.g., in a Mixture-of-Experts model) is expected to detect this signal to decide routing.

3. **Feature-noise (confounder) patch:** exactly one patch equals $\epsilon \gamma \, v_{k'}^{(t)}$, where $k'$ is typically a different cluster index in the same task $t$, modeling an intra-task confounding feature.

4. **Random-noise patches:** the remaining $P - 3$ patches are i.i.d. draws from an isotropic Gaussian, $\mathcal{N}\!\left(0, \frac{\sigma_p^2}{d} I_d\right)$.

Thus the encoder receives an unordered set $\{x_1, \ldots, x_P\} \subset \mathbb{R}^d$ containing exactly one action signal, one cluster-center signal (task-specific), one intra-task confounder, and Gaussian noise patches.

**MoE Model with Expert Specialization**

We now analyze how a Mixture-of-Experts (MoE) model can leverage expert specialization to address the aforementioned issues.

**Experts.** We consider $M$ linear experts. The value function of expert $m$ is defined as

$$V_m(s; w_m) = \langle w_m, \phi(s) \rangle. \tag{11}$$

**Router.** The gating network assigns a score to each expert $m$:

$$h_m(s; \theta_m) = \langle \theta_m, \phi(s) \rangle. \tag{12}$$

**MoE Output.** The overall value function is given by

$$V_{\text{MoE}}(s; W, \Theta) = \sum_{m=1}^{M} \pi_m(s) V_m(s; w_m), \tag{13}$$

where the softmax gating weights are

$$\pi_m(s) = \frac{\exp(h_m(s))}{\sum_{j=1}^{M} \exp(h_j(s))}. \tag{14}$$

**Loss Function.** We minimize the mean squared Bellman error (MSBE) with a stop-gradient:

$$L(W, \Theta) = \frac{1}{2} \sum_{s \in \mathcal{S}} \left( V_{\text{MoE}}(s) - \text{stop\_grad} \left( (\mathcal{T}^* V_{\text{MoE}})(s) \right) \right)^2 = \frac{1}{2} \sum_{s \in \mathcal{S}} \delta(s)^2, \tag{15}$$

where $\delta(s)$ denotes the temporal-difference (TD) error.

### E.2.3 LEARNING DYNAMICS ANALYSIS

We follow an analysis path similar to prior workChen et al. (2022), dividing the learning process into two stages: an early *expert exploration* stage and a later *router learning* stage.

**Initialization.**

The expert weights $w_m^{(0)}$ are randomly initialized from a small zero-mean Gaussian distribution:

$$w_m^{(0)} \sim \mathcal{N}(0, \sigma_w^2 I), \tag{16}$$

while the router weights $\theta_m^{(0)}$ are initialized as zero vectors:

$$\theta_m^{(0)} = \mathbf{0}. \tag{17}$$

**Expert Exploration Stage.**

According to [Chen et al. (2022), Lemma E.3], the zero initialization of the router implies that at the beginning of training ($t = 0$),

$$\max_{m \in [M]} \left| P(m_{i,t} = m) - \frac{1}{M} \right| = \tilde{O}(\sigma_0^{1.5}) \quad \text{for all } i \in [n], \, m \in [M].$$

where $P(m_{i,t} = m)$ is the probability that input sample $x_i$ is routed to expert $m$ at iteration $t$ and $\tilde{O}(\sigma_0^{1.5})$ is a negligible value. The equation means all experts have approximately uniform gating weights at the expert exploration stage. That is, the routing selection can be approximated as:

$$\pi_m(s) \approx \frac{1}{M}. \tag{18}$$

This ensures that each expert has an equal opportunity to learn from the data.

During the early stage of training, due to random initialization, each expert $m$ has a weight vector $w_m^{(0)}$ that exhibits a slightly larger inner product with some value basis vector $v_k$. We define the initial preference cluster of expert $m$ as

$$k_m^* = \arg \max_{k \in [K]} \left| \langle w_m^{(0)}, v_k \rangle \right|. \tag{19}$$

Under gradient descent updates, the weight vector $w_m$ of expert $m$ will predominantly grow along the direction of its preferred basis vector $v_{k_m^*}$ .**[Chen et al. (2022), Lemma E.5]** In other words, during training, an expert progressively specializes in modeling a specific cluster $c_k$. Building upon this, we derive an upper bound on the gradient conflicts during the Expert Exploration Stage.

**Theorem E.2** (Upper Bound on Single-Expert and Full-Layer MoE Gradient Conflict with Uniform Sparse Routing). *Consider a Mixture-of-Experts (MoE) layer with $M$ experts and $K$ independent tasks. Each task independently selects one expert uniformly at random: $P = \frac{1}{M}$.*

*Then:*

*1. For any single expert $m$, the expected gradient conflict is upper bounded by*

$$\mathbb{E}[conflict^{(m)}_{t_1,t_2}] \leq G\, q_{single}, \quad q_{single} = 1 - \left(1 - \frac{1}{M}\right)^K - K\frac{1}{M}\left(1 - \frac{1}{M}\right)^{K-1}.$$

*2. For the full-layer MoE with sparse router, the expected gradient conflict is upper bounded by*

$$\mathbb{E}[conflict^{MoE}_{t_1,t_2}] \leq G\, q_{layer}, \quad q_{layer} = \begin{cases} 1 - \dfrac{M!}{(M-K)!\, M^K}, & K \leq M \\ 1, & K > M \end{cases}.$$

*Proof.* Consider an arbitrary expert $m$ and the $K$ tasks. Each task independently selects expert $m$ with probability $1/M$. Let $X$ denote the number of tasks that select expert $m$. Then

$$X \sim \text{Bin}(K, 1/M).$$

A single-expert gradient conflict occurs if and only if $X \geq 2$, and by assumption, the maximum conflict for a single expert is $G$. Therefore, the probability that expert $m$ experiences a conflict is

$$\Pr(X \geq 2) = 1 - \left(1 - \frac{1}{M}\right)^K - K\frac{1}{M}\left(1 - \frac{1}{M}\right)^{K-1}.$$

By linearity of expectation, the expected gradient conflict for a single expert is upper bounded by

$$\mathbb{E}[\text{conflict}^{(m)}_{t_1,t_2}] \leq G\, q_{\text{single}}.$$

The full-layer gradient $G_t$ is constructed by concatenating all expert gradients. A full-layer conflict occurs if at least one expert has a conflict (i.e., at least two tasks select that expert).

- For $K \leq M$, the probability that all $K$ tasks select distinct experts (no collision) is given combinatorially by

$$\frac{M!}{(M-K)!\, M^K},$$

so the probability of at least one collision (full-layer conflict) is

$$q_{\text{layer}} = 1 - \frac{M!}{(M-K)!\, M^K}.$$

- For $K > M$, by the pigeonhole principle, at least one expert must be selected by two or more tasks, so

$$q_{\text{layer}} = 1.$$

Each conflict in an expert can contribute at most $G$ to the full-layer gradient conflict. By worst-case analysis and linearity of expectation, the expected full-layer MoE gradient conflict is therefore upper bounded by

$$\mathbb{E}[\text{conflict}^{\text{MoE}}_{t_1,t_2}] \leq G\, q_{\text{layer}}.$$

- The single-expert bound $G\, q_{\text{single}}$ gives a local (per-expert) expected conflict. - The full-layer bound $G\, q_{\text{layer}}$ accounts for collisions across all experts in the concatenated MoE layer. - Together, they describe the expected gradient conflict behavior of a MoE layer with $K$ tasks and $M$ experts under uniform sparse routing. $\qquad \square$

> **Takeaways: Gradient Conflict in Expert Exploration**
>
> • *In the expert exploration stage, the full MoE layer shows a lower upper bound on gradient conflict, which further decreases as the number of experts $M$ grows relative to tasks $K$.*

**Router Learning Stage**

During the router-learning stage, the sparse MoE layer exhibits two decisive properties that justify its use in a multitask setting.

First, the gating network rapidly identifies the latent task structure: after only

$$T_2 = \lfloor \eta^{-1} M^{-2} \rfloor$$

iterations, its weight vectors $\boldsymbol{\theta}_m$ become strongly aligned with the cluster-center signal $\mathbf{c}_k$ of every task $k$, while simultaneously suppressing spurious correlations with label signals and noise.**[Chen et al. (2022)-lemma E.14]** Consequently, any input drawn from task $k$ is routed to the corresponding subset of experts $\mathcal{M}_k$ with probability $1 - o(\frac{1}{d})$, even though the model is never given explicit task labels. **[Chen et al. (2022)-Lemma E.18]**

Second, each expert in $\mathcal{M}_k$ remains tightly specialised to its assigned task, but only chance-level performance on all other tasks. **[Chen et al. (2022)-Lemma E.12,Lemma 5.2]**.

Taken together, these two properties guarantee automatic task separation and expert specialisation without external supervision, making the MoE layer an effective backbone for multitask problems in which tasks form separable clusters in the input space.

**Theorem E.3** (Expected Gradient Conflict on Task-specific Expert Sets). *Consider a Mixture-of-Experts (MoE) model with $K$ tasks and $M$ experts. Let tasks $i$ and $j$ have expert sets $S_i$ and $S_j$ with sizes $|S_i| = a$, $|S_j| = b$, and intersection $U = |S_i \cap S_j|$. Assume that each task selects an expert from its own expert set with probability $1 - O(1/d)$ uniformly, and from the complement set with probability $O(1/d)$ uniformly. Let $G$ denote the maximum gradient conflict if two tasks select the same expert. Then the expected gradient conflict between tasks $i$ and $j$ satisfies*

$$\mathbb{E}[conflict_{i,j}] \leq G \cdot \left( P^{(1)} + P^{(2)} + P^{(3)} \right), \tag{20}$$

*where*

$$P^{(1)} = U \cdot \left( \frac{1 - O(1/d)}{a} + \frac{O(1/d)}{M - a} \right) \left( \frac{1 - O(1/d)}{b} + \frac{O(1/d)}{M - b} \right), \tag{21}$$

$$P^{(2)} = (a - U + b - U) \cdot \left( \frac{1 - O(1/d)}{a} + \frac{O(1/d)}{M - a} \right) \frac{O(1/d)}{M - b}, \tag{22}$$

$$P^{(3)} = (M - (a + b - U)) \cdot \frac{O(1/d)}{M - a} \cdot \frac{O(1/d)}{M - b}. \tag{23}$$

*In particular, the dominant contribution comes from the shared experts (first class), leading to*

$$\mathbb{E}[conflict_{i,j}] \approx G \cdot \frac{U}{ab} + O(G/d). \tag{24}$$

*Moreover, for the full-layer MoE gradient formed by concatenating all expert parameters, the expected layer-level gradient conflict satisfies the same upper bound:*

$$\mathbb{E}[conflict]_{layer} \leq G \cdot \left( P^{(1)} + P^{(2)} + P^{(3)} \right) \approx G \cdot \frac{U}{ab} + O(G/d). \tag{25}$$

*Proof.* Consider two tasks $i$ and $j$ with expert sets $S_i$ and $S_j$. Partition the set of all experts into three classes: (1) shared experts $S_i \cap S_j$, (2) task-specific experts $S_i \setminus S_j$ and $S_j \setminus S_i$, and (3) unrelated experts outside $S_i \cup S_j$. For each class, we compute the probability that both tasks select the same expert.

For a shared expert $e \in S_i \cap S_j$, task $i$ selects it either from its own set with probability $(1 - O(1/d))/a$ or from outside with probability $O(1/d)/(M - a)$. Similarly for task $j$. Therefore, the probability that both tasks select $e$ is

$$p^{(1)} = \left( \frac{1 - O(1/d)}{a} + \frac{O(1/d)}{M - a} \right) \left( \frac{1 - O(1/d)}{b} + \frac{O(1/d)}{M - b} \right),$$

and there are $U$ such experts, giving total contribution $P^{(1)} = U \cdot p^{(1)}$.

For task-specific experts $e \in (S_i \setminus S_j) \cup (S_j \setminus S_i)$, one task can select it from its own set while the other selects from outside. This yields

$$p^{(2)} = \left( \frac{1 - O(1/d)}{a} + \frac{O(1/d)}{M - a} \right) \frac{O(1/d)}{M - b},$$

with a total of $(a - U + b - U)$ experts, giving $P^{(2)} = (a - U + b - U) \cdot p^{(2)}$.

For unrelated experts $e \notin S_i \cup S_j$, both tasks must select from outside, giving

$$p^{(3)} = \frac{O(1/d)}{M - a} \cdot \frac{O(1/d)}{M - b},$$

and there are $M - (a + b - U)$ such experts, yielding $P^{(3)} = (M - (a + b - U)) \cdot p^{(3)}$.

Summing over all classes and multiplying by $G$, we obtain the expected single-expert gradient conflict:

$$\mathbb{E}[\text{conflict}_{i,j}] \leq G \cdot \left( P^{(1)} + P^{(2)} + P^{(3)} \right).$$

Since the layer-level gradient is formed by concatenating all expert gradients, the layer-level conflict is the sum over all expert contributions, hence

$$\mathbb{E}[\text{conflict}]_{\text{layer}} \leq G \cdot \left( P^{(1)} + P^{(2)} + P^{(3)} \right).$$

Finally, the dominant term is from the shared experts, giving the simplified approximation

$$\mathbb{E}[\text{conflict}]_{\text{layer}} \approx G \cdot \frac{U}{ab} + O(G/d),$$

where $O(G/d)$ absorbs all higher-order small contributions from selecting non-preferred experts. $\square$

---

**Takeaways: Gradient Conflict in Router Learning Stage**

- Gradient conflicts mainly arise from shared experts between tasks, while task-specific experts contribute little.
- The full-layer MoE reduces overall gradient conflict compared to single experts, and conflicts decrease as tasks are routed to more disjoint expert subsets.

---

**Corollary E.4** (Expected Full-layer MoE Gradient Conflict for $K$ Tasks). *Consider a MoE layer with $M$ experts and $K$ tasks. Each task $t$ has its own expert set $S_t$ with size $|S_t| = a_t$. Each expert's gradient norm satisfies $\|g_t^{(m)}\| \leq R$, and if two tasks select the same expert, the maximum gradient conflict is $G$. Each task chooses an expert according to the following rule: with probability $1 - O(1/d)$ it selects uniformly from its own set, and with probability $O(1/d)$ it selects uniformly from the remaining experts.*

*For any subset of tasks $\mathcal{T} \subseteq \{1, 2, \ldots, K\}$, let the number of shared experts be*

$$U_{\mathcal{T}} = \left| \bigcap_{t \in \mathcal{T}} S_t \right|.$$

*Then the expected full-layer gradient conflict is upper bounded by*

$$\mathbb{E}[\text{conflict}]_{\text{layer}} \leq G \sum_{\mathcal{T} \subseteq [K], |\mathcal{T}| \geq 2} U_{\mathcal{T}} \prod_{t \in \mathcal{T}} \left( \frac{1 - O(1/d)}{a_t} + \frac{O(1/d)}{M - a_t} \right), \quad (26)$$

*and its leading term can be approximated as*

$$\mathbb{E}[\text{conflict}]_{\text{layer}} \approx G \sum_{\mathcal{T} \subseteq [K], |\mathcal{T}| \geq 2} \frac{U_{\mathcal{T}}}{\prod_{t \in \mathcal{T}} a_t} + O(G/d). \quad (27)$$

*Proof.* For each task subset $\mathcal{T} \subseteq \{1, 2, \ldots, K\}$ with $|\mathcal{T}| \geq 2$, consider any expert $e \in \bigcap_{t \in \mathcal{T}} S_t$. The probability that all tasks in $\mathcal{T}$ simultaneously select this expert is given by the product of individual selection probabilities. For each task $t \in \mathcal{T}$, the probability of choosing $e$ is

$$\frac{1 - O(1/d)}{a_t} + \frac{O(1/d)}{M - a_t},$$

where the first term accounts for selecting $e$ from the task's own expert set and the second term accounts for selecting $e$ from outside the set. Since the tasks are independent, the joint probability that all tasks in $\mathcal{T}$ select the same shared expert $e$ is

$$\prod_{t \in \mathcal{T}} \left( \frac{1 - O(1/d)}{a_t} + \frac{O(1/d)}{M - a_t} \right).$$

Multiplying by the number of shared experts $U_{\mathcal{T}}$ gives the total expected conflict contribution from this subset:

$$P_{\mathcal{T}} = U_{\mathcal{T}} \prod_{t \in \mathcal{T}} \left( \frac{1 - O(1/d)}{a_t} + \frac{O(1/d)}{M - a_t} \right).$$

Summing over all task subsets with size at least 2, and multiplying by $G$, yields the full-layer expected gradient conflict:

$$\mathbb{E}[\text{conflict}]_{\text{layer}} \leq G \sum_{\mathcal{T} \subseteq [K], |\mathcal{T}| \geq 2} P_{\mathcal{T}}.$$

In the leading order, we can neglect the $O(1/d)$ small-probability contributions from selecting experts outside each task's own set, which gives

$$\mathbb{E}[\text{conflict}]_{\text{layer}} \approx G \sum_{\mathcal{T} \subseteq [K], |\mathcal{T}| \geq 2} \frac{U_{\mathcal{T}}}{\prod_{t \in \mathcal{T}} a_t} + O(G/d),$$

where the dominant contribution comes from shared experts that are simultaneously selected by multiple tasks, and the higher-order terms are absorbed in $O(G/d)$. □

### E.2.4    TASK EMBEDDING FOR IMPROVED TASK SEPARATION IN MULTITASK LEARNING

In multitask learning, tasks often share overlapping input spaces and underlying features (e.g., sentiment classification tasks using the same text encoder), making it difficult for the routing network to distinguish tasks. This overlap leads to routing confusion, where data from one task may be misrouted to experts specialized in other tasks, reducing performance. To address this, *task embedding* introduces a learnable vector $e_\tau$ for each task, which can be combined with input features. Since $e_\tau$ is *orthogonal* across tasks (or constrained via a regularization term), it helps the routing network more clearly distinguish between tasks, even when inputs are similar. Task embedding also acts as a *regularization term*, reducing expert load imbalance and improving the stability and performance of multitask MoE training.

Our theoretical analysis suggests that introducing distinct task embeddings provides crucial clustering signals, enabling the router to more effectively differentiate between tasks. This, in turn, mitigates gradient conflicts and enhances overall performance. However, our empirical results, as presented in Figure 3, show that the inclusion of naive task embeddings did not yield a significant improvement. We hypothesize that this is due to the absence of explicit constraints to maintain separation between the embeddings during optimization, which could lead to their collapse in the later stages of training. For future work, we plan to investigate more effective methods for explicitly injecting task-specific information. The goal is to empower the router to autonomously leverage the unique priors of each task, thereby further advancing multitask learning performance.

## F    DETAILED RELATED WORK

**MCTS with Learned World Models.** Planning within a learned latent space, popularized by MuZero (Schrittwieser et al., 2019), has become a dominant paradigm in reinforcement learning,

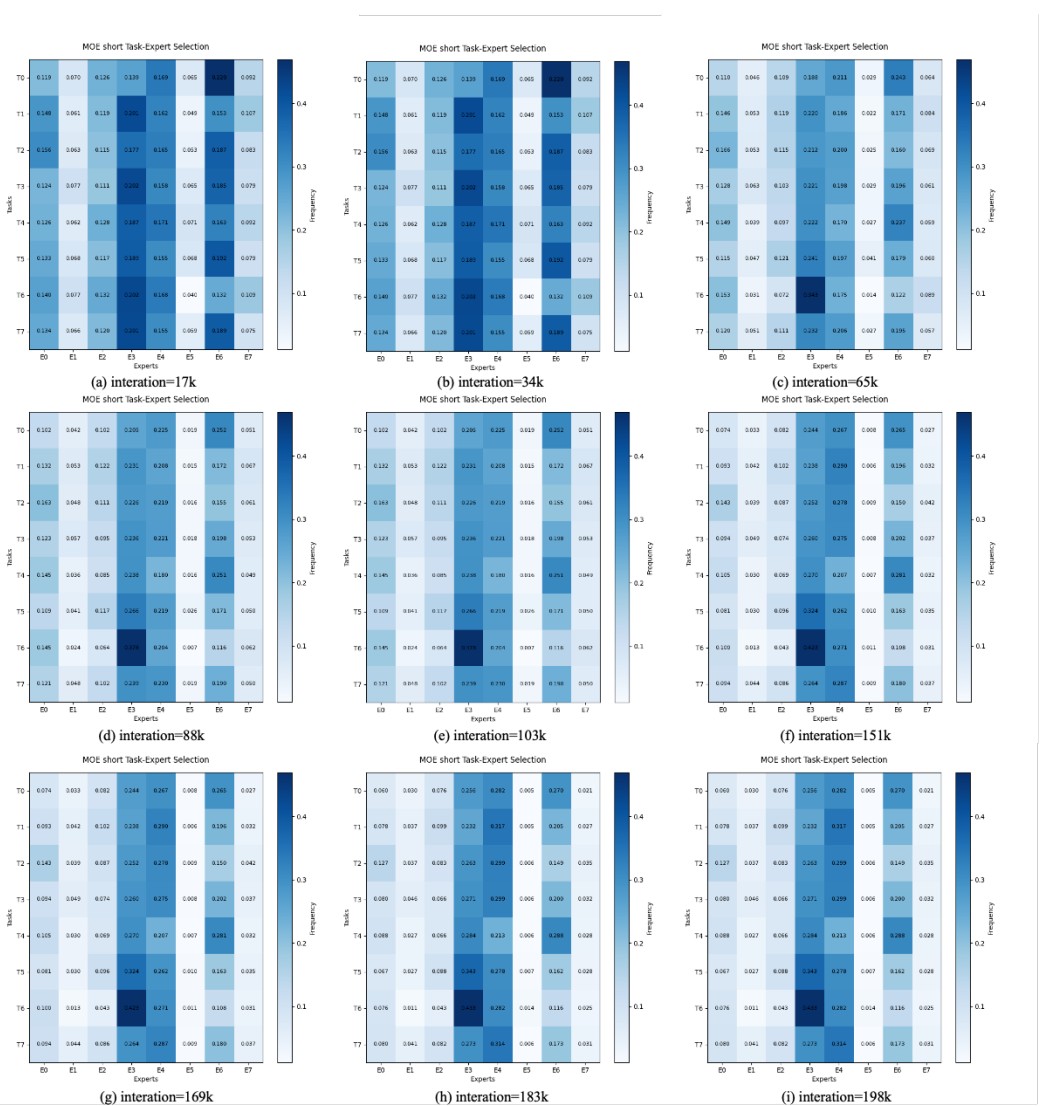

**Figure 18:** Nine heatmaps illustrating the evolution of expert selection distributions in the MoE-based Transformer across different tasks and training interations. Each row corresponds to a specific task, where the color intensity reflects the frequency of expert selections within a sliding window of $S = 1000$. This visualization reveals how task-specific expert utilization patterns emerge and evolve during training.

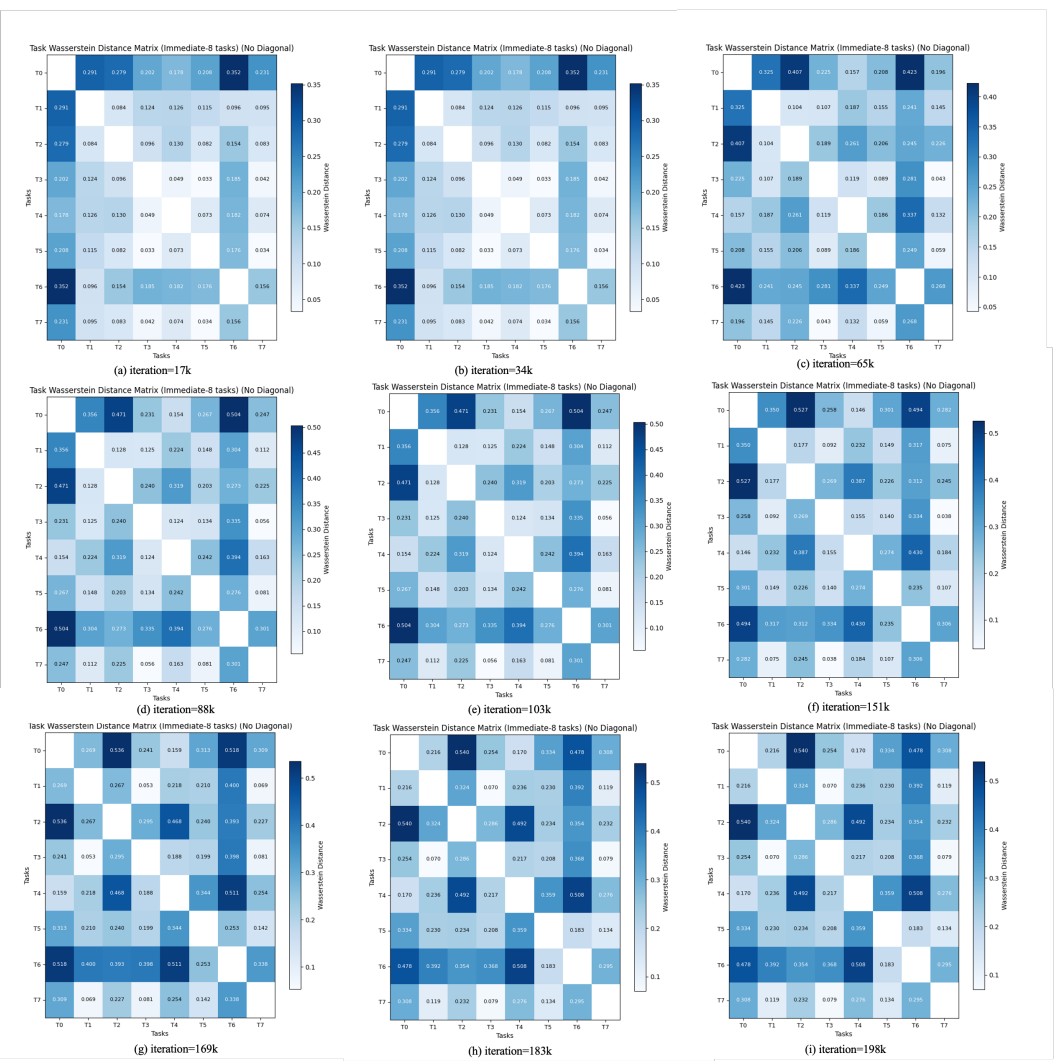

**Figure 19:** Nine heatmaps illustrating the Wasserstein distances between expert selection distributions of different tasks at successive training interations with $S_{immediate} = 100$. Each heatmap corresponds to a specific training interation, revealing how inter-task expert selection similarity changes over time . Diagonal elements are removed to exclude self-comparisons.

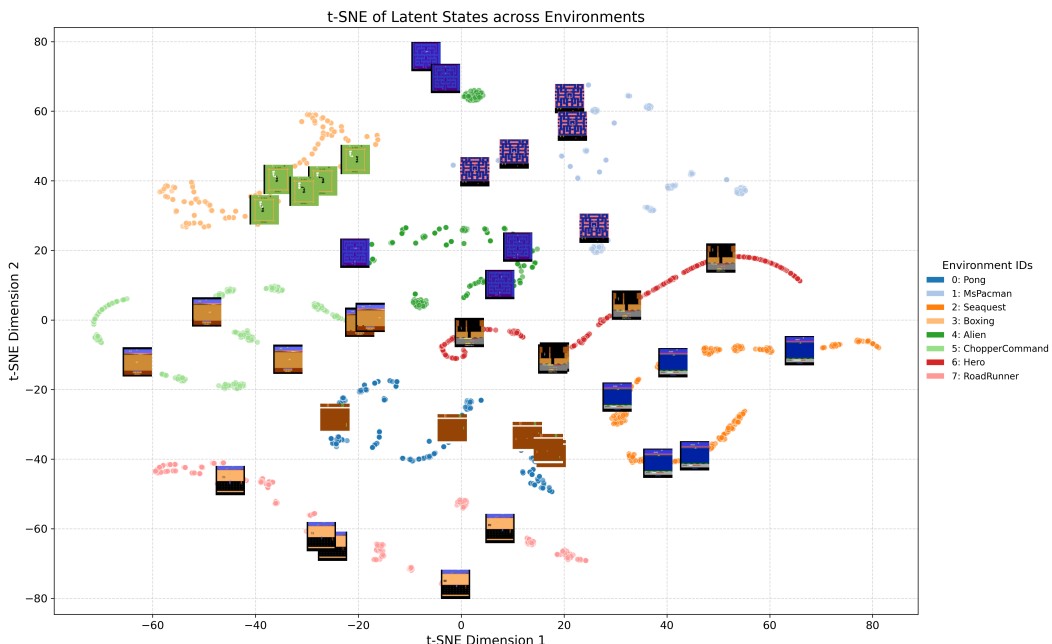

**Figure 20:** t-SNE visualization of the learned latent representations. Each point corresponds to a 2D projection of a 768-dimensional latent embedding generated by the world model. Embeddings are color-coded by environment ID to distinguish between distinct Atari games. The projection illustrates a structured latent space where intra-game states exhibit high cohesion, while distinct environments maintain clear separation. This confirms that the model effectively captures task-specific semantics while preserving discriminative global features. The analysis aggregates 8,000 data points derived from 400 observation sequences, each spanning 20 time steps.

building upon the success of AlphaZero (Silver et al., 2016; 2017). This approach enables a model to predict its own dynamics, rewards, and policies, allowing for powerful lookahead search within a compact, learned representation (Hubert et al., 2021; Antonoglou et al., 2021; Danihelka et al., 2022; Xuan et al., 2024). Recent advancements have incorporated Transformers as the backbone for these world models (Micheli et al., 2022; Robine et al., 2023; Pu et al., 2024; Zhou et al., 2024), significantly enhancing representational capacity and the ability to model long-horizon dependencies. However, the monolithic nature of these architectures, while powerful for single tasks, becomes a critical liability in heterogeneous multitask settings. They are prone to *representational interference* and *plasticity collapse*, where a single shared model struggles to accommodate the diverse and often conflicting properties of multiple tasks. Our work confronts this fundamental architectural bottleneck by systematically investigating and proposing a more robust design.

**Multi-Task Reinforcement Learning (MTRL).** MTRL seeks to improve data efficiency and generalization by sharing knowledge across a distribution of tasks (Vithayathil Varghese & Mahmoud, 2020; D'Eramo et al., 2024). A common strategy is to use a shared backbone with task-specific heads to handle diverse action or observation spaces (Kumar et al., 2022; Hansen et al., 2023; Lee et al., 2022; Gallouédec et al., 2024). To mitigate inter-task interference within the shared backbone, various approaches have been proposed. Architectural methods aim to disentangle knowledge through context-based conditioning (Rakelly et al., 2019; Sodhani et al., 2021), learnable modulation modules that adapt network activations (Schmied et al., 2023), or explicit modularization with task-aware routing and parameter composition (Sun et al., 2022; He et al., 2023). In parallel, optimization-based methods focus on managing gradient conflicts at the training level by projecting gradients to avoid negative interference or re-weighting task losses dynamically (Fernando et al., 2023; Lin et al., 2024; Ma et al., 2023). However, these strategies have been predominantly studied in model-free RL or supervised settings, largely outside the domain of latent space planning, where the additional challenge of disentangling dynamics prediction is paramount. Few works have investigated the unique failure modes of multitask learning in this context. We address this gap by analyzing plasticity-related metrics to identify the *root cause* of performance collapse—an architectural bottleneck in the world model—and propose a novel design to resolve it.

**Sparse and Parameter-Efficient Architectures.** To overcome the limitations of dense, monolithic models, we turn to sparse and parameter-efficient architectures. Sparse models like Mixture-of-Experts (MoE) (Dai et al., 2024) increase model capacity at a near-constant computational cost through conditional computation. This provides a natural architectural prior for multitask specialization, as different tasks can be routed to different "expert" sub-networks (Obando-Ceron et al., 2024). Concurrently, parameter-efficient fine-tuning (PEFT) methods, notably Low-Rank Adaptation (LoRA) (Hu et al., 2022), enable lightweight, task-specific adjustments. By freezing the bulk of the model and training only a small subset of injected parameters, LoRA can efficiently adapt a model to new tasks or data distributions (Wang et al., 2023; Yang et al., 2025; Zhang et al., 2025). However, we identify a paradigm shift required to apply these techniques to online multitask learning. Classic LoRA techniques are predominantly designed for the static fine-tuning of offline or nearly static datasets (Agiza et al., 2024; Huang et al., 2023). In contrast, our approach operates within a non-stationary online data stream. Unlike standard methods that rely on predefined task boundaries, our DPS introduces an in-training decision mechanism to make real-time judgments on when and how to expand model capacity. By allocating independent parameter spaces for distinct distribution shifts, DPS effectively balances plasticity and stability, surpassing the rigidity of standard fine-tuning. Our work bridges this gap by unifying MoE and this DPS mechanism within a *transformer-based world model*, creating a system that is both architecturally specialized and dynamically adaptable for large-scale MTRL.

## G LIMITATIONS AND FUTURE WORK

While our main results demonstrate ScaleZero's robustness across 48 tasks and the efficiency of DPS on DMControl, we identify specific limitations in our current methodology. These limitations directly chart the course for our future research roadmap toward more capable generalist agents.

**Generalizing DPS via Unified Training across Heterogeneous Domains.** A primary limitation of this study is that the DPS strategy has been empirically validated only on the continuous DMC-18 suite. We acknowledge that claims of domain-agnosticism remain speculative without benchmarks on discrete video games (Atari) or language domains (Jericho). However, we posit that DPS is theoretically positioned to generalize beyond DMC due to two structural factors:

- **Structural Decoupling:** DPS operates exclusively on the world model backbone (the latent transition function), mathematically decoupling it from specific I/O modalities. Since differences in observation spaces (e.g., pixels vs. proprioception) and action spaces are encapsulated within the Encoder and Task Heads, DPS is designed to function effectively regardless of the environmental interface.

- **Hypothesis of Higher Utility via Heterogeneity:** We hypothesize that the on-demand allocation mechanism of DPS will yield higher marginal utility in highly heterogeneous domains like Atari (e.g., the reactive simplicity of *Pong* vs. the planning depth of *Seaquest*) compared to DMC. Static architectures often struggle with such internal variance, whereas DPS can adaptively match capacity to the specific complexity of each task.

To validate these hypotheses, our immediate future work will transition from single-domain testing to a *Unified Training* regime involving *Atari-26, DMC-18, and Jericho-4* simultaneously. This will allow us to rigorously analyze whether the adaptive mechanism of DPS can mitigate negative transfer across vastly different state-action spaces, thereby enhancing generalization in a truly multi-modal context.

**Deepening MoE-LoRA Synergy for Architectural Adaptation.** Our current framework employs a modular design where MoE handles architectural specialization and LoRA manages dynamic capacity scaling. However, a limitation of the current implementation is that these components operate somewhat orthogonally. We believe that a deeper synergy is required to fully exploit the architecture's potential. Future work will investigate advanced coordination mechanisms, such as using LoRA to adapt the MoE gating network itself to stabilize routing during rapid distribution shifts, or utilizing MoE routing decisions to dynamically activate specific LoRA adapters. This aims to create a tightly integrated system capable of fine-grained, context-aware architectural adaptation.

**Addressing Sample Efficiency via Hybrid Offline-Online Learning.** Currently, our framework operates in a purely online learning setting. While effective, this approach is inherently limited by the need to collect fresh data for every task, creating a bottleneck for sample efficiency and "cold start" performance. A powerful extension to address this limitation is to integrate large-scale offline pre-training. By initializing the ScaleZero model with weights from a foundation model pre-trained on vast offline datasets, we aim to leverage broad prior knowledge. This hybrid approach is expected to provide superior initialization, faster adaptation to new tasks, and higher peak performance, bridging the gap between specialized reinforcement learning and generalist foundation models.

## H  THE USE OF LARGE LANGUAGE MODELS

During the preparation of this manuscript, the large language models Gemini 2.5 Pro and Gemini 3 Pro were used to assist with language refinement. The primary goal was to enhance the clarity, precision, and readability of the text. The authors have reviewed and approved the final content.

