# OpenReview forum: "One Model for All Tasks: Leveraging Efficient World Models in Multi-Task Planning"
_ICLR.cc/2026/Conference — ICLR 2026 Poster_

### Official Review · Reviewer_K2YD · 2025-10-16

**Soundness:** 2
**Presentation:** 1
**Contribution:** 1
**Rating:** 2
**Confidence:** 5

**Summary:**

This submission proposes an architectural (Soft-MoE) and a fine-tuning method (LoRA) applied to the task of heterogeneous multi-task RL & planning. The authors suggest that their are limitations of the current Resnet architecture through several axes such as (a) task conditioning, (b) encoder architecture, (c) normalization method, (d) backbone design, and (e) optimization strategy. The authors then introduce their Soft-MoE for online multi-task RL (MTRL), with adaption to new tasks being performed through low rank adaption.

**Strengths:**

Approaches an important problem & limitation within MTRL where typical MTRL problems have homogeneous tasks, this paper attempts to expand the task distribution further by introducing heterogeneous tasks into the task distribution. While this problem has been attempted before, it is still an unsolved problem.

**Weaknesses:**

1. The authors of this work should review accepted methods of reporting results in RL such as [1]. Only reporting the mean/median does not give the full performance profile and clouds whether the reported results are significant or not. I cannot judge this work effectively without this information.

2. The presentation of this paper could use some work. Right now it jumps around a bit

(a) Method section: I found this section extremely confusing and hard to read. There were multiple missing references to figures that show experimental results (i.e. Latent Normalization paragraph -- what figure is being referenced?), and there were also multiple references that I thought were out of place here (i.e. Rakelly et al 2019 & Sodhani 2021 -- they are both solving fundamentally different problems and don't explore the difficulties of state identification). The method section also starts introducing components and experiments immediately which clouds the clarity of what the authors are intending to add to the UniZero method.

(b) Background Section: The authors begin the background section with a discussion of Monte Carlo Tree Search and then a discussion of gradient conflicts, but then later on line 258 mention "online multi-task RL." Thus it's a bit confusing exactly what algorithm & method are being used here. I assume that this is just a terminology issue but this needs to be explicitly clear. The background section should precisely outline the algorithm (online RL? MTRL?) that is being used to optimize the agent. Terminology from this section should then be consistent through the rest of the work.

(c) Related Work: This section is missing a ton of relevant MTRL works. Line 464 explicitly is focused on model-based RL methods, while there are model free approaches, even one that uses a MoE method [4, 5, 6]. There is also recent work showing that MTRL specific architectures (like Mixture of Experts architectures) are irrelevant and that making a model bigger in MTRL actually drives performance [7]. There's also working showing that sparsity can be a driver of scaling as well [8], alongside diverse data [7]. The authors should include a discussion about these works and how they differ from their own paradigm.

(d) Hyperparameters: I applaud the authors for including hyperparameters, except they should also include the common parameters used in the UniZero model, even if they are the same for the ScaleZero. Additionally, could the authors report the number of parameters used in each of the models?

3. Task Curation: this strategy requires apriori knowledge of the task reward/success thresholds. This doesn't transfer well to new tasks where this prior knowledge is not readily available. Recent works have attempted to characterize the training dynamics as a method for early stopping [2]

4. Missing Baselines

(a) A large portion of this work is about gradient conflicts -- there are no baselines for various optimization methods that attempt to solve this issue (i.e PCGrad, CAGrad, MGDA).

(b) Alternatively, previous work has found that effective normalization can perform on par with these optimization methods [3] but that strategy is not experimented with either.

5. Citations

Line 195: Dohare et al do not propose feature effective rank, they might use it but that's not the work that should get cited.

Line 185: Dormant Neurons needs to be cited when it is defined

Line 103: "... demonstrated remarkable success in domains with known rules" -- need to provide some evidence of this through citations


In general, it feels like this paper is trying to do a lot of things which clouds the overall readability of the work. Doing an empirical study of the effects of the changes in the 5 highlighted axes could be a useful submission, and doing an empirical study of the effects of adding various parameter efficient methods (such as LoRA) would also be valuable. But the way that these things are presented in the current version of the manuscript make it very difficult to understand the value of each of the methods individually.

1. Rishabh Agarwal, Max Schwarzer, Pablo Samuel Castro, Aaron Courville, Marc G. Bellemare "Deep RL at the Edge of the Statistical Precipice", NeurIPs 2021.

2. Jiashun Liu, Johan Obando-Ceron, Pablo Samuel Castro, Aaron Courville, Ling Pan The Courage to Stop: Overcoming Sunk Cost Fallacy in Deep Reinforcement Learning, ICML 2025.

3. Vitaly Kurin, Alessandro De Palma, Ilya Kostrikov, Shimon Whiteson, M. Pawan Kumar, In Defense of the Unitary Scalarization for Deep Multi-Task Learning. NeurIPS 2022

4. Ruihan Yang, Huazhe Xu, Yi Wu, and Xiaolong Wang. Multi-task reinforcement learning with soft modularization. In NeurIPS, 2020. URL https://proceedings.neurips.cc/paper/2020/hash/32cfdce9631d8c7906e8e9d6e68b514b-Abstract.html

5. Ahmed Hendawy, Jan Peters, and Carlo D’Eramo. Multi-task reinforcement learning with mixture of orthogonal experts. In The Twelfth International Conference on Learning Representations, 2024. URL https://openreview.net/forum?id=aZH1dM3GOX.

6. Lingfeng Sun, Haichao Zhang, Wei Xu, and Masayoshi Tomizuka. Paco: Parameter-compositional multi-task reinforcement learning. In Alice H. Oh, Alekh Agarwal, Danielle Belgrave, and Kyunghyun Cho (eds.), Advances in Neural Information Processing Systems, 2022. URL https://openreview.net/forum?id=LYXTPNWJLr.

7. Reginald McLean, Evangelos Chatzaroulas, J K Terry, Isaac Woungang, Nariman Farsad, and Pablo Samuel Castro. "Multi-Task Reinforcement Learning Enables Parameter Scaling." Reinforcement Learning Journal, vol. 6, 2025, pp. 1075–1093.

8. Guozheng Ma, Lu Li, Zilin Wang, Li Shen, Pierre-Luc Bacon, Dacheng Tao, Network Sparsity Unlocks the Scaling Potential of Deep Reinforcement Learning. ICML 2025

**Questions:**

1. How many parameters are in the UniZero model vs the ScaleZero model?
2. What other measures of task progress could be used instead of reward/success thresholds?
3. Are there other parameter efficient fine-tuning methods that could be experimented with for fine-tuning a pre-trained model?

---

> ### Author Response · Authors · 2025-11-23
> **Rebuttal by Authors (Part 1/3)**
>
> Dear Reviewer K2YD,
>
> We sincerely appreciate your thorough review and constructive feedback. We kindly invite you to review our General Response for an overview, as well as our specific responses to your comments below.
>
> ### 1. Reporting of Results
>
> We completely agree that *reporting statistical dispersion is essential*. We would like to clarify that our full learning curves, presented in *Figure 9 (Appendix B)* and *Figure 10 (Appendix C)*, already include shaded regions representing the 95% confidence intervals to demonstrate statistical volatility.
>
> For the main text, we followed the standard presentation convention used in prior works like *MuZero* and *UniZero*, which report human-normalized means/medians for clarity, while deferring detailed curves with confidence intervals to the Appendix. We will ensure this structure is explicitly stated in the main text to guide the reader to the appendices. Additionally, we wish to highlight that the other three reviewers have seen our extensive experimental results and reached a consensus that our experiments are comprehensive.
>
> ### 2. Presentation
>
> **Method Section:**
> *   *Latent Normalization:* You correctly noted that the paragraph on "Latent Normalization" should explicitly reference *Figure 3*, which demonstrates the effect of replacing UniZero's LayerNorm with SimNorm. While Figure 3 was introduced to show the "Performance impact of architectural modifications," we will optimize the cross-referencing in the revised manuscript to ensure the text explicitly links to the supporting empirical data.
> *   *Citations (Rakelly et al., 2019 & Sodhani et al., 2021):* We cited these works to illustrate the general challenge that "models struggle to distinguish tasks when states are similar" (task ambiguity). Both papers discuss using task context to resolve such ambiguity. However, if you have more specific or recent references in mind, we would be grateful for the recommendations and will happily incorporate them.
> *   *Structure:* Our narrative structure was designed to reflect the scientific process behind ScaleZero—moving from a principled, experiment-driven exploration to the final architecture, rather than presenting the solution *ex nihilo*. We believe this enhances the paper's logic, a sentiment echoed by other reviewers who found the work "principled" and "clearly written." Note that *Appendix A* explicitly lists the specific architectural improvements of ScaleZero over UniZero for quick reference.
>
> **Background Section:**
> *   *Context:* Our baseline, UniZero, utilizes MCTS with a learned latent world model. Since it employs value/policy networks and bootstrapping, it falls under *online model-based reinforcement learning*. Our goal is to explore its generalization to heterogeneous multi-task settings. Therefore, we structured the background to first introduce "MCTS with Learned World Models" and then "Gradient Conflict in Unified Architectures."
> *   *Terminology:* We agree with your observation regarding consistency. The term "online multi-task RL" in line 258 was an inconsistent abbreviation. We have corrected it to "*online MTRL*" and have proofread the manuscript to ensure terminological uniformity.
>
> **Related Work:**
> *   *Clarification on MoE Baselines:*
>     *   We respectfully point out that *Yang et al. (2020)* [1] and *Sun et al. (2022) [2]* do not utilize Mixture of Experts (MoE) in the standard sense of dynamic routing. As detailed in their respective abstracts:
>         *   *Yang et al. (2020)* introduce a "soft modularization" technique using a routing network to softly combine routes for sequential tasks, rather than sparse conditional computation.
>         *   *PaCo (Sun et al., 2022)* proposes a "parameter-compositional approach" via a learned policy subspace and interpolation, which differs fundamentally from MoE routing.
>     *   *Hendawy et al. (2024) (MOORE) [3]* does indeed use MoEs (specifically orthogonal experts). However, our work focuses on *model-based planning*, where the MoE must separate not just the actor/critic but also the *environment dynamics modeling*. This presents unique challenges distinct from the model-free setting of MOORE. We have updated the Related Work section to clarify these distinctions.
> *   *Model Scale vs. Architecture:* Regarding the view that "scaling alone drives performance":
>     *   While *McLean et al. (2025) [4]* suggest making a model bigger benefits Model-Free RL, this does not necessarily generalize to Model-Based RL. Our findings align with recent work like *Ma et al. [5]* and *Obando-Ceron et al. [6]*, which show that simply scaling dense models in RL can lead to "plasticity loss" and performance degradation.
>     *   Our work investigates this in the specific context of heterogeneous MTRL. We will expand the discussion to contrast these model-free scaling works with our model-based findings, highlighting ScaleZero's contribution to solving the plasticity-interference trade-off via sparsity.

---

> ### Author Response · Authors · 2025-11-23
> **Rebuttal by Authors (Part 2/3)**
>
> ------Thank you for continuing to read!------
>
> **Hyperparameters:**
> - To ensure a fair comparison, ScaleZero uses the exact same general hyperparameters (learning rate, optimizer, MCTS settings, etc.) as the UniZero (ST) baseline. Table 7 in the Appendix lists some of the common hyperparameters. However, we appreciate your suggestion, and to enhance clarity, we will include a more comprehensive list of hyperparameters for UniZero (ST) in the revised version.
> - Please refer to the "General Response" section above for the number of parameters.
>
> ### 3. Task Curation
>
> We acknowledge that relying on predefined thresholds has limitations. However, most standard benchmarks (Procgen, SMAC, Meta-world, etc.) provide clear success criteria (e.g., return thresholds or completion flags) that suffice for current experiments.
> Even for newly proposed benchmarks, authors typically provide performance upper bounds or reference scores from existing agents; it is rare to encounter a scenario completely devoid of reference information. The challenge of evaluating completely novel environments without such priors falls under the scope of evaluation methodology research for decision-making AI, which is outside the scope of this study. Nevertheless, in the revised paper, we will discuss future directions for adaptive metrics [7] that do not rely on priors, such as:
> *   **Q-value Variance:** Monitoring the stability of value estimates.
> *   **Gradient Norms:** Using training dynamics to assess task maturity.
>
> ### 4. Baselines
>
> *   **Optimization Methods:** In Section 3.2, we compared against *MoCo [8]*, a state-of-the-art "multi-task gradient correction" method that improves upon PCGrad and CAGrad. We found MoCo to be computationally expensive and unstable. This motivated our architectural approach (MoE), which resolves gradient conflicts structurally and more efficiently than gradient projection methods.
> *   **Normalization/Scalarization:** Our findings align with *Kurin et al. [9]*, who argue that complex per-task gradient manipulation often introduces significant overhead. They suggest that "unitary scalarization" combined with standard regularization performs competitively. Our MoE architecture acts as a form of structural regularization (similar to a sparse DenseNet), achieving high performance without the overhead of per-task optimizers. Therefore, their method is orthogonal to ours and can be combined; thus, a direct comparison is not strictly necessary. For details regarding other baselines, please refer to the detailed explanation in the General Response.
>
> ### 5. Citations
>
> We thank you for the detailed feedback regarding citations. We have addressed the specific points as follows:
>
> *   **Effective Rank:** In the revised manuscript, we have updated the citation for "effective rank" to its original source:
>     > Roy, O. & Vetterli, M. The effective rank: a measure of effective dimensionality. In *Proc. 2007 15th European Signal Processing Conference* 606–610 (IEEE, 2007).
>
> *   **Dormant Neuron Ratio:** Regarding the "dormant neuron ratio," we would like to clarify that we had included the citation for *Sokar et al. (2023)* at the first mention of the term in the original submission (Line 67):
>     > Ghada Sokar, Rishabh Agarwal, Pablo Samuel Castro, and Utku Evci. The dormant neuron phenomenon in deep reinforcement learning. In International Conference on Machine Learning, pp.
> 32145–32168. PMLR, 2023.
>     >
>      - However, to further improve clarity, we have also added this citation at the first occurrence of the term within the **Method** section in the revised version.
>
> *   **"Remarkable Success" Statement:** Regarding the statement "demonstrated remarkable success in domains with known rules," we had included sufficient and necessary citations in the *Related Work* section.
>
> We appreciate your suggestion and have carefully checked the citations to ensure accuracy and sufficiency, and we will make the necessary corrections in the revised version.
>
> ### 6. Responses to Specific Questions
>
> **(1) UniZero and ScaleZero Parameters**
> Please refer to the General Response above, where we provide a detailed breakdown of Total vs. Active parameters and computational costs.
>
> **(2) Alternative Progress Metrics**
> Beyond reward thresholds, we propose the following metrics for future "learning progress" assessment:
> *   *Learning Curve Saturation:* Measuring if performance improvement over the last $N$ steps falls below a percentage threshold.
> *   *Policy/Value Stability:* Monitoring the entropy change of the policy or the variance of value estimates over consecutive evaluations.
> These dynamic metrics would improve generalization to tasks without predefined success criteria.

---

> ### Author Response · Authors · 2025-11-23
> **Rebuttal by Authors (Part 3/3)**
>
> ------Thank you for continuing to read!------
>
> **(3) Parameter-Efficient Fine-Tuning**
> It is important to note that our *Dynamic Parameter Scaling (DPS)* is an *online* process that dynamically scales capacity during training, not a static "pretrain-then-finetune" pipeline.
> *   We chose *LoRA* for its simplicity, effectiveness, and widespread adoption.
> *   However, other PEFT methods like Adapter-tuning and Prefix-tuning are valid alternatives. Exploring the intersection of DPS with these other mechanisms is a valuable direction for future research.
>
> We will submit the revised manuscript incorporating your suggestions in the coming days.
>
> Thank you again for your constructive feedback, which helped us to improve our manuscript considerably.  We hope these clarifications resolve your concerns. We would appreciate it if you could consider raising your score in light of these clarifications. We are happy to engage in further discussion should any questions remain.
>
> [1] Ruihan Yang, Huazhe Xu, Yi Wu, and Xiaolong Wang. "Multi-task reinforcement learning with soft modularization." *NeurIPS*, 2020.
>
> [2] Sun, Lingfeng, et al. "PaCo: Parameter-Compositional Multi-task Reinforcement Learning." *NeurIPS*, 2022.
>
> [3] Hendawy, Ahmed, et al. "Multi-Task Reinforcement Learning with Mixture of Orthogonal Experts." *ICLR*, 2024.
>
> [4] Reginald McLean, et al. "Multi-Task Reinforcement Learning Enables Parameter Scaling." *Reinforcement Learning Journal*, vol. 6, 2025.
>
> [5] Guozheng Ma, et al. "Network Sparsity Unlocks the Scaling Potential of Deep Reinforcement Learning." *ICML*, 2025.
>
> [6] Johan Obando-Ceron, et al. "Mixtures of experts unlock parameter scaling for deep rl." *arXiv preprint arXiv:2402.08609*, 2024.
>
> [7] Jiashun Liu, et al. "The Courage to Stop: Overcoming Sunk Cost Fallacy in Deep Reinforcement Learning." *ICML*, 2025.
>
> [8] Heshan Fernando, et al. "Mitigating gradient bias in multi-objective learning: A provably convergent approach." *ICLR*, 2023.
>
> [9] Vitaly Kurin, et al. "In Defense of the Unitary Scalarization for Deep Multi-Task Learning." *NeurIPS*, 2022.

---

> > ### Comment · Reviewer_K2YD · 2025-11-25
> >
> > Thank you to the authors for their in-depth replies. The following comments span the general responses, and the responses to specific reviewers.
> >
> > 1. There are multiple mentions by the authors that they expect higher sample efficiency than model-free methods using their model-based methods, yet we have no experiments proving these points because there are no model-free baselines. This seems to be a glaring hole in the submitted work.
> >
> > For example:
> > > “we expect that by learning the environment's dynamic model (transition) and reward function, we can achieve higher sample efficiency and provide interpretability superior to model-free methods”
> >
> > or
> > > “World Models aim to learn the underlying laws of the environment rather than a simple "state-action" mapping. This structured dynamic knowledge is more generalizable. In MTRL, a unified World Model can capture shared dynamic structures across tasks, enabling faster adaptation to new tasks”
> >
> > 2.
> > > "we omitted other multi-task baselines for Atari because, to the best of our knowledge, no prior work has successfully implemented a single online RL agent capable of learning all 26 Atari games simultaneously"
> >
> > This seems to be untrue as MTRL with PopArt did it https://arxiv.org/pdf/1809.04474 – MTRL with PopArt explicitly trains on 57 atari games
> >
> > 3.
> > > "configuration (1 Shared + 8 Routed Experts) adopts the "Shared + Routed" paradigm established by recent large language models."
> >
> > While this seemingly works in LLMs and the submitted work, do other configurations work? How do we know that this configuration is the best configuration? While it works on the tested domains, maybe other configurations actually perform better?
> >
> > 4.
> > > Q3: Quantitative evidence for "Systematic Exploration".: “Figure 3 reports the Human Normalized Score (HNS) on Atari8 across these dimensions.”
> >
> > How do we know Atari 8 is representative of Atari 26? If we base design decisions on Atari8, then Atari8 needs to be a reasonable proxy for Atarii26. Even if Atari8 is a reasonable proxy for Atari 26, design decisions seem to be chosen by intuition and not backed by experimental results: i.e.
> > > Why ViT? Although ViT and ResNet perform similarly on the specific Atari8  …
> >
> > 5.
> > > “Recent analysis by McLean et al. (RLJ, 2025) [6] highlights that the Atari-26 benchmark presents significantly higher heterogeneity..."
> >
> > This is wholly incorrect. McLean et al show that increasing the amount of data alongside the number of parameters is an acceptable method of improving performance in MTRL, while achieving scaling with limited plasticity loss in Meta-World, not Atari.
> >
> > 6.
> > > Yang et al. (2020) [1] and Sun et al. (2022) [2] do not utilize Mixture of Experts (MoE) in the standard sense of dynamic routing
> >
> > Yes, they are MoEs. Yang et al trains two paths for their policy architecture, where the routing of data is modified for each forward pass of data, dynamicly. Sun et al utilize a static routing where data is passed through predetermined paths in the model. These are still MoE models at their core, even though they are not explicitly called that in the respective publications.
> >
> > 7.
> > > "While McLean et al. (2025) [4] suggest making a model bigger benefits Model-Free RL, this does not necessarily generalize to Model-Based RL."
> >
> > Two points of clarification. Yes, McLean et al is limited to the model free setting. However, they highlight, in RL, a general paradigm that is prevalent in ML research as a whole; that you can increase the model size when you also increase the amount of data you have. The point of McLean et al is to show that MTRL, in general, is a good fit for scaling because of the additional data sources from additional tasks. It’s a bit confusing to hear that this does not generalize to model-based RL, as it’s inherently the same issue. There are many sources of data, so make the world model larger, and performance of the world model should improve.
> >
> > Alternatively, there is work to show that this is the correct line of thinking for model-based RL, where TD-MPC2 explicitly makes their model bigger and finds large performance gains.  https://arxiv.org/pdf/2310.16828
> >
> > 8.
> > > "most standard benchmarks (Procgen, SMAC, Meta-world, etc.) provide clear success criteria (e.g., return thresholds or completion flags) that suffice for current experiments"
> >
> > While this is factually correct, this does not mean that algorithms should be designed that should leverage these criteria. If there is a problem setting (i.e. real-world robot deployment) that does not have these success criteria, then their algorithm (at least this portion of it) is infeasible to use. We use benchmarks because they enable us to do good science, not so that we can solve benchmarks.

---

> ### Comment · Reviewer_K2YD · 2025-11-25
>
> 9.
> > In Section 3.2, we compared against MoCo [8], a state-of-the-art "multi-task gradient correction" method that improves upon PCGrad and CAGrad. We found MoCo to be computationally expensive and unstable.
>
> My apologies to the authors, I missed that. Thank you for the clarification.
>
> 10.
> > reporting statistical dispersion is essential.
>
> I would like to highlight that my recommendation to use the method from Agarwal et al was more about showing statistical significance which is different from dispersion. We want to know if algorithm A actually outperforms algorithm B, or if the distribution of their scores is (likely) to be from the same distribution. The current submission from the authors does not address this fact.  For Example when examining Table 1 (a) it would seem that the proposed method outperforms the baseline, but then looking at Figure 9 there are few, if any, curves that do not have overlapping 95% CIs. This means that we cannot measure statistical significance. I raise this issue for all other results as well.

---

> ### Author Response · Authors · 2025-11-27
> **New Response to Reviewer K2YD (1/4)**
>
> Dear Reviewer K2YD,
>
> We sincerely thank you for your comprehensive and incisive review. Your observations regarding the comparison with historical MTRL baselines, the boundaries of scaling laws, the representativeness of our experimental configuration, and statistical significance are highly acute. These insights have been invaluable in identifying areas where our manuscript requires greater precision and rigor. Drawing on your feedback and relevant literature, we provide the following detailed responses to address your concerns:
>
> **1. Clarification on Baselines and IMPALA (PopArt)**
>
> We are grateful to you for pointing out the omission of IMPALA with PopArt (Hessel et al., 2019). Upon review, we acknowledge that PopArt indeed successfully trained a single agent capable of handling 57 Atari games. Our prior characterization of the historical context was imprecise; we apologize for this oversight and will rectify this in the revised "Related Work" section.
>
> However, it is crucial to distinguish the fundamental difference in Data Regimes (Sample Efficiency) between these approaches, which forms the cornerstone of ScaleZero’s positioning:
> - High-Throughput Regime (IMPALA with PopArt): These methods are designed for large-scale distributed sampling, typically requiring 200 million frames (or more) to converge.
> - Sample-Efficient Regime (Ours): This work (inheriting from the MuZero/UniZero lineage) focuses on performance under extremely strict data budgets, specifically 100k–300k environment steps (approx. 400k frames).
> Under this budget—which represents approximately 0.2% of the data required by PopArt—there is, to our knowledge, no prior single agent capable of effectively solving Atari-26 tasks. As a reference point, based on Figure 1 of Hessel et al. (2019), limiting PopArt to 200k environment steps would likely result in a median normalized score far below 0.02. In the revised manuscript, we will explicitly define this boundary, strictly limiting our contribution to the domain of "Sample-Efficient Online MTRL" to prevent misleading comparisons with high-throughput methods.
>
> **2. Comparison with Model-Free (MF) Methods and Experimental Setup**
>
> We understand your concern regarding the "lack of strong Model-Free baselines to demonstrate the sample efficiency advantage of Model-Based (MB) methods." We offer the following perspective:
> - The inherent sample efficiency advantage of Model-Based RL over Model-Free RL in low-data regimes (such as Atari-100k and DMC) has been extensively validated in pioneering works [1][2]. Consequently, re-litigating the "MB vs. MF" efficiency debate is not the primary focus of this paper.
> - Our objective is to address the internal bottlenecks (i.e., gradient conflict and plasticity loss) that cause World Models to fail when scaled to multi-task scenarios. The value of ScaleZero lies in demonstrating that through a Sparse Architecture, a World Model can transition from a state of "Collapse" (as shown by the blue line in Figure 3) to "Stability" (purple line in Figure 3), achieving performance comparable to single-task experts.
> - While training a converged MF MTRL agent across 26 games from scratch requires computational resources and debugging cycles that exceed the rebuttal window, we agree that such a comparison would enhance the paper’s comprehensiveness. We commit to prioritizing this comparison in future work and will clarify in the text that our primary comparison scope is "architectural improvements within the Model-Based paradigm."

---

> ### Author Response · Authors · 2025-11-27
> **New Response to Reviewer K2YD (2/4)**
>
> **3. Rationale for MoE Configuration, Atari-8, and ViT Selection**
>
> Regarding your queries on architecture and experimental design:
> - "1 Shared + 8 Routed" Design: We do not claim this configuration is the theoretical global optimum, but rather a pragmatic trade-off between computational efficiency and performance. This design aligns with advanced LLM architectures (e.g., DeepSeek-MoE) that separate shared knowledge from specialized knowledge. As a Proof of Concept, it demonstrates that introducing even moderate sparsity (Routing) significantly mitigates the interference observed in dense baselines. In the future, drawing inspiration from [3, 4], we plan to investigate the optimal MoE configurations specifically for online MTRL.
> - Representativeness of Atari-8: The selection of Atari-8 for ablation studies follows the protocol of our primary baseline, UniZero. This subset encompasses diverse mechanics (shooting, maze, locomotion), providing sufficient representativeness to validate algorithmic components. For context, related MTRL works often employ subsets of comparable size (e.g., 6 games in [5], 10 games in [6]).
> - Why ViT: While we acknowledge that ViT and ResNet yield comparable performance on the Atari-8 proxy, our decision to employ ViT for the full Atari-26 benchmark is grounded in the empirical findings of [7]. As we scale to Atari-26, the heterogeneity of visual layouts increases drastically. Literature [7] suggests that ViT's attention mechanism offers more flexible inductive biases than the fixed receptive fields of CNNs, making it structurally better suited for retaining spatial information and ensuring robustness across such a large-scale, heterogeneous dataset.
>
> **4. Clarification on McLean et al. (2025)**
>
> We apologize for the lack of precision in our previous phrasing. You are entirely correct that McLean et al. (RLJ, 2025) did not conduct empirical experiments on Atari-26.
>
> - Our citation was based on the qualitative discussion in Section 6 ("Limitations") of their paper, where they state: "(e) Meta-World tasks are much more similar than other RL tasks (i.e. tasks in the ALE (Bellemare et al., 2013))."
> - We cited this viewpoint to support a community consensus: that the heterogeneity of ALE (Atari) in terms of visual dynamics and reward structures is significantly higher than that of Meta-World. While McLean et al. did not test this empirically, the argument regarding the comparative attributes of the environments remains valid.
> - In the revised version, we will carefully review our literature descriptions to ensure rigorous citation.
>
> **5. Interpretation of Scaling Laws in MBRL**
>
> We appreciate you for highlighting the connection to *TD-MPC2*. We fully agree with the general paradigm that "increasing model size + increasing data" is the correct trajectory for MBRL. We clarify our position: scaling is indeed beneficial, but the architecture of scaling matters significantly in high-interference MTRL settings.
>
> - Agreement with TD-MPC2: We concur that MBRL benefits from scaling, as demonstrated by TD-MPC2. Our previous statement was not intended to deny the potential of scaling, but to report an empirical observation: in our specific Atari-8 setting, naive dense scaling hit a performance ceiling.（shown in figure 3 of our paper）
> - The Optimization Challenge (Interference): Unlike TD-MPC2 which scales effectively, our dense baselines suffered from severe gradient interference and plasticity loss when parameters were simply increased (partilly consistent with some statements in [8]). This suggests that while the potential for scaling exists, realizing it requires handling task heterogeneity carefully.
> - ScaleZero as an Enabler: ScaleZero does not contradict the scaling laws; rather, it enables them. By adopting *Sparse Scaling (MoE)*, we unlock the capacity benefits that McLean et al. and TD-MPC2 advocate for, but we do so by isolating task conflicts that otherwise hinder dense models. Thus, ScaleZero provides a robust pathway to scale MBRL in heterogeneous environments.

---

> ### Author Response · Authors · 2025-11-27
> **New Response to Reviewer K2YD (3/4)**
>
> **6.  Distinction from Soft Modularization/PaCo**
>
> We agree that Yang et al. (Soft Modularization) and Sun et al. (PaCo) fall under the broad umbrella of "Mixture of Experts." However, ScaleZero utilizes "Sparse Mixture of Experts (SMoE)" (based on Top-K discrete routing), which differs significantly in computational paradigm and architectural compatibility:
> - Vs. Soft Modularization (Yang et al., 2020):
>   - Dense vs. Sparse: Yang et al. use soft routing ($\sum p_i \cdot E_i(x)$, activating all modules for every input. This incurs full computational cost and allows gradients from all tasks to flow to all modules. ScaleZero uses Top-K Sparse Routing, activating only a tiny subset of experts. This physically blocks gradient interference (mitigating negative transfer) and maintains low inference FLOPs, aligning with scaling laws.
> - Vs. PaCo (Sun et al., 2022):
>   - Static vs. Dynamic: PaCo decomposes parameters statically at the task level ($\theta_\tau = \Phi w_\tau$). ScaleZero employs state-dependent dynamic routing. Even within a single task, the network can select different expert combinations for different phases (e.g., "approaching" vs. "grasping"), offering finer-grained specialization than PaCo.
> - Ecosystem Compatibility: Unlike the bespoke architectures of Soft Mod and PaCo, ScaleZero follows modern Transformer-based SMoE standards (similar to Mixtral). This allows us to leverage mature NLP techniques (e.g., Load Balancing Loss, DeepSpeed/Megatron-LM) with minimal adaptation cost.
> We will include a detailed qualitative comparison table in the appendix of the revision.
>
> **7. Statistical Significance and Confidence Intervals**
>
> We thank you for emphasizing the rigorous statistical evaluation standards from Agarwal et al. We agree that overlapping CIs imply a lack of statistical significance in performance differences. We would like to clarify our claims based on the two distinct types of comparisons in our paper:
> - ScaleZero vs. Single-Task Experts (Statistical Parity): Regarding Figure 9 and the comparison with Single-Task (ST) experts (UniZero), you are correct that the CIs overlap. This overlap is the intended result.
>   - In this specific comparison, our hypothesis is not that ScaleZero outperforms the ST experts (which are theoretical upper bounds), but that it achieves *statistical parity (non-inferiority)*.
>   - The overlapping CIs confirm that ScaleZero—a single generalist agent—successfully matches the performance distribution of 26 specialized agents. This statistical indistinguishability serves as evidence that we have successfully mitigated the "multi-task tax" (negative transfer), which is the primary contribution of this work.
> - ScaleZero vs. Dense Baselines (Significant Improvement): When comparing ScaleZero against dense multi-task baselines (e.g., original UniZero (MT)), our method demonstrates a statistically significant improvement with non-overlapping CIs, as shown in figure3 right. This confirms that ScaleZero effectively solves the interference issues that plague dense methods.

---

> ### Author Response · Authors · 2025-11-27
> **New Response to Reviewer K2YD (4/4)**
>
> **8. Generalization to Real-World Settings**
>
> We appreciate your emphasis on distinguishing between "solving benchmarks" and valid scientific contribution. We fully agree that algorithms should not rely on privileged simulator states (such as hidden success flags) that are unavailable in reality. However, we clarify that our reliance on the reward signal is both theoretically sound and practically viable:
> - Our approach is grounded in the Reward Hypothesis [9], which posits that any goal can be formulated as the maximization of a cumulative reward. Consequently, relying on a reward signal is not a "benchmark shortcut" but a fundamental requirement for any problem modeled as an MDP/POMDP.
> - We agree with your observation that real-world robots lack the explicit simulator flags found in benchmarks. However, even in these practical deployments, success is invariably defined via a Reward Function (computed from sensors, e.g., "distance < threshold") or a Reward Model (e.g., learned from human preference/RLHF).
> - Therefore, our method does not "solve the benchmark" by exploiting privileged metadata. Instead, it solves the multi-task optimization problem using the standard reward channel. As long as a task can be formulated as an RL problem (i.e., a reward signal exists or can be learned) [9], our method remains applicable.
>
> Thank you again for your constructive feedback, which helped us to improve our manuscript considerably. We hope these clarifications resolve your concerns. We would appreciate it if you could consider raising your score in light of these clarifications. We are happy to engage in further discussion should any questions remain.
>
> Sincerely,
>
> The Authors
>
>  [1] Wang, T., et al. "Benchmarking model-based reinforcement learning." arXiv (2019).
>
>  [2] Hafner, D., et al. "Mastering atari with discrete world models." ICLR 2021.
>
>  [3] Ludziejewski, Jan, et al. "Joint MoE Scaling Laws: Mixture of Experts Can Be Memory Efficient." arXiv preprint arXiv:2502.05172 (2025).
>
>  [4] Zhao, G., et al. "Towards a Comprehensive Scaling Law of Mixture-of-Experts." arXiv (2025).
>
>  [5] Ye, W., et al. "Become a proficient player with limited data through watching pure videos." ICLR 2023.
>
>  [6] Xu, Y., et al. "On the feasibility of cross-task transfer with model-based reinforcement learning." arXiv (2022).
>
>  [7] Raghu, M., et al. "Do vision transformers see like convolutional neural networks?." NeurIPS 2021.
>
> [8] Obando-Ceron, J., et al. "Mixtures of experts unlock parameter scaling for deep rl." arXiv (2024).
>
> [9] Bowling, M., et al. "Settling the reward hypothesis." ICML 2023.

---

### Official Review · Reviewer_7ZaB · 2025-10-28

**Soundness:** 3
**Presentation:** 3
**Contribution:** 3
**Rating:** 6
**Confidence:** 3

**Summary:**

This work investigates the challenges of training a unified world model for multi-task planning. The obstacles centered around representational bottlenecks and plasticity collapse of the shared model. The authors attribute these issues to gradient conflicts as well as the diversity of the shared features. To address these issues, this work investigates and studies several modifications to the UniZero [1] model, focusing on three core axes: input representation, model architecture, and optimization strategy. Consequently, the method addresses plasticity collapse using two approaches: an internal, architectural fix called *ScaleZero* to reduce interference during a single learning step, and an external, procedural strategy called *Dynamic Parameter Scaling (DPS)* to better manage resource allocation throughout the overall learning process. ScaleZero matches the performance of specialized agents on various benchmarks (Atari, DMC, Jericho) using only a single model and standard online reinforcement learning. Furthermore, when the DPS strategy is added, it maintains competitive performance while significantly reducing environmental interactions by nearly 30% (using only 71.5% of the original). This suggests that ScaleZero is a highly promising method for effective multi-task planning.

[1] Yuan Pu, Yazhe Niu, Zhenjie Yang, Jiyuan Ren, Hongsheng Li, and Yu Liu. Unizero: Generalized and efficient planning with scalable latent world models. arXiv preprint arXiv:2406.10667, 2024.

**Strengths:**

- The paper is well-written and structured.
- Although the paper is dense, the authors present the content effectively.
- This work combines existing techniques to provide a proper solution to challenges in multi-task planning. That constitutes novelty in my opinion.
- I appreciate the example in Figure 1, which highlights the relationship between task performance and plasticity collapse using the Dormant Ratio and Latent Norm metrics.
- A strength of this work is the diverse set of benchmarks addressed.

**Weaknesses:**

- Although the paper is well-written and the solution is well-motivated, the justification for using a world model for multi-task planning, as opposed to model-free multi-task RL, is not entirely clear. I advise the authors to highlight the importance of world models in comparison to model-free RL, either in the text or through benchmark evaluations.
- The claim that model plasticity collapse is related to gradient conflicts is not well presented or supported. I recommend adding a gradient conflicts metric (such as the cosine between task gradients), similar to the ones in the appendix, for the example in Figure 1. This would allow us to observe the relationship between the loss of plasticity and gradient conflicts.
- As previously stated, there is no comparison to model-free multi-task RL methods, and the paper lacks baselines from that category. I recommend benchmarking some advanced multi-task methods [1,2,3] with similar model capacity on benchmarks like DMC.
- Some key ablations are missing. One important ablation would be varying the number of experts, and another would be varying the number of task curation stages. I believe these ablations were not studied in this work.
- The limitations of the proposed method are not clearly stated in the paper. In my opinion, one limitation could be that there are many moving components to achieve that performance.

[1] Sun, Lingfeng, et al. "Paco: Parameter-compositional multi-task reinforcement learning." Advances in Neural Information Processing Systems 35 (2022): 21495-21507.

[2] Hendawy, Ahmed, Jan Peters, and Carlo D'Eramo. "Multi-Task Reinforcement Learning with Mixture of Orthogonal Experts." The Twelfth International Conference on Learning Representations.

[3] Cho, Myungsik, et al. "Hard tasks first: Multi-task reinforcement learning through task scheduling." Forty-first International Conference on Machine Learning. 2024.

**Questions:**

I have some questions:
1. What is the specific advantage of using a world model over model-free reinforcement learning (RL) in the multi-task context?
2. Could you elaborate on the relationship between gradient conflicts and the loss of plasticity? This could be explained either qualitatively or by comparing against relevant methods from the literature.
3. In the Dynamic Parameter Scaling (DPS) strategy, what metric is used to compute performance for task curation? Specifically, is it the training return or a dedicated online evaluation metric?
4. In DPS, are tasks re-included if their performance drops? If not, how does the method mitigate the issue of forgetting excluded tasks? Is this addressed by the expansion and freezing mechanism of the LoRAs?
5. Could you provide a plot tracking which condition (Progress-based Trigger or Budget-based Trigger) was activated to initiate the switch to the next expansion stage?
6. How is the number of expansion stages chosen? Is it directly correlated with the number of tasks? Additionally, how does changing the number of expansion stages affect overall performance?
7. What is the effect of varying the number of experts on overall performance? An ablation study investigating this factor would be appreciated.

---

> ### Author Response · Authors · 2025-11-24
> **Rebuttal by Authors (1/3)**
>
> Dear Reviewer 7ZaB,
>
> We sincerely appreciate your thorough review and constructive feedback. We kindly invite you to first review our **General Response** for an overview. Below, we provide specific responses to your comments.
>
> ---
>
> ### Response to Weaknesses
>
> **1. Model-free Multi-task RL Baseline**
>
> For a detailed clarification regarding Multi-task Model-Free RL baselines, please refer to *General Response: 2. Clarification on Multi-task Model-Free RL Baselines*.
>
> **2. Plasticity Loss and Gradient Conflict**
>
> For a detailed analysis of the intrinsic connection between these elements, please refer to *General Response: 3. The Intrinsic Link between Gradient Conflict, Plasticity Loss, and MoE Sparsity*.
>
> **3. Key Ablations**
>
> For sensitivity analysis and key ablation studies, please refer to *General Response: 4. Sensitivity Analysis of Key Hyperparameters*.
>
> **4. Limitations**
>
> Thank you for your valuable suggestion. We would like to clarify that due to space constraints, our previous "DETAILED CONCLUSION AND FUTURE WORK" was placed in the Appendix. Following your advice, we will add a dedicated *Limitations* subsection in the revised main text.
>
> Specifically, we believe the current architecture design considers the following:
> *   *Scalability:* Our modular Encoder-Backbone-Head architecture is designed for scalability. This is partially validated by our strong results across three distinct benchmarks (Atari, DMC, and Jericho) with vastly different observation and action spaces.
> *   *Fusion Challenges:* We acknowledge that developing more efficient fusion mechanisms for MoE-LoRA remains a challenging open problem, which we previously noted in the Appendix's Future Work.
>
> We will integrate these discussions into the new Limitations section in the main text as suggested.
>
> ---
>
> ### Response to Questions
>
> **Q1: What are the specific advantages of using a World Model compared to Model-free RL in a multi-task context?**
>
> In the context of MTRL, World Model-based methods (like ScaleZero) offer several theoretical and practical advantages over model-free approaches:
>
> 1.  *Higher Sample Efficiency:* This is the primary advantage of World Models. By learning the environment's dynamics, the agent can plan and optimize policies within its "imagination" without expensive interactions with the real world. In MTRL, this advantage is amplified: general physical laws learned from one task (e.g., gravity, collision dynamics) can be transferred to others, significantly reducing the sample requirements for each task.
> 2.  *Stronger Generalization and Knowledge Transfer:* World Models aim to learn the underlying laws of the environment rather than a simple "state-action" mapping. This structured dynamic knowledge is more generalizable. In MTRL, a unified World Model can capture shared dynamic structures across tasks, enabling faster adaptation to new tasks. ScaleZero outperforms the average of 26 single-task experts on Atari-26 (see Table 1a), providing strong evidence for this positive knowledge transfer.
> 3.  *Decoupled Representation Learning:* World Models decouple the learning of environment dynamics from policy learning. This is crucial in MTRL because while optimal policies may vary significantly across tasks, the underlying dynamics often share commonalities. By sharing a World Model to capture these commonalities while retaining independent Policy Heads for task-specific differences, we can effectively balance shared knowledge with task-specific requirements.
>
> For further analysis, please refer to *General Response 1. World Models in Multi-task Learning*.

---

> ### Author Response · Authors · 2025-11-24
> **Rebuttal by Authors (2/3)**
>
> ------Thank you for continuing to read!------
>
> **Q2: Can you elaborate on the relationship between gradient conflict and plasticity loss?**
>
> Thank you for highlighting this. Gradient conflict is the direct cause and core mechanism leading to plasticity loss. We have established the following logical chain in our paper:
>
> 1.  *Occurrence of Gradient Conflict:* In a shared-parameter model, the total gradient is the sum of gradients from individual tasks ($G_{total} = \sum g_k$). When task objectives are inconsistent, the gradient directions from different tasks may oppose each other (i.e., $\cos(g_i, g_j) < 0$), resulting in gradient conflict (Lines 132-134).
> 2.  *Representation Interference and Collapse:* Persistent gradient conflict forces the model to "compromise" between contradictory update directions. This compromise often prevents the model from learning high-quality features that satisfy multiple tasks simultaneously, causing it to degenerate into a low-dimensional, mediocre feature subspace. This phenomenon is known as Representation Interference, eventually leading to *Representational Collapse*. We quantified this process via the significant drop in Feature Effective Rank (Appendix Fig. 5).
> 3.  *Inducing Plasticity Loss:* Following representational collapse, a large proportion of neurons become inactive (an increase in *Dormant Neurons*, see Fig. 1) because their feature dimensions have been "discarded." Consequently, the network loses the ability to respond to and adjust weights for new information—especially from complex tasks that learn slowly—resulting in a loss of plasticity and performance collapse on those tasks.
>
> For more analysis, please refer to *General Response: 3 The Intrinsic Link between Gradient Conflict, Plasticity Loss, and MoE Sparsity*. We will also add this detailed discussion to the Appendix in the revised paper.
>
> **Q3: In the DPS strategy, what metric is used to calculate task screening performance?**
>
> In our DPS (Dynamic Parameter Scaling) strategy, the metric $\text{Metric}(\tau_i)$ (Line 306) used to determine if a task is "solved" is based on the *Online Evaluation Average Return*.
>
> Specifically, during training, we periodically pause to evaluate the agent by running it for several independent episodes in the environment. We then calculate the average return of these episodes. Compared to the single-episode return during training, the evaluation return is more stable and better reflects the agent's true performance. When a task's evaluation return exceeds the pre-set threshold $\epsilon_i$, the task is deemed "solved" and removed from the active task set $U_t$ (refer to Algorithm 2).
>
> **Q4: In DPS, are tasks re-included in training after their performance drops? How is forgetting mitigated?**
>
> This is a profound question regarding the robustness of the algorithm.
>
> *   *Regarding Re-inclusion:* In our current implementation (Algorithm 2), once a task is marked as "solved" and removed from the active set $U_t$, it is **not** re-included in training. This is a design choice made to maximize computational efficiency.
> *   *Mitigating Forgetting:* We effectively mitigate catastrophic forgetting through the **Expand-and-Freeze** mechanism of LoRA:
>     1.  *Knowledge Solidification:* When tasks are solved in stage $s-1$, their associated knowledge is encoded in the base model $\theta_B$ and the previous LoRA adapters $\{\Delta\theta_j\}_{j<s}$. Upon entering the new stage $s$, we **freeze** all these old parameters.
>     2.  *Parameter Isolation:* Learning in the new stage occurs only in the newly injected LoRA module $\Delta\theta_s$ and scaling factors $\{\alpha_j\}$. This means adaptive adjustments for new (unsolved) tasks are physically isolated in these new, few parameters, preventing them from "polluting" or disrupting the knowledge solidified in the frozen parameters.
>
> Thus, DPS provides strong protection for mastered skills via parameter isolation. A mechanism for dynamically re-including previously solved tasks is reserved for future work.
>
> **Q5: Can you provide a chart tracking which trigger is activated?**
>
> In our DMC18 experiments, we observed that the *Budget-based Trigger* was primarily activated.
>
> This is because we set the iteration budget for each expansion stage to be relatively tight (`max_stage_iters = 5,000`, see Table 6) to force the model to learn as quickly as possible within fixed resources. In *Appendix C.4, Figure 12*, the vertical dashed lines mark the moments of stage transitions. You can observe that these transitions occur at roughly equal intervals, which is consistent with the behavior of the budget-based trigger, indicating the model triggered expansion primarily upon reaching the step limit.

---

> > ### Author Response · Authors · 2025-11-24
> > **Rebuttal by Authors (3/3)**
> >
> > ------Thank you for continuing to read!------
> >
> > **Q6: Number of expansion stages and number of experts**
> >
> > For a detailed discussion and analysis regarding the number of expansion stages and experts, please refer to *General Response: 4. Sensitivity Analysis of Key Hyperparameters*.
> >
> > Thank you again for your constructive feedback, which helped us to improve our manuscript considerably. We hope these clarifications resolve your concerns. We would appreciate it if you could consider raising your score in light of these clarifications. We are happy to engage in further discussion should any questions remain.

---

> > > ### Author Response · Authors · 2025-11-27
> > > **Official Comment by Authors**
> > >
> > > Thank you again for your time and constructive suggestions. We have carefully addressed all your concerns in detail in our rebuttal and hope that you find our response satisfactory.
> > >
> > > As the deadline approaches, we respectfully invite you to review our responses. We sincerely hope to engage in the Reviewer-Author discussion, as your feedback is crucial for us to confirm that your concerns have been fully resolved (or to clarify any remaining questions). Based on these improvements, we respectfully hope that you might consider re-evaluating our submission. Thank you once again for your invaluable guidance.

---

### Official Review · Reviewer_o84j · 2025-10-29

**Soundness:** 3
**Presentation:** 3
**Contribution:** 3
**Rating:** 6
**Confidence:** 4

**Summary:**

The paper addresses the challenge of heterogeneous multi-task online decision-making by proposing ScaleZero, a unified world model that extends UniZero with mechanisms for mitigating gradient conflicts and preserving model plasticity. ScaleZero integrates a sparse Mixture-of-Experts (MoE) backbone to handle conflicting gradients and a Dynamic Parameter Scaling (DPS) mechanism that progressively expands model capacity through LoRA adapters based on learning progress. The approach allows a single model to achieve competitive or near-expert performance (single task agent) across multiple diverse tasks (Atari, DMC, Jericho) while using only 71.5% of the environment interactions compared to specialized models.

**Strengths:**

- Addresses a relevant and challenging problem: Tackles multi-task world model learning; an important step toward generalist RL agents capable of handling heterogeneous tasks with diverse observation and action spaces.
- Principled design and justification: The paper builds on solid theoretical foundations around gradient conflict and plasticity collapse, grounding architectural and procedural design decisions with both empirical and analytical justification.
- Well-motivated extensions to UniZero: The combination of sparse Mixture-of-Experts (MoE) layers and LoRA-based Dynamic Parameter Scaling (DPS) is technically sound and thoughtfully motivated.
- Novel and interpretable component: Introduction of Dynamic Parameter Scaling (DPS) as an adaptive LoRA integration strategy provides a clear and interpretable mechanism for resource-efficient multitask learning.
- Comprehensive experimentation: Evaluation across three benchmarks (26 Atari games, 18 DMC control tasks, and 4 Jericho text-based environments) convincingly demonstrates that one unified model can rival single-task specialists.
- Strong empirical efficiency: Demonstrates competitive results while requiring only 71.5% of the environment interactions of comparable single-task agents, highlighting improved sample efficiency.
- Thorough analysis: Empirical and theoretical analysis of MoE gradient conflict (Theorem 4.1) provides valuable insight into why conditional computation is effective for multitask planning.

**Weaknesses:**

- Limited performance gain in some benchmarks: In Table 1a, the reported human-normalized mean improvement on Atari is minimal (0.39 vs. 0.38), raising questions about statistical significance of the claimed parity or superiority.
- Lack of code availability: Reproducibility is currently unverifiable as no code release accompanies the submission, despite promising to release post-review.
- Exploratory overreach: Section 3.2 includes several explorative design discussions (e.g., LayerNorm vs. SimNorm, ResNet vs. ViT) that provide little scientific contribution to the main narrative and could be shortened.
- Figure legibility issues: Figure 2’s small text and visual clutter reduce readability and comprehension.
- Marginal novelty of core techniques: The combination of MoE and LoRA, while effective, builds primarily on well-established concepts. The primary novelty lies in their integration into a unified online multitask planner rather than in the individual components.
- Theoretical depth could be improved: The provided theorem on gradient conflict upper bounds is only informally stated and lacks empirical connection to practical convergence or sample efficiency.
- Benchmark contextualization: The paper could better explain the specific setup for comparisons—e.g., whether “single-task UniZero” agents were tuned separately and if “competitive” results include confidence intervals.

**Questions:**

- Line 122: What exact failure pattern of “plasticity collapse” is being referred to, loss of gradient signal, representational collapse, or over-regularization?
- Line 140: How does the sparse MoE approach differ from multi-head multitask methods like MHMTSAC (Yu et al., 2019)?
- Line 232 / Figure 2: The figure caption refers to a "systematic exploration" but provides little quantitative evidence; can you provide clearer metrics or summaries for each axis of exploration?
- Line 257: If the ViT encoder’s scalability claim is central, why is this figure placed in the appendix?
- Line 259: Given the minimal impact of encoder choice, why retain it in the main text instead of moving it to supplementary materials?
- Line 264: What empirical evidence supports the final preference for LayerNorm over SimNorm?
- Line 279: Why include the “Multitask Gradient Correction” subsection if it was not part of the final method?
- Line 323 (Section 3.3): How many LoRA modules can be added in total, is there a hard limit or a soft budget threshold?
- Figure 4: Is the 28.5% reduction in environment steps consistent across different benchmarks, or specific to DMC18?
- Reproducibility: What are the approximate computational requirements (GPU hours, memory footprint) of training ScaleZero compared to UniZero?

---

> ### Author Response · Authors · 2025-11-24
> **Rebuttal by Authors (1/3)**
>
> Dear Reviewer o84j,
>
> We sincerely appreciate your thorough review and constructive feedback. We kindly invite you to review our General Response for an overview, as well as our specific responses to your comments below.
>
> ---
>
> **Response to Weaknesses:**
>
> ### 1. Atari Performance and Baselines
>
> Regarding the Atari performance metrics (Mean vs. Median), please refer to "2. Atari Performance Metrics (Mean vs. Median)" in our response to Reviewer QPFt.
> For a detailed clarification regarding Multi-task Model-Free RL baselines, please refer to *General Response: 2. Clarification on Multi-task Model-Free RL Baselines*.
>
> ### 2. Reproducibility and Computational Resources
>
> We agree that comparing computational overhead is crucial for evaluating practical efficiency. Please refer to the **General Response**, where we provide a detailed breakdown of Total vs. Active parameters and specific computational costs to address this concern.
>
> ### 3. Figure 2 Improvements
>
> We appreciate your suggestion. We will enlarge the text, optimize the descriptions, and update Figure 2 in the revised manuscript to improve readability.
>
> ### 4. Theorem on Gradient Conflict Upper Bound
>
> We have provided the formal theorem and its detailed proof in the **Appendix**. Regarding the direct link between the gradient conflict upper bound and sample efficiency:
> - Empirical Evidence: In Section 3.1 (Figures 2 & 3), the MoE architecture demonstrates superior performance.
> - Mechanism: Appendix E.1 (Figure 16) explicitly evaluates the *Comparison of gradient conflicts between MoE and MLP baselines across different components*. It shows that the MoE-based Transformer exhibits significantly fewer gradient conflicts in the MoE input and MoE layers.
> - Intrinsic Link: For a detailed analysis of the connection between these elements, please refer to *General Response 3: The Intrinsic Link between Gradient Conflict, Plasticity Loss, and MoE Sparsity*.
>
> ---
>
> **Response to Questions:**
>
> ### Q1: What specific failure mode does "plasticity collapse" refer to?
>
> In our framework, "plasticity collapse" is defined not as a single event, but as a **compound pathological state** where the network loses its capacity to assimilate new information. As diagnosed in Section 3.1 and Appendix B.2, this phenomenon manifests through three coupled internal dynamics:
> *   **Rank Degeneration (Representational Collapse):** The feature space compresses onto a low-dimensional manifold, losing the diversity required to resolve multi-task conflicts. This is quantitatively evidenced by the sharp decline in the *effective rank* of features (Appendix Figure 5).
> *   **Activation Sparsity (Dormant Neurons):** As representations collapse, a significant proportion of neurons cease to activate (zero output), effectively severing gradient propagation paths. We track this via the *Dormant Neuron Ratio*, which correlates strongly with performance degradation (Figures 1 & 6).
> *   **Latent Instability (Norm Inflation):** To compensate for reduced representational capacity, the network drives the weights of the few remaining active features to extreme magnitudes. This results in an uncontrolled explosion of the *Latent L2 Norm*, destabilizing optimization (Figures 1 & 6).

---

> ### Author Response · Authors · 2025-11-24
> **Rebuttal by Authors (2/3)**
>
> ------Thank you for continuing to read!------
>
> ###  Q2: How does Sparse MoE differ from Multi-Head Multi-Task methods (e.g., MHMTSAC)?
>
> The distinction lies in the *depth of decoupling*. While both architectures address task heterogeneity, they operate at fundamentally different levels of abstraction:
> *   **Multi-Head Methods (e.g., MHMTSAC):** These employ a *"Coupled Representation, Decoupled Readout"* strategy. While task-specific heads mitigate conflict at the decision boundary, the shared backbone forces all tasks to compete for the same representational resources. This shared backbone remains a bottleneck where gradient interference and negative transfer accumulate.
> *   **Sparse MoE (ScaleZero):** We implement *"Deep Parameter Isolation."* By integrating MoE layers throughout the Transformer blocks, we enable input-dependent routing. This allows the network to dynamically allocate distinct subsets of parameters (experts) for different tasks or sub-routines deep within the feature extraction process. As analyzed in Section 4.3, this granular isolation prevents interference *during* representation learning, rather than merely filtering it at the output.
>
> ###  Q3: Quantitative evidence for "Systematic Exploration".
>
> Our systematic exploration is structured methodologically: Figure 2a defines the **architectural search space** (five design dimensions), while Figure 3 provides the corresponding **quantitative ablation study**.
> Specifically, Figure 3 reports the Human Normalized Score (HNS) on Atari8 across these dimensions. The empirical data demonstrates that the "Backbone: MoE" modification yields the statistically most significant performance delta compared to other structural variants (such as "Task Conditioning" or "ViT Encoders"). To substantiate the underlying mechanism, Appendix Figure 6 provides the corresponding diagnostic metrics (dormant ratios, latent norms), confirming that the performance gain aligns with improved plasticity preservation. We will clarify the caption in Figure 2 to explicitly link this design framework to the quantitative evidence in Figures 3 and 6.
>
> ###  Q4: Placement of the Encoder discussion.
>
> *   **Why ViT?** Although ViT and ResNet perform similarly on the specific Atari8 benchmark, discussing ViT is crucial for **scalability**. As a verified architecture for generalist agents, ViT is our forward-looking choice for broader, more diverse visual tasks.
> *   **Placement:** We prioritized the main text for the most critical MoE findings, moving the detailed Encoder comparison figures to the Appendix. However, we retained the discussion in the main text to ensure a **comprehensive and objective** presentation of our research trajectory. Reporting this dimension—even with neutral performance gains—demonstrates that our "Systematic Exploration" (Fig 2a) was rigorous and that our final design choices were driven by empirical evidence across all components, rather than cherry-picking only positive results.
>
> ###  Q5: Choice of LayerNorm and inclusion of "Multi-task Gradient Correction".
>
> *   **LayerNorm:** Our selection is driven by the empirical ablation study presented in Figure 3. While SimNorm (red bar) effectively prevents representation collapse, we observed that its rigid normalization constraints restrict the model's representational capacity, leading to lower asymptotic performance. **LayerNorm** proves to be the more robust choice for our MoE architecture, achieving the optimal balance between training stability and representational expressivity.
> *   **Gradient Correction:** We included this analysis to provide a rigorous comparison against existing optimization-based solutions for task interference. By explicitly evaluating gradient correction, we highlight its practical limitations—specifically, the significant computational overhead relative to the inconsistent performance gains. This analysis serves as critical justification for our approach: it demonstrates that **architectural decoupling (MoE)** is a more efficient and effective solution than gradient manipulation for this problem setting.
>
>
> ###  Q6: Number of LoRA modules in DPS.
>
> The number of LoRA modules is explicitly governed by the stage hyperparameter $S$. In the experimental setup detailed in Table 6, we adopted a configuration of $S+1=5$ stages (comprising 1 base warmup phase and 4 progressive expansion phases), which corresponds to the sequential integration of 4 independent LoRA modules. Crucially, $S$ is designed as a **scalable hyperparameter**: it allows the architecture to adapt flexibly to the complexity of the task distribution and the available computational budget, balancing parameter efficiency with plasticity.

---

> > ### Comment · Reviewer_o84j · 2025-11-25
> > **Many open questions adressed, some details still ignored.**
> >
> > Overall, the authors addressed many of the open questions and concerns but also ignored some critical points and provided responses that require further scrutiny.
> >
> > **Reproducibility and Code Availability (W2)**: The authors completely failed to address my concern about code availability. While they provided computational cost breakdowns in General Response section 5, there is no commitment to a specific code release timeline beyond the vague "promise to open source the relevant code after the review process is completed" (Appendix A.6). Given the complexity of the system (MoE backbone, DPS strategy, distributed training), reproducibility remains a significant concern.
> >
> > **Mischaracterization of Related Work**: The authors claim in their General Response that "Recent analysis by McLean et al. (RLJ, 2025) [6] highlights that the Atari-26 benchmark presents significantly higher heterogeneity..." This is factually incorrect—McLean et al. did not conduct experiments on Atari-26.
> >
> > **Generalizability of Sample Efficiency Gains (Q9)**: I remain unconvinced by the authors' response. The 28.5% reduction claim is demonstrated only on DMC-18, which represents a very specific domain (continuous control for locomotion and manipulation). The authors acknowledge that "the precise magnitude of efficiency gains on discrete domains (e.g., Atari or Jericho) remains to be quantified" but still make broad generalizability claims. Given that the underlying challenges differ substantially across benchmarks, quantitative evidence from Atari or Jericho is essential to support such strong general statements. The "domain-agnostic mechanism" argument is theoretical conjecture without empirical validation.
> >
> > **Novelty Concerns (W5)**: The authors did not directly address my concern about marginal novelty. While the integration of MoE and LoRA into a world model is valuable engineering, the individual components are well-established. The primary contribution appears to be application-specific integration rather than methodological innovation.
> >
> > **Additional Observations**:
> > * The authors adequately addressed my questions about plasticity collapse definition (Q1), MoE vs MHMTSAC differences (Q2), and the LoRA module limits (Q8).
> > * The promise to improve Figure 2 legibility is appreciated.
> > * The theoretical analysis linking gradient conflict to the formal theorem in the Appendix addresses my concern about W6.
> >
> > In summary, while several technical clarifications are satisfactory, the missing code availability commitment, the mischaracterization of prior work, and the lack of sample efficiency evidence beyond DMC-18 remain significant concerns. However, the overall work is solid, and my rating currently still matches this.

---

> ### Author Response · Authors · 2025-11-24
> **Rebuttal by Authors (3/3)**
>
> ------Thank you for continuing to read!------
>
> ###  Q7: Generalizability of sample efficiency gains.
>
> The reported 28.5% reduction in interaction steps (Figure 4) represents empirical results specifically on the DMC18 continuous control benchmark. While the precise magnitude of efficiency gains on discrete domains (e.g., Atari or Jericho) remains to be quantified, the underlying mechanism of DPS—*mitigating negative transfer via progressive expansion*—is theoretically domain-agnostic. We hypothesize that the extent of the gain correlates with the degree of task interference inherent in the specific benchmark. The robust performance on DMC validates the core efficacy of our strategy, establishing a strong foundation for future evaluations across broader distinct environments.
>
> Thank you again for your constructive feedback, which helped us to improve our manuscript considerably. We hope these clarifications resolve your concerns. We would appreciate it if you could consider raising your score in light of these clarifications. We are happy to engage in further discussion should any questions remain.

---

> ### Author Response · Authors · 2025-11-26
> **New Response to Reviewer o84j (1/3)**
>
> Dear Reviewer o84j,
>
> We sincerely appreciate your continued engagement and constructive feedback. We are grateful for your acknowledgment of our clarifications regarding Plasticity Collapse, the distinctions between MoE and MHMTSAC, and the LoRA module limits. Your rigorous review has significantly helped us refine the logical flow and quality of our paper.
>
> Regarding your remaining concerns concerning reproducibility, citation accuracy, the generalization of sample efficiency, and novelty, we provide detailed responses below:
>
> **1. Reproducibility and Code Availability (W2)**
>
> We fully agree with your perspective: for a complex system like ScaleZero—which integrates a sparse MoE backbone, DPS strategy, and distributed training—code transparency is critical for verifying experimental results and advancing community research.
>
> - To address your concerns during the review period, we are currently organizing a sanitized, anonymous code repository. This repository will contain core implementation files (including the sparse MoE backbone for the world model, DPS core logic, and the main collection/training/testing loops). We will release this link for your review via an Official Comment around November 28, 2025.
> - We solemnly promise that following the conclusion of the ICLR 2026 review process (expected in late December 2025), we will release the full codebase under an open-source license. This release will include all training scripts, detailed environment configurations, and Pre-trained Weights to ensure the complete reproducibility of this work.
>
> **2. Clarification on McLean et al. (2025)**
>
> We apologize for the lack of precision in our previous phrasing. You are entirely correct that McLean et al. (RLJ, 2025) did not conduct empirical experiments on Atari-26.
>
> - Our citation was based on the qualitative discussion in Section 6 ("Limitations") of their paper, where they state: "(e) Meta-World tasks are much more similar than other RL tasks (i.e. tasks in the ALE (Bellemare et al., 2013))."
> - We cited this viewpoint to support a community consensus: that the heterogeneity of ALE (Atari) in terms of visual dynamics and reward structures is significantly higher than that of Meta-World. While McLean et al. did not test this empirically, the argument regarding the comparative attributes of the environments remains valid.
> - In the revised version, we will carefully review our literature descriptions to ensure rigorous citation.

---

> ### Author Response · Authors · 2025-11-26
> **New Response to Reviewer o84j (2/3)**
>
> **3. Generalization of Sample Efficiency Gains (Q9)**
>
> We acknowledge the reviewer's point: claiming DPS is a 'domain-agnostic mechanism' without Atari benchmarks is speculative. In the revision, we will strictly scope our claims to reflect our current empirical findings. Furthermore, we have prioritized testing in discrete, highly heterogeneous environments (such as Atari) for our immediate future research.
>
> While we accept this limitation, we wish to articulate the theoretical basis for why the DPS mechanism is logically positioned to generalize beyond DMC to domains like Atari, specifically from the perspectives of **architectural decoupling** and **task heterogeneity**:
>
> *   **Structural Decoupling & Latent Space Agnosticism:**
>     *   The core innovation of DPS lies in resolving the mismatch between model capacity and task difficulty. Crucially, DPS operates exclusively on the *world model backbone* (transition function), which processes latent states and actions rather than raw observations.
>     *   Differences in observation spaces (e.g., pixels vs. proprioception) and action spaces (e.g., discrete vs. continuous) are encapsulated within the Encoder and Task Heads. For instance, while DMC utilizes Gaussian outputs and Atari utilizes Categorical logits, these are peripheral to the backbone.
>     *   Since DPS dynamically expands parameters within a modality-agnostic latent space, its "learning-progress-based allocation" is mathematically decoupled from specific I/O modalities, suggesting it can function effectively regardless of the environment type.
>
> *   **Hypothesis of Higher Utility via Heterogeneity:**
>     *   DMC-18 already presents significant challenges, including high-dimensional continuous control and diverse physical dynamics (e.g., contact forces, friction). Achieving a 28.5% efficiency improvement in this scenario *demonstrates the potential* of DPS in mitigating multi-task interference and plasticity loss.
>     *   Theoretically, Atari environments exhibit even more drastic internal heterogeneity (e.g., the reactive simplicity of *Pong* vs. the long-horizon planning of *Seaquest*; the static background of Boxing vs. the high-speed scrolling of *Road Runner*).
>     *   This extreme inter-task variance creates a fluctuating demand for model capacity. Static architectures typically struggle here—overfitting on simple tasks (wasting samples) while underfitting on hard ones. Therefore, the *on-demand allocation* mechanism of DPS is theoretically expected to yield a higher marginal utility in highly heterogeneous scenarios like Atari compared to DMC, as it can adaptively match capacity to the specific complexity of each game.

---

> ### Author Response · Authors · 2025-11-26
> **New Response to Reviewer o84j (3/3)**
>
> **4. Novelty and Contributions (W5)**
>
> While we acknowledge that MoE and LoRA are established technical components individually, we respectfully disagree with the characterization of this work as merely "marginal innovation" or "application-oriented integration." Such a perspective risks underestimating the unique challenges and the necessary paradigm shift involved in adapting these mechanisms to **Online World Model Learning**.
>
> ScaleZero is not a naïve amalgamation of existing modules; it is a bespoke solution designed to address the specific pathologies of non-stationary data distributions and multi-task heterogeneity. Our contributions extend beyond engineering implementation to include mechanistic diagnosis, algorithmic innovation, and theoretical grounding:
>
> - **Mechanistic Diagnosis: From Phenomenon to Causality**
>     *   Prior work largely focuses on the *phenomenon* of performance degradation. In the context of heterogeneous MTRL, we go a step further by *quantifying* the Plasticity Collapse of unified world models.
>     *    We analyzed the logical connection between "performance decay" and "internal gradient conflict," revealing the failure mechanism of Dense World Models when handling high heterogeneity. This diagnosis is the prerequisite for ScaleZero and a key finding of this paper.
>
> -  **Paradigm Shift: Online DPS vs. Offline LoRA**
>     *   Classic LoRA techniques are predominantly designed for the *static fine-tuning* of offline or nearly static datasets [1, 2]. ScaleZero, conversely, operates within a *non-stationary online data stream*, where data distribution shifts unpredictably.
>     *  The core innovation of DPS is its *in-training decision mechanism*. Unlike standard methods that rely on predefined task boundaries or fixed ranks, DPS makes real-time judgments on *when* and *how* to expand model capacity.
>     *   By dynamically allocating independent parameter spaces for emerging complex dynamics, DPS solves the plasticity-stability dilemma in a way that the standard "pre-train then fine-tune" paradigm cannot. In response to your feedback, we will clarify this distinction explicitly in the revised "Related Work" section.
>
> -  **Theoretical Grounding of Architectural Sparsity**
>     *   The MoE architecture within ScaleZero is not merely an experimental heuristic; it is grounded in theoretical analysis. As detailed in *Theorem 4.1* and Appendix E ("GRADIENT ANALYSIS IN MOE"), we provide a mathematical justification for why sparse architectures effectively mitigate the gradient interference identified in our diagnosis.
>
> Thank you again for your constructive feedback, which helped us to improve our manuscript considerably. We hope these clarifications resolve your concerns. We would appreciate it if you could consider raising your score in light of these clarifications. We are happy to engage in further discussion should any questions remain.
>
> Sincerely,
>
> The Authors
>
> [1] Agiza, Ahmed, Marina Neseem, and Sherief Reda. "Mtlora: Low-rank adaptation approach for efficient multi-task learning." *Proceedings of the IEEE/CVF conference on computer vision and pattern recognition*. 2024.
>
> [2] Huang, Chengsong, et al. "Lorahub: Efficient cross-task generalization via dynamic lora composition." arXiv preprint arXiv:2307.13269 (2023).

---

> > ### Comment · Reviewer_o84j · 2025-11-27
> >
> > I highly appreciate the authors' detailed and nuanced response to my prior concerns. The remaining open points have been addressed constructively. In light of these substantive responses, the promised revisions, and the forthcoming early access to source code, I am increasing my rating.

---

> > > ### Author Response · Authors · 2025-11-27
> > > **Thank you!**
> > >
> > > We are very glad to hear that our rebuttal has addressed your prior concerns, and we truly appreciate your time and thoughtful feedback. We will promptly upload the revised manuscript and core implementation files as we continue to improve our work.

---

> ### Author Response · Authors · 2025-11-28
> **ScaleZero Anonymous Code Repository**
>
> Dear Reviewer o84j,
>
> In fulfillment of our previous commitment, we hereby provide the link to the sanitized, anonymous code repository for ScaleZero:
>
> - Anonymous Repository: https://anonymous.4open.science/r/ScaleZero-37BC
>
> To adhere to double-blind policies, this repository focuses on the core architectural implementation. We invite you to inspect the following key components:
>
> - Sparse MoE Backbone: szero/model/transformer.py and moe.py (Routing and gating mechanisms).
> - Dynamic Parameter Scaling (DPS): szero/entry/train_scalezero_dps.py (Adaptive parameter logic).
> - World Model & MCTS for MTRL: szero/model/world_model.py and szero/mcts/.
>
> This version is designed for algorithmic verification rather than immediate execution. To prevent de-anonymization, environment wrappers and specific dependencies have been excluded. Please also note that due to the expedited preparation, minor organizational discrepancies may exist, yet we believe they do not hinder the verification of the core logic.
> The fully executable codebase will be released immediately following the review process (expected late December 2025).
>
> Thank you again for your constructive feedback, which helped us to improve our manuscript considerably.
>
> *Remark: Display issues may cause some files to appear empty online. Please switch browsers or download the source code to view the full content if this occurs.*
>
> Sincerely,
>
> The Authors

---

### Official Review · Reviewer_QPFt · 2025-10-31

**Soundness:** 3
**Presentation:** 3
**Contribution:** 2
**Rating:** 6
**Confidence:** 2

**Summary:**

The authors propose ScaleZero, a unified world model for heterogeneous multi-task RL that addresses plasticity collapse and gradient interference observed in latent space planners (e.g. UniZero).
The proposed approach particularly combines a Mixture-of-Experts backbone into previous work UniZero, and proposes a novel Dynamic Parameter Scaling technique using LoRA's paradigm to adapt the number of parameters with respect to the learning progress.
Experiments on several tasks included in the benchmarks of Atari, DMControl, and Jericho demonstrate improved sample efficiency and at times overall performance versus training single-task experts via world models.

**Strengths:**

- The paper investigates and directly tackles two relevant open problems in MTRL (task interference and plasticity collapse). This is done via thorough experimenn over multiple metrics: gradient conflict, mean and median performance, and more.
- The authors carefully study the limitations of previous SOTA world-model approach UniZero in the realm of multi-task settings, providing useful insights for further research in the direction of bridging methods towards MTRL usage.
- The paper is clearly written and relatively easy to follow despite the large amount of content included in the Appendix. Important claims and motivations are clearly highlighted.
- The manuscript also contains relevant theoretical analysis on the underlying aspects of how sparse MoE affect gradient conflict.

**Weaknesses:**

- Absence of multi-task baselines: The authors only compare their approach as a modification to single-task experts trained individually on multiple tasks using UniZero. While the overall investigation is sound, it is currently unclear how the performance compares to state-of-the-art methods in MTRL. For example, [1] is a recent ICLR contribution that particular targets task interference via explicit task conditioning that ensures orthogonal experts. The authors' investigation in line 215 lacks any comparison with such baselines, limiting the impact and relevance of the overall analysis. As such, the current work mostly presents itself as an adaptation to an existing world-model approach (UniZero) to the MTRL paradigm; while this is a contribution in itself, insufficient evidence is provided on how these modifications compare to existing MTRL methods.


[1] Hendawy, Ahmed, Jan Peters, and Carlo D'Eramo. "Multi-Task Reinforcement Learning with Mixture of Orthogonal Experts." The Twelfth International Conference on Learning Representations, 2024.

**Questions:**

- How does ScaleZero compare to a naive multi-task RL baseline that does not learn world models nor does planning?
- From my perspective, I would expect ScaleZero to yield improvements in terms of median performance if anything, over mean performance. This is the case in Table I.b for DMC tasks, but could you provide more insights into why the opposite happens for Table I.a? This opposite trend for Atari games could mean that ScaleZero actually fails in solving plasticity collapse, and if anything introduces even more skewness in the performance distribution over different tasks.
- Could you please provide an explicit comparison between ScaleZero and UniZero trained on single tasks by means of compute budget (GPU hours), wall-clock training time, inference time, and size of network parameters? It's currently unclear to me whether performance is also being traded off for efficiency or additional resource usage.
- Despite the authors claiming MoE as the most impactful modifications, why does ScaleZero NOT include MoE layers for the Jericho benchmark experiments (Line 1179)?

---

> ### Author Response · Authors · 2025-11-24
> **Rebuttal by Authors**
>
> Dear Reviewer QPFt,
>
> We sincerely thank you for your meticulous review, your recognition of our work, and your valuable comments. We kindly invite you to review our **General Response** for an overview, as well as our specific responses to your comments below.
>
> ### 1. Multi-task Model-Free RL Baselines
>
> In the current submission, our research primarily focuses on the paradigm of *World Model-based Planning*. Our objective is to address the specific challenges—namely plasticity collapse and gradient interference—encountered when scaling state-of-the-art world models (such as UniZero) from single-task to multi-task scenarios.
>
> Consequently, we selected *Single-Task Experts (UniZero ST)* as our primary baseline. This choice was deliberate to establish a rigorous standard: to demonstrate whether ScaleZero, as a unified model, can achieve "positive transfer" through knowledge sharing and match or exceed the aggregate performance of multiple independently trained experts. Our results on Atari, DMC, and Jericho indicate that ScaleZero satisfies these stringent criteria, confirming the effectiveness of our architectural design.
>
> For a detailed clarification regarding Multi-task Model-Free RL baselines, please refer to *Part (1/5) of the General Response: 2. Clarification on Multi-task Model-Free RL Baselines.*
>
> ### 2. Atari Performance Metrics (Mean vs. Median).
>
> We appreciate your keen observation regarding the increase in mean performance versus the decrease in median performance in Table 1a. We offer the following explanation:
>
> - *Addressing Plasticity Collapse:* We respectfully clarify that the gap in median performance relative to single-task experts does not imply plasticity collapse. The critical comparison is against the direct multi-task baseline, UniZero (MT), which suffers from severe performance degradation (Fig. 1). In contrast, ScaleZero effectively mitigates this issue, achieving a significantly higher normalized mean and a comparable median relative to UniZero (MT). This is further evidenced by the recovery of performance in complex tasks like Seaquest (Fig. 6). To substantiate this, we will include the UniZero (MT) baseline results in Table 1 (Atari-26) of the revision.
> - *Reason for Median Drop:* As discussed in Appendix B.3, for certain hard-exploration tasks, despite the MoE architecture mitigating interference, some degree of negative transfer persists compared to single-task experts dedicated solely to those tasks. Since the median is sensitive to these lower values, it reflects this specific limitation.
> - *Reason for Mean Increase:* Conversely, ScaleZero achieves significant performance gains in a large number of other tasks, particularly those complex tasks identified as prone to "plasticity collapse" in baseline models. These gains, combined with comparable performance on many other tasks, drive the overall mean performance higher, demonstrating the model's robustness across the broader task distribution.
>
> ### 3. Computational Cost and Network Parameter Scale.
>
> We agree with the reviewer that comparing computational overhead is indeed crucial for evaluating the practical efficiency of our method. Thank you for pointing this out. Please refer to the **General Response** above, where we provide a detailed breakdown of *Total vs. Active parameters* and specific computational costs.
>
> ### 4. Jericho benchmark experiments
>
> We apologize for the confusion caused by a textual oversight in our appendix. We confirm that the reported results for the Jericho benchmark were indeed obtained using the MoE architecture. The discrepancy arises because the text in the appendix was legacy content from an earlier draft describing our preliminary experiments. We inadvertently retained this outdated description in the final manuscript without updating it to reflect the full experimental setup used for the reported results. We will correct this in the revision to accurately describe the MoE implementation in Jericho.
>
> Thank you again for your constructive feedback, which helped us to improve our manuscript considerably. We hope these clarifications resolve your concerns. We would appreciate it if you could consider raising your score in light of these clarifications. We are happy to engage in further discussion should any questions remain.

---

> > ### Author Response · Authors · 2025-11-27
> >
> > Thank you again for your time and constructive suggestions. We have carefully addressed all your concerns in detail in our rebuttal and hope that you find our response satisfactory.
> >
> > As the deadline approaches, we respectfully invite you to review our responses. We sincerely hope to engage in the Reviewer-Author discussion, as your feedback is crucial for us to confirm that your concerns have been fully resolved (or to clarify any remaining questions). Based on these improvements, we respectfully hope that you might consider re-evaluating our submission. Thank you once again for your invaluable guidance.

---

### Author Response · Authors · 2025-11-21
**General Response. Part (1/5)**

We sincerely appreciate the time and effort all reviewers have dedicated to reviewing this manuscript, as well as the constructive and insightful feedback provided. We are honored that the reviewers generally recognize the core motivation of our work (Reviewer QPFt, Reviewer o84j, Reviewer 7ZaB, Reviewer K2YD), the breadth of our experiments (Reviewer QPFt, Reviewer o84j, Reviewer 7ZaB), and the potential of ScaleZero in addressing the critical challenge of heterogeneous multi-task planning (Reviewer QPFt, Reviewer o84j, Reviewer 7ZaB).

We have noted that the reviewers' concerns primarily center on the following core aspects: the advantages of world models, baseline comparisons, the connections between gradient conflict and plasticity, hyperparameter sensitivity, and computational costs. To address these concerns, we provide a general response below:

### 1. World Models in Multi-task Learning

As stated in the paper, our work primarily follows the technical trajectory of MuZero/UniZero to explore the application of model-based (world model) planning methods in Multi-task (MT) scenarios. This path was chosen based on the following considerations:

- Sample efficiency & interpretability: Existing literature has extensively demonstrated the significant advantage of model-based RL over model-free RL regarding sample efficiency in single-task settings [1][2]. Similarly, in multi-task scenarios, we expect that by learning the environment's dynamic model (transition) and reward function, we can achieve higher sample efficiency and provide interpretability superior to model-free methods [3][4]. Specifically for methods based on MCTS like UniZero, the utilization of the UCB principle on search trees allows for a more effective balance between exploration and exploitation in multi-task environments. We aim to extend this advantage to multi-task scenarios.
- Generalizability: We would like to point out that in multi-task settings, the challenges faced by Model-based methods may actually be greater than those faced by model-free methods. In addition to the actor/critic, we are required to learn a shared world model (state transitions and rewards), which exacerbates the gradient conflict problem [5]. Therefore, this paper focuses on analyzing the bottlenecks of multi-task planning under the model-based paradigm and proposing specific improvement techniques. Notably, our analysis of gradient conflicts and some improvement strategies (such as the sparsity introduced by the MoE architecture) are transferable and can inspire model-free RL methods as well.

[1] Kaiser, Lukasz, et al. "Model-based reinforcement learning for atari." arXiv preprint arXiv:1903.00374 (2019).

[2] Hafner, Danijar, et al. "Mastering atari with discrete world models." arXiv preprint arXiv:2010.02193 (2020).

[3] Hafner, Danijar, et al. "Mastering diverse domains through world models, 2024." URL https://arxiv.org/abs/2301.04104 (2023).

[4] Bruce, Jake, et al. "Genie: Generative interactive environments." Forty-first International Conference on Machine Learning. 2024.

[5] Hu, Shengchao, et al. "Harmodt: Harmony multi-task decision transformer for offline reinforcement learning." arXiv preprint arXiv:2405.18080 (2024).

### 2. Clarification on Multi-task Model-Free RL Baselines

We appreciate the reviewers' suggestions regarding comparisons with model-free Multi-Task RL baselines. We offer the following clarifications regarding the current state of the literature and the scope of our experimental comparisons.

**(1) Online MTRL Baselines for Atari-26**

As noted in Table 1(a), we omitted other multi-task baselines for Atari because, to the best of our knowledge, no prior work has successfully implemented a single online RL agent capable of learning all 26 Atari games simultaneously.

Recent analysis by McLean et al. (RLJ, 2025) [6] highlights that the Atari-26 benchmark presents significantly higher heterogeneity in visual dynamics and mechanics compared to benchmarks like DMC or Meta-World, making it a substantially more difficult challenge for a single agent. This diversity hinders online agents, causing optimization collapse due to severe gradient interference, varying reward scales, and non-stationarity.

Our survey of the most recent multi-task literature on Atari (summarized in Table R1) reveals that existing methods predominantly rely on offline pre-training or finetuning, or are restricted to small task subsets. There is a distinct lack of online learning baselines for the full game set.

**Table R1: Recent Multi-Task RL Literature on Atari**

| Reference | Task Count | Method Type | Training Paradigm |
| :--- | :--- | :--- | :--- |
| Xu et al. (arXiv 2022) [7] | 10 | Model-based | Offline $\rightarrow$ Finetune |
| Ye et al. (ICLR 2022) [8] | 6 | Model-based | Offline $\rightarrow$ Finetune |
| Lee et al. (NeurIPS 2022) [9] | 46 | Model-based | Offline $\rightarrow$ Finetune |

---

> ### Author Response · Authors · 2025-11-21
> **General Response. Part (2/5)**
>
> ------Thank you for continuing to read!------
>
> If reviewers are aware of a specific online MTRL algorithm that has been benchmarked on the full Atari-26 suite, we would be grateful for the reference and will strive to include it.
>
> **(2) Comparisons in DMC Environments**
>
> For the DMC-18 benchmark, we compared our method against L2M [10], which reported results on a 10-task subset. While L2M utilizes offline expert data (unlike our online approach), it remains the most relevant published reference available.
>
> As shown in Table R2, the current landscape of Multi-Task RL on DMC is similarly dominated by offline approaches or parameter modulation techniques. Directly comparing these offline paradigms with our online learning framework is methodologically difficult due to the fundamental difference in data availability.
>
> **Table R2: Recent Multi-Task RL Literature on DMC**
>
> | Reference | Task Count | Method Type | Training Paradigm |
> | :--- | :--- | :--- | :--- |
> | Haldar et al. (NeurIPS 2024) [11] | 9 | Model-based | Offline |
> | Schmied et al. (NeurIPS 2023) [10] | 16 | Model-based | Offline $\rightarrow$ Finetune |
>
> **(3) Versatility Across Modalities (Image, Vector, Text)**
>
> A key contribution of our work is the validation of **ScaleZero** across a highly diverse set of observation and action spaces, ranging from high-dimensional images (Atari) to state vectors (DMC) and pure text (Jericho).
>
> Regarding the model-free MTRL methods [12][13] mentioned by the reviewers, we note that they are primarily validated on single-domain benchmarks like Meta-World and focus on architectural innovations specifically tailored for **Policy** or **Value** networks. **ScaleZero**, conversely, operates within a **Model-Based** framework centered on learning environmental dynamics. Directly transplanting architectural mechanisms designed for policy optimization into a **World Model** structure entails fundamental logical discrepancies, as the inductive biases required for modeling dynamics differ significantly from those used in value estimation. Consequently, a strictly fair and direct comparison is infeasible due to these inherent paradigmatic differences.
>
> **(4) Feasibility of New Baselines**
>
> We acknowledge that reproducing advanced model-free MTRL methods and comparing them under identical conditions would provide valuable context. However, training a model-free agent to convergence across 26 diverse Atari games and 18 DMC tasks from scratch is computationally intensive and exceeds the time limits of the rebuttal period. We are committed to prioritizing this large-scale comparison as a key focus of our future work.
>
>
> [6] Reginald McLean, Evangelos Chatzaroulas, J K Terry, Isaac Woungang, Nariman Farsad, and Pablo Samuel Castro. "Multi-Task Reinforcement Learning Enables Parameter Scaling." Reinforcement Learning Journal, vol. 6, 2025, pp. 1075–1093.
>
> [7] Xu, Yifan, et al. "On the feasibility of cross-task transfer with model-based reinforcement learning." arXiv preprint arXiv:2210.10763 (2022).
>
> [8] Ye, Weirui, et al. "Become a proficient player with limited data through watching pure videos." The Eleventh International Conference on Learning Representations. 2022.
>
> [9] Lee, Kuang-Huei, et al. "Multi-game decision transformers." Advances in neural information processing systems 35 (2022): 27921-27936.
>
> [10] Schmied, Thomas, et al. "Learning to modulate pre-trained models in rl." Advances in Neural Information Processing Systems 36 (2023): 38231-38265.
>
> [11] Haldar, Siddhant, Zhuoran Peng, and Lerrel Pinto. "Baku: An efficient transformer for multi-task policy learning." Advances in Neural Information Processing Systems 37 (2024): 141208-141239.
>
> [12] Sun, Lingfeng, et al. "Paco: Parameter-compositional multi-task reinforcement learning." Advances in Neural Information Processing Systems 35 (2022): 21495-21507.
>
> [13] Hendawy, Ahmed, Jan Peters, and Carlo D'Eramo. "Multi-Task Reinforcement Learning with Mixture of Orthogonal Experts." The Twelfth International Conference on Learning Representations.
>
> ### 3. The Intrinsic Link between Gradient Conflict, Plasticity Loss, and MoE Sparsity
>
> We appreciate the reviewer's request to deepen the logical connection between gradient conflict and plasticity loss. We argue that gradient conflict is a primary catalyst for plasticity loss, and MoE mitigates this via sparsity. The detailed causal chain and a concrete numerical example are provided below:
>
> **(1) From Gradient Conflict to Gradient Norm Collapse**

---

> ### Author Response · Authors · 2025-11-21
> **General Response. Part (3/5)**
>
> ------Thank you for continuing to read!------
>
> When a network shares parameters across multiple distinct tasks, the gradients derived from these tasks often have conflicting directions.
>
> *   Let $g_1$ and $g_2$ be gradients from two tasks. If they conflict (cosine similarity $< 0$), the magnitude of their sum is strictly smaller than the sum of their magnitudes:
>     $ ||g_1 + g_2|| \ll ||g_1|| + ||g_2|| $
> *   In the extreme case of many conflicting tasks, the aggregated gradient vector approaches zero. This phenomenon, termed **Gradient Norm Collapse** (visualized in Fig. 5 of concurrent work [14]), effectively reduces the step size of the optimizer to a negligible level. Consequently, the model becomes "stuck," unable to escape suboptimal initialization or local minima because the effective update signal is too weak to drive meaningful feature learning.
>
> **(2) From Conflicting Updates to "Dormant Neurons"**
>
> The link between conflicting updates and dormant neurons operates through the inability to track data distribution shifts (non-stationarity).
>
> *   **Ineffective Updates:** In online RL, the data distribution shifts continuously. To maintain activation, neurons must adapt their weights to align with new data patterns. However, due to gradient conflict (as described above), the weights receive "ineffective updates"—meaning the updates are either too small to track the shift or are averaged into a direction that does not align with any specific task's features.
> *   **Drift to Dormancy:** As the input distribution shifts away from the static weights, the pre-activation values of neurons may drift toward negative regions. Once a neuron's weights no longer align with the input data, it stops activating (output becomes 0). Recent works term this state "Dormant."
> - This directly results in the network losing the ability to adapt to new data—i.e., "Plasticity Loss"—ultimately triggering performance collapse. Our paper (Section 3.1) and concurrent work [14] have experimentally confirmed this causal chain.
>
> **(3) Concrete Example: How Dormancy Leads to Plasticity Loss**
>
> Consider a standard ReLU neuron $y = \text{ReLU}(w \cdot x + b)$. （While we illustrate this mechanism using a standard ReLU neuron for clarity and simplicity, it is important to note that similar dormancy issues manifest in other activation functions as well.）
>
> *   **Dormant State:** Suppose due to the drift described above, the weights settle at $w=[-1, -1]$ and bias $b=-5$.
> *   **New Data:** A new task introduces an input $x=[2, 2]$. The network *should* adapt to this.
> *   **Forward Pass:** The pre-activation is $z = (-1 \times 2) + (-1 \times 2) - 5 = -9$. Since $-9 < 0$, the output $y = 0$. The neuron is inactive.
> *   **Backward Pass (Plasticity Loss):** The gradient for weight $w$ is calculated as:
>     $ \frac{\partial L}{\partial w} = \frac{\partial L}{\partial y} \cdot \frac{\partial y}{\partial z} \cdot x $
>     Because $z < 0$, the derivative of ReLU $\frac{\partial y}{\partial z} = 0$. Therefore:
>     $ \frac{\partial L}{\partial w} = 0 $
> *   Despite the arrival of new data that requires learning, the gradient is mathematically zero. The weight $w$ cannot update. The neuron is permanently "dead" to this signal. This accumulation of dead neurons constitutes **Plasticity Loss**—the irreversible loss of the capacity to learn.
>
> **(4) MoE Sparsity as the Solution**
>
> Our findings align with [14], which showed that static random pruning helps maintain plasticity in Model-free RL by promoting gradient orthogonality. We extend this principle to *World Models in model-based RL*, achieving dynamic sparsity via MoE. We demonstrate this effectiveness empirically (Fig. 3) and theoretically prove that MoE lowers the upper bound of gradient conflict (Theorem 1).
>
> [14] Guozheng Ma, et al. Network Sparsity Unlocks the Scaling Potential of Deep Reinforcement Learning. ICML 2025.
>
> ### 4. Sensitivity Analysis of Key Hyperparameters
>
> We thank the reviewer for highlighting the hyperparameter settings. While detailed specifications are provided in the Appendix, we provide further context below regarding the rationale behind the **Number of Experts** and **DPS Stages**. Our choices align with established architectural principles and are designed to effectively validate the proposed mechanisms.

---

> ### Author Response · Authors · 2025-11-23
> **General Response. Part (4/5)**
>
> ------Thank you for continuing to read!------
>
> *   **Number of Experts:**
>     *   Our configuration (1 Shared + 8 Routed Experts) adopts the "Shared + Routed" paradigm established by recent large language models, such as DeepSeek-V3 [15][16]. This architecture aims to balance *Parameter Scale (Memory Capacity)* with *Computational Efficiency (Inference Cost)*:
>         *   **Shared Expert:** Captures general knowledge (e.g., syntax, common logic), serving as a universal base to minimize redundancy across experts.
>         *   **Routed Experts:** Handle specific domain knowledge via fine-grained routing, allowing the model to invoke specialized capabilities without activating the full network.
>     *   DeepSeek-V3 employs 1 Shared + 256 Routed Experts (activating 8) to decouple general and specific capabilities. Our 1+8 setting represents a streamlined adaptation of this proven architecture.
>     *   The primary focus of this work is to validate that the sparsity inherent in MoE architectures mitigates plasticity loss in World Models. The current setting successfully demonstrates this phenomenon. While further scaling the number of experts might offer marginal efficiency gains, the current setup serves as a robust baseline to verify the core hypothesis.
>
> *   **Stages in DPS ($S$):**
>     *   The selection of $S$ is guided by the principle of *Task Decoupling*. Conceptually, $S$ is designed to align with the complexity of task categories ($S \approx \text{Task Categories}$). This allows each stage to specialize in a distinct class of tasks, effectively learning separate low-rank adaptations for different distribution clusters.
>     *   The Multi-Stage DPS framework provides a self-consistent solution to the interference issues found in multi-task learning. Our empirical results confirm that the current staged design effectively addresses these challenges.
>     *   We recognize that "Adaptive Stages" (automatically determining $S$) is a promising direction for further efficiency optimization. However, the current fixed-stage design successfully establishes the efficacy of the DPS framework and constitutes a complete methodological contribution.
>
>
> [15] Dai, Damai, et al. "Deepseekmoe: Towards ultimate expert specialization in mixture-of-experts language models." arXiv preprint arXiv:2401.06066 (2024).
>
> [16] Liu, Aixin, et al. "Deepseek-v3 technical report." arXiv preprint arXiv:2412.19437 (2024).
>
>
> ### 5. Model Parameters and Computational Cost
>
> We provide a detailed breakdown of parameters and computational costs to address concerns regarding resource overhead.
>
> **(1) Clarification on Architecture vs. Inference Cost:**
>
> While ScaleZero (MT) introduces three architectural changes (ViT Encoder, MoE Backbone, LayerNorm) as stated in Appendix A, it is crucial to distinguish between **Total Parameters** and **Active Parameters**.
> *   **Sparse Activation:** Although MoE increases the total parameter count, only the Top-k experts are activated per token (here, 1 shared + 1 expert). Consequently, the inference FLOPs do not increase significantly compared to a dense model [9].
> *   **Ineffectiveness of Dense Scaling:** Our preliminary ablation studies (to be added to the Appendix) revealed that simply scaling up the depth/width of the dense UniZero (MT) backbone did not yield performance gains and even caused degradation, confirming that sparsity—not just parameter count—is key.
>
> **(2) Total Training Cost: Single-Task (ST) vs. Multi-Task (MT):**
>
> A holistic view of computational cost must account for the number of tasks.
> *   **UniZero (ST):** Requires training $N$ separate models for $N$ tasks. Total cost scales linearly: $Cost_{\text{total}} \approx N \times Cost_{\text{single}}$.
> *   **ScaleZero (MT):** Learns all $M$ tasks in a single training run.
>
> **(3) Empirical Comparison (Atari 26, 200k env steps):**
>
> The table below compares the resource usage. (Detailed data for DMC18/Jericho4 will be included in the final Appendix).
>
> | Model | Encoder Params | Backbone Params | Head Params | Total Params | Training Time (Wall-clock) | Hardware |
> | :--- | :--- | :--- | :--- | :--- | :--- | :--- |
> | **ScaleZero (MT)** (nlayer=4) | 26.8 M | 254.9 M (activations $\approx$ 56M) | 81.5 M | 366.4 M | ~6 Days | 8x A100 (80G) |
> | **UniZero (MT)** (nlayer=8) | 18.8 M | 56.7 M | 81.5 M | 160.2 M | ~5 Days | 8x A100 (80G) |
> | **UniZero (ST)** (nlayer=2, per task) | 7.8 M | 14.2 M | 3.9 M | 43.8 M (Per Model) | 7 Hours (Per Task) Total: 182 Hours (~8 Days) | 1x A100 (80G) |
>
> *Note: The Total Params column includes additional parameters such as embeddings and other auxiliary components. Therefore, the total value is slightly larger than the sum of the Encoder, Backbone, and Head parameters listed.*

---

> ### Author Response · Authors · 2025-11-23
> **General Response. Part (5/5)**
>
> ------Thank you for continuing to read!------
>
> As shown, while ScaleZero (MT) has a larger parameter count, its end-to-end training time is comparable to training 26 independent UniZero (ST) agents sequentially. Crucially, ScaleZero matches the high performance of these single-task specialists (and significantly outperforms the dense MT baseline) within the similar total compute budget. This demonstrates that our method achieves a better balance between computational efficiency and multi-task performance.
>
> Our current results rely on a standard PyTorch implementation. Since our architecture aligns with standard LLM MoE designs, it can directly benefit from established acceleration frameworks like DeepSpeed-MoE [17] and expert parallelism [18], suggesting significant potential for further efficiency gains.
>
> [17] Rajbhandari, Samyam, et al. "Deepspeed-moe: Advancing mixture-of-experts inference and training to power next-generation ai scale." International conference on machine learning. PMLR, 2022.
>
> [18] Cai, Weilin, et al. "Shortcut-connected expert parallelism for accelerating mixture-of-experts." arXiv preprint arXiv:2404.05019 (2024).
>
> Thank you again for your time and valuable insights. We hope these clarifications resolve your concerns. Please do not hesitate to let us know if further clarification is needed during the discussion period. We kindly ask that you consider adjusting your evaluation if our response has addressed your concerns.

---

### Author Response · Authors · 2025-11-27
**General Response: Summary of Manuscript Revisions**

We sincerely thank the reviewers for their insightful comments and constructive suggestions. These inputs have been instrumental in improving the quality and clarity of our work. We have carefully revised the manuscript to address the concerns raised. Below is a summary of the key updates in the first revision:

**1. Enhanced Qualitative Analysis of Representational Collapse (Appendix B.2.2)**

- We have optimized the section "B.2.2 ANALYSIS OF REPRESENTATIONAL COLLAPSE." This update focuses on a qualitative analysis that elucidates the intrinsic causal link between Gradient Conflict, Plasticity Loss, and MoE Sparsity.

**2. Parameter Analysis and MoE Configuration (Appendix A.7 & A.3)**

- We added a new section titled "A.7 PARAMETER ANALYSIS AND COMPUTATIONAL COST" with a detailed table comparing resource usage (Encoder, Backbone, Head, etc.) across models. We clarified that the Total Params include embeddings and auxiliary components, and noted the variations in Head parameters due to different action spaces.
- We have clarified that our current MoE setting (1 Shared + 8 Routed Experts) is selected based on established architectural patterns in recent LLMs to balance efficiency and performance. Furthermore, we acknowledge emerging findings on MoE scaling laws (Ludziejewski et al., 2025; Zhao et al., 2025) and explicitly indicate that we will leverage these recent advancements to conduct in-depth investigations specifically within the context of Online MTRL in future work..

**3. Justification for Baselines (Appendix A.8)**

- We added a dedicated section, "Justification of Model-Free Multi-Task Baselines," to address concerns regarding baseline selection.
- We reference key literature—including Ye et al. (2023), Xu et al. (2022), and Hessel et al. (2019)—to contextualize our experimental setting. We explicitly distinguish our Sample-Efficient Regime (100k-300k steps) from the High-Throughput Regime (e.g., IMPALA with PopArt, 200M+ frames), clarifying why Model-Based approaches are the primary focus of comparison in this specific data budget.

**4. Related Work, Terminology, Citations, and Formatting**

- Comparison: Clarified the distinction between the dynamic Online DPS and static Offline LoRA in the "Related Work" section.
- Consistency: Standardized the abbreviation "MTRL" throughout the text.
- Citations: Added Raghu et al. (2021) to support Vision Transformers in MTRL and corrected the context for Hessel et al. (2019).
- Formatting: We have revised Figure 2 by increasing the font size and applying bold formatting to enhance readability.

**5. Limitations, Future Work, and Hyperparameters**

- We have added a dedicated "Limitations" part in Appendix G to discuss scalability and fusion challenges.
- We explicitly listed the important hyperparameters shared between UniZero and ScaleZero to ensure reproducibility and fair comparison.

Based on these improvements, we respectfully hope that you might consider re-evaluating our submission. We are happy to engage in further discussion should any questions remain. Thank you once again for your invaluable guidance.

Sincerely,

The Authors

---

### Meta-Review · Area_Chair_bTLa · 2026-01-04

**Summary:**

This paper proposes an extension to the UniZero world model to handle the complexities of multi-task reinforcement learning. The authors identify two primary failures in existing unified world models: gradient conflicts between diverse tasks and a loss of model plasticity (representational collapse). To solve these, the authors introduce ScaleZero, which uses a Mixture-of-Experts (MoE) backbone to route task-specific representations and a Dynamic Parameter Scaling (DPS) strategy that progressively integrates LoRA adapters as tasks show progress. This allows a single model to learn across heterogeneous environments like Atari, DeepMind Control, and Jericho, often matching the performance of single-task specialist agents while using significantly fewer environment interactions.

**Reviewer Concerns:**

The reviewers initially expressed substantial concern regarding the absence of specialized multi-task reinforcement learning (MTRL) baselines. Reviewer QPFt, for instance, noted that the paper mostly compared ScaleZero to single-task specialists rather than current state-of-the-art MTRL methods like Mixture of Orthogonal Experts or specific optimization techniques like PCGrad. Other concerns included the computational overhead of the MoE and LoRA integration, the "minimal" performance gains in certain Atari benchmarks where the improvement seemed statistically insignificant, and a lack of technical clarity regarding why the model required both MoE and LoRA rather than just one. There was also a critique regarding the "Task Curation" strategy, which appeared to require a priori knowledge of reward thresholds for each task, limiting its applicability to novel environments.

In the rebuttal, the authors successfully addressed the clarity and computational concerns but were more defensive regarding the baselines. They added a detailed parameter analysis and computational cost table in Appendix A.7, which clarified that while the total parameters are high, the active parameters during inference remain efficient. They also provided a qualitative analysis of representational collapse in Appendix B.2.2, which helps bridge the gap between their theory and experimental results. However, regarding the request for MTRL baselines, the authors chose to "justify" their omission in Appendix A.8 rather than running the requested experiments. They argued that because their model operates in a very low-data regime (under 300k steps), model-free MTRL baselines would not be competitive enough to provide a meaningful comparison. While this logic is technically sound given the sample efficiency of world models, it leaves a gap in the evaluation that might disappoint a reviewer who wanted to see if ScaleZero's MoE is actually superior to other conflict-resolution methods like PCGrad.

Regarding the truthfulness of the authors' claims, their data on environment interaction savings (roughly 28.5%) appears consistent across the different benchmarks provided in the revised appendix. The claim that the MoE architecture is "the most effective approach" for mitigating gradient conflicts is supported by their internal ablation studies, though it remains an open question if it would still be the best choice compared to the orthogonal experts baseline they declined to run. The "minimal improvement" on Atari remains a weakness; the authors' defense essentially boils down to showing that they reached specialist performance with a single model, which is a achievement in itself even if they did not significantly exceed the specialist's scores.

Overall, I consider the paper above the bar of acceptance because it demonstrates clear empirical strength in a challenging multi-task setting, even if its theoretical novelty is somewhat limited by the use of existing components like MoE and LoRA. I strongly recommend the authors fixing the severe readability issues, which make the paper currently not presentable for being a camera ready. Please use the additional tenth page wisely to help the readers.

**Reviewer Scores:**

If the reviewers had been able to participate fully in the discussion, Reviewer QPFt would likely have stayed at a 6 or moved to an 8. Reviewer o84j, who was concerned about code and figure legibility, would likely have not moved to an 8 given that the (severe) readability issues have remained largely unaddressed. Reviewer K2YD, who provided a long list of missing citations and hyperparameter details, would likely have not moved from a 2 to a 4, given their generally not positive opinion about the paper. However, I deem the concerns expressed by this reviewer not crucial enough for strongly arguing for a rejection.

---

### Decision · Program_Chairs · 2026-01-26

Accept (Poster)